https://doi.org/10.1038/s42003-022-03184-4　　**OPEN**
# A comparative whole-genome approach identifies bacterial traits for marine microbial interactions

Luca Zoccarato [1✉], Daniel Sher [2✉], Takeshi Miki[3], Daniel Segrè [4,5] & Hans-Peter Grossart [1,6,7✉]

Microbial interactions shape the structure and function of microbial communities with profound consequences for biogeochemical cycles and ecosystem health. Yet, most interaction mechanisms are studied only in model systems and their prevalence is unknown. To systematically explore the functional and interaction potential of sequenced marine bacteria, we developed a trait-based approach, and applied it to 473 complete genomes (248 genera), representing a substantial fraction of marine microbial communities. We identified genome functional clusters (GFCs) which group bacterial taxa with common ecology and life history. Most GFCs revealed unique combinations of interaction traits, including the production of siderophores (10% of genomes), phytohormones (3–8%) and different B vitamins (57–70%). Specific GFCs, comprising Alpha- and Gammaproteobacteria, displayed more interaction traits than expected by chance, and are thus predicted to preferentially interact synergistically and/or antagonistically with bacteria and phytoplankton. Linked trait clusters (LTCs) identify traits that may have evolved to act together (e.g., secretion systems, nitrogen metabolism regulation and B vitamin transporters), providing testable hypotheses for complex mechanisms of microbial interactions. Our approach translates multidimensional genomic information into an atlas of marine bacteria and their putative functions, relevant for understanding the fundamental rules that govern community assembly and dynamics.

---

[1] Department of Plankton and Microbial Ecology, Leibniz Institute of Freshwater Ecology and Inland Fisheries (IGB), 16775 Stechlin, Germany. [2] Department of Marine Biology, Leon H. Charney School of Marine Sciences, University of Haifa, 3498838 Haifa, Israel. [3] Faculty of Advanced Science and Technology, Ryukoku University, 520-2194 Otsu, Japan. [4] Departments of Biology, Biomedical Engineering, Physics, Boston University, 02215 Boston, MA, USA. [5] Bioinformatics Program & Biological Design Center, Boston University, 02215 Boston, MA, USA. [6] Berlin-Brandenburg Institute of Advanced Biodiversity Research (BBIB), 14195 Berlin, Germany. [7] Institute of Biochemistry and Biology, Potsdam University, 14476 Potsdam, Germany. ✉email: luca.zoccarato@boku.ac.at; dsher@univ.haifa.ac.il; hanspeter.grossart@igb-berlin.de

Interactions among aquatic microorganisms such as symbiosis, parasitism, predation and competition, greatly shape the composition and activity of microbial communities[1–3]. In particular, interactions between heterotrophic bacteria and primary producers (phytoplankton) influence the growth of both organisms[4,5] with consequences for the ecosystem functioning and the biogeochemical cycles[6,7]. For instance, heterotrophic bacteria consume up to 50% of the organic matter released by phytoplankton, significantly affecting the dynamics of the huge pool of dissolved organic carbon in the oceans[8]. Thus, if and how a bacterium can interact with other microorganisms may have important consequences for the biological carbon pump in the current and future oceans[9,10].

Studies using model bacteria in binary co-cultures have started to elucidate the mechanisms underlying specific interactions with other marine microbes (mostly phytoplankton, but also zooplankton or other bacteria, e.g. on particles)[1,4,5,7,11]. Although these results do not reflect the complexity of natural environments and the potential for higher-order effects[12], they allow to identify the chemical signals and resulting changes in gene expression and physiology that underlie these interactions. For example, bacteria associated with phytoplankton (e.g. within the phycosphere[5,13]) gain access to labile organic carbon released by the primary producers, e.g. amino acids and small sulfur-containing compounds[14–19]. In return, phytoplankton benefit from an increased accessibility to nutrients via bacteria-mediated processes, e.g. nitrogen and phosphorus remineralization[20], vitamin supply[15,21] and iron scavenging via formation of siderophores[22,23]. In addition to such metabolic interactions, direct signalling may also occur between bacteria and phytoplankton, with heterotrophic bacteria directly controlling the phytoplankton cell cycle through phytohormones[14,24] or harming it via toxins[19,25]. Through such specific infochemical-mediated interactions, bacteria may also directly affect the release rate of organic carbon from phytoplankton, as well as rates of mortality and aggregation[19,24,26].

While much is known about how model organisms interact with other bacteria and with phytoplankton (e.g. specific strains of Roseobacter[14,19–21,25], Alteromonas[27–29], Vibrio[30,31] or Cyanobacteria[20,32]), relatively little is known regarding how widely distributed the relevant interaction mechanisms are across natural bacterial taxa. The few experimental studies that measure microbial interactions across different taxa (e.g. refs. [33–35]) are usually constrained to a fairly narrow phylogenetic scope and are performed under conditions different from natural marine environments. Conversely, relevant field studies are still quite limited (e.g. refs. [11,36]). However, the knowledge obtained from model organisms on the molecular mechanisms underlying microbial interactions and the increasing availability of high-quality genomes presents an opportunity to map known interaction mechanisms to a large set of bacterial species from various taxa. Here, we re-analyse 421 previously published genomes of diverse marine bacteria that represent a substantial fraction of marine microbial communities (213 genera), providing an atlas of their functional metabolic capacity. The atlas includes also 52 bacteria isolated from extreme marine habitats, humans and plant roots which serve as functional out-groups and/or represent well known symbiotic bacteria. Several previous studies have aimed to characterize and cluster genomes based on their predicted functional similarity defined usually using individual genes (e.g. refs. [37–40]) or coarse functional categories (e.g. COGs, refs. [39,41,42]) (Supplementary Data 1, Supplementary Note 1). We chose to take a trait-based approach rather than a gene-based one, which is an intermediate level of resolution between individual genes and coarse functions. Trait-based approaches offer a new perspective to investigating microbial functional capacity with a

more mechanistic understanding[43] but have been used only in a few specific cases to highlight putative bacterial interactions (e.g.[44]). We focused on the following traits: (1) KEGG modules representing the overall functional and metabolic capacity (i.e. pathways for the synthesis and degradation of specific biomolecules, or gene sets for processing of genetic and environmental information, cell signalling and drug resistance); (2) specific gene pathways related to the main discovered mechanisms of bacteria–bacteria and bacteria–phytoplankton interactions, such as motility, chemotaxis and the capability to produce molecules such as siderophores, phytohormones and antibiotics. The combination of these traits in individual genomes allows to classify genomes into coherent functional units, some of which recapitulate known bacterial groups with well-defined ecological roles, while others refer to potential yet undescribed groups. Furthermore, genetic traits can be grouped into linked trait clusters, representing functions that likely evolved together and maybe functionally connected (i.e. participating in the same process). Our approach maps the mechanisms of microbial interactions identified in model organisms across multiple bacterial taxa, suggests specific groups of bacteria likely to interact using similar trait combinations, and helps to hypothesise how these traits act together to mediate microbial interactions.

## Results and discussion

**Genome functional clusters (GFCs) group genomes with similar ecology.** To obtain an overview on the functional capabilities of marine bacteria, we re-annotated a set of 473 high-quality genomes and analysed them using a trait-based workflow, which focuses on the detection of complete genetic traits rather than on the presence of individual genes (Supplementary Fig. 1 and Supplementary Note 2). Genetic traits were represented by metabolic KEGG modules, secondary metabolite pathways, transporters, phytohormones and siderophores production, as well as the degradation of specific sulfur metabolites. Among the identified genetic traits, those known to mediate cell–cell interactions in bacterial model systems[4,5,45] were flagged as interaction traits (e.g. production of certain vitamins, vibrioferrin or a specific secretion system; Supplementary Data 2).

Based on the occurrence patterns of all traits (Fig. 1), we could cluster the genomes into 47 genome functional clusters (GFCs; Supplementary Data 3). In each GFC, genomes encode similar genetic traits, and thus the bacteria within each GFC are expected to be coherent in terms of functional and metabolic capacity, including the ways that they respond to abiotic cues and interact with other microbes. Previous genome comparison approaches have identified genome clusters that match ecologically relevant groups (e.g. ecotypes, as defined for Bacillus pumilus[38] and Prochlorococcus[32]) or lifestyles (e.g. oligotrophic and copiotrophic species[41]). Similarly, in our analysis, we found GFCs that represent a group of organisms with a defined ecology and life history, such as the Pelagibacterales group (GFC 2), different ecotypes of Cyanobacteria (GFCs 15 and 36), or Vibrio groups, characterized by different host-specificity and pathogenicity (GFCs 25 and 47) (Supplementary Note 3). Specific GFCs were also identified for each of three groups of Gammaproteobacteria (Alteromonas, Marinobacter and Pseudoalteromonas) which are typically considered as copiotrophs, often associated with organic particles or phytoplankton[46–49]. A detailed analysis of the traits found in each of the respective GFCs (Supplementary Fig. 2) suggested that Pseudoalteromonas and Alteromonas bore more genetic traits involved in the resistance against antimicrobial compounds, as well as regulation for osmotic and redox stresses in comparison to Marinobacter. They also had similar vitamin B1

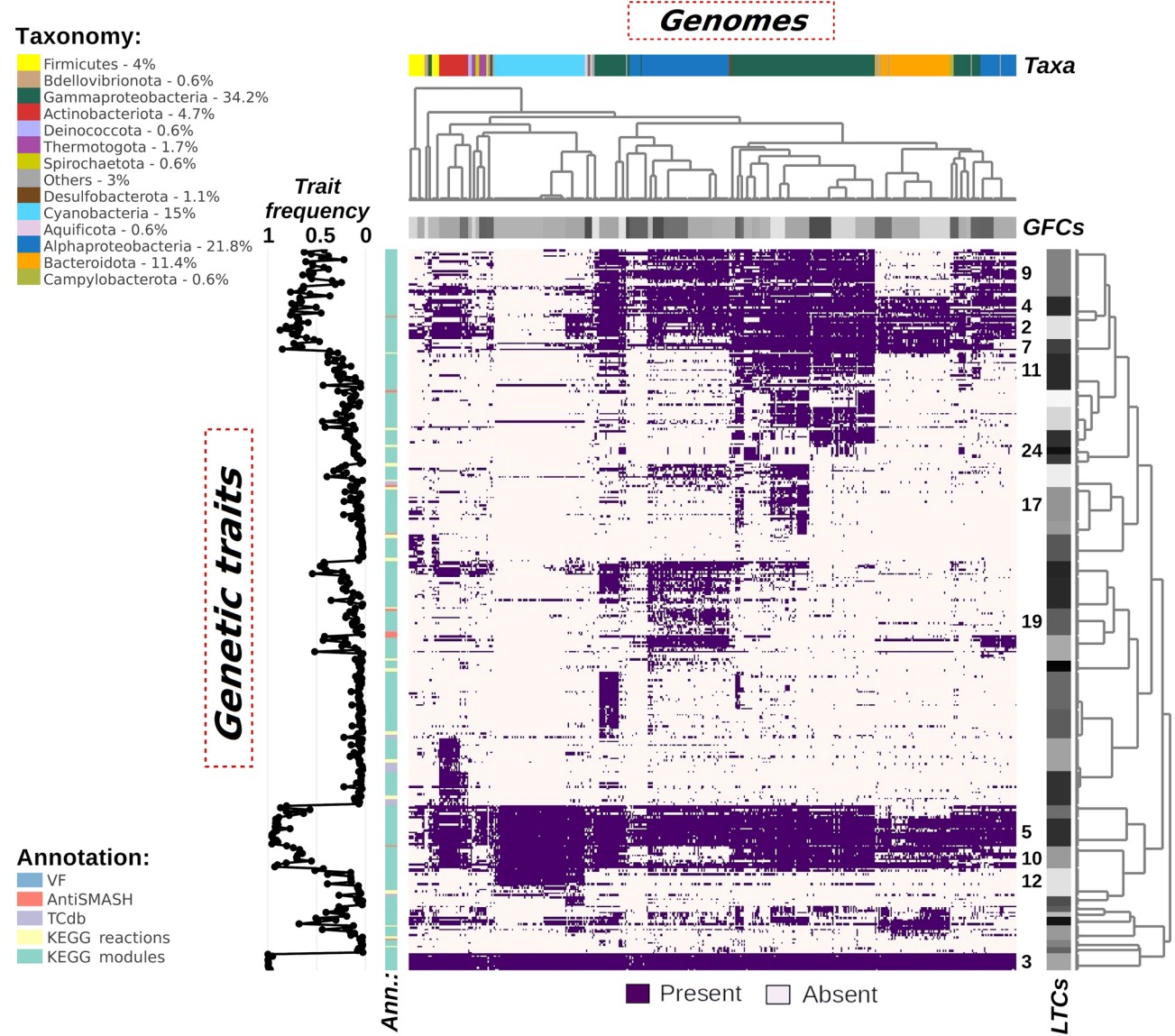

**Fig. 1 Atlas of Marine Microbial Functional Traits showing patterns of genetic traits across all analysed genomes.** Columns represent genomes grouped into genome functional clusters (GFCs) as shown by the horizontal grey bar. The horizontal colour bar represents the taxonomic affiliation of genomes (mainly phyla, with the exception of Proteobacteria that are represented at the class level). The number next to each taxon in the legend represents the percent from the total genomes analysed. Rows represent the genetic traits grouped into linked trait clusters (LTCs) as shown by the vertical grey bar. LTCs discussed in the text are labelled with the related number alongside the grey bar. The vertical line plot shows the frequency of each genetic trait across all genomes and the vertical colour bar represents the annotation tool used to identify each trait. Both dendrograms are computed using the aggExCluster function (R package apcluster) that generates hierarchal clustering from an affinity propagation result. An interactive version of this figure is available at https://doi.org/10.6084/m9.figshare.16942780.

and siderophore transporters, which are different from those encoded by *Marinobacter*. *Marinobacter* possessed several more transporters for phosphonate and amino acids, as well as specific regulatory systems for adhesion (e.g. alginate and type 4 fimbriae production) and chemotaxis. These patterns advocated that there might be coherent physiological and/or ecological differences between these three groups. Overall, our GFC framework recapitulates previous knowledge on bacterial groups with defined ecology and life history (e.g. the Pelagibacterales, different Cyanobacteria and *Vibrio*), and provides a way to delineate and characterize yet undescribed ecological groups (e.g. *Alteromonas*, *Pseudoalteromonas* and *Marinobacter*).

The correlation between functional and taxonomic classifications still represents an open question and, in marine environments, recent studies have provided both supporting[37,50] and

disproving argumentations about the strength of such correlation[39,51]. Therefore, we sought to understand the extent to which our retrieved GFCs overlapped with the genomes' taxonomy. In our analysis, we defined a GFC as taxonomically coherent (i.e. monophyletic) when all grouped genomes belonged to the same taxon and all genomes of that taxon were grouped in that GFC (see Supplementary Fig. 3 and Supplementary Note 4 for more information). Monophyletic GFCs imply that the taxonomic affiliation of these bacteria can predict the traits encoded in their genomes. According to this metric, 23 out of the 47 GFCs were taxonomically coherent, most of them at the genus level, including all Firmicutes and half of the Alpha- and Gammaproteobacteria GFCs. The remaining non-monophyletic GFCs (i.e. paraphyletic and polyphyletic), contained genomes of multiple taxa (differing at genus, family or even phylum rank) or

included taxa that were partitioned among multiple GFCs. These GFCs comprised genomes of Cyanobacteria, Bacteroidota and the remaining half of the Alpha- and Gammaproteobacteria.

Overall, half of the detected GFCs were monophyletic and support the existence of a strong correlation between taxonomy and functionality in marine bacteria, while the remaining non-monophyletic GFCs highlight that, in some cases, the taxonomic partitioning (based on the Genome Database Taxonomy[52,53]) do not completely reflect the functional differentiation. Such discrepancy may be due to processes of convergent evolution (e.g. via horizontal gene transfer) which have the highest occurrence in some of the niches known to be occupied by bacteria grouped in specific non-monophyletic GFCs (e.g. inhabiting extreme environments, particles and biofilms; see Supplementary Note 4)[54–56]. At the whole-community level, it has been shown that taxonomically distinct communities exhibit similar functional profiles, which led to the suggestion that some bacterial clades have similar genetic capacity, and can replace each other while maintaining unchanged the community functioning[57,58]. The polyphyletic GFCs may group such taxonomically different but functionally similar organisms, and this is supported by examples in GFCs 33 and 41 (grouping thermo- and halotolerant bacteria), or GFC 17 (grouping sulfur-oxidizing and facultative anaerobe bacteria). One of the main biological processes that mediate such functional homogenisation is horizontal gene transfer[59,60]. This process of genetic exchange has a higher incidence on particle/host-associated bacteria (e.g. ref. [55]) and, indeed, most of the paraphyletic GFCs group organisms with such lifestyles (e.g. Rhodobacteraceae in GFCs 9, 30 and 40, Vibrionaceae in GFCs 25 and 47, Alteromonadaceae in GFCs 21 and 24, Oleiphilaceae in GFC 35, or Halomonadaceae in GFC 43[14,19,47,61,62]). Conversely, another study suggested that this perceived similarity in community function reflects only known metabolic pathways, and it is, therefore, possible that adding to our analysis also unknown genes might separate these GFCs into monophyletic ones[63]. Although hypothetical genes would be of no use (not informative) in a trait-based approach, we hypothesize that they could complement the horizontal gene transfer hypothesis in explaining the blurred taxonomic profiles of the paraphyletic GFCs.

**GFCs are ecologically relevant entities in natural communities**. To quantify the extent of natural diversity covered by the GFCs, we mapped the 16S rDNA reads from a natural coastal community that was sampled at high temporal frequency[64] to the 16S rDNA of the GFCs (Supplementary Note 5 and Supplementary Fig. 4). Firstly, a considerable fraction of the natural community was represented with high fidelity in the GFCs (mean 22.9% of the 16S rDNA reads, range 12.7–44.3%). Thus, despite the inherent bias derived from using only high-quality, closed genomes available (mainly) from cultured bacteria (legend of Fig. 1), the GFCs represented a substantial fraction of bacterial diversity. Similar results were obtained with an open-ocean community from the Eastern Mediterranean sampled each season for 2 years[65] (mean 13.9% of the 16S rDNA reads, range 0.5-60.0%), where the GFCs represented a considerably higher fraction of the microbial community on particles >11 μm compared to free-living bacteria (5–0.22 μm; Supplementary Fig. 5). Secondly, using a temporal deconvolution analysis of the coastal site[64], we found that individual 16S phylotypes belonging to the same GFC displayed significantly more synchronous temporal trends ($p$-value < 0.001) than phylotypes belonging to different GFCs (Supplementary Note 5 and Supplementary Fig. 6). Assuming that similar temporal trends suggest similar ecological niches, these results advocate that (at least some of) the GFCs display dynamics that are expected from ecological units in the oceans.

**Specific GFCs are enriched in interaction traits**. We next focused on selected traits potentially involved in microbial interactions—vitamin exchange, siderophore and phytohormone production and antibiosis—asking whether we could observe patterns in their distribution across the genome dataset. As shown in Fig. 2, these traits are not equally distributed among the GFCs—rather, some GFCs were significantly enriched in interaction traits (Supplementary Fig. 7a, b). As the number of genes is strongly correlated with genome size[66], we expected that large genomes may encode for more interaction traits than small genomes, as previously demonstrated e.g. for the biosynthetic pathways of secondary metabolites[66,67]. However, while the number of interaction traits depended to some extent on genome size, we found that Gammaproteobacteria and several Alphaproteobacteria encoded more interaction traits than expected by their genome size, while Bacteroidota encoded fewer (Supplementary Fig. 7c, d). Overall, GFCs grouping genomes of typical host- or particle-associated bacteria, such as of (most) Alpha- and Gammaproteobacteria[62,68], are predicted to bear almost the full combination of these traits to sense (chemotaxis, quorum sensing), reach (motility and adhesion) and fight (production/resistance towards antimicrobial compounds, secretion systems) for a targeted hotspot. Conversely, some ubiquitous copiotrophs (e.g. Bacteroidota)[62,68] and known free-living taxa (e.g. pico-Cyanobacteria and Pelagibacterales)[32,69] possess only a scarce and scattered combination of such traits and are expected to exhibit a rather independent lifestyle. Below, we describe in more detail some of the main observations on the distribution of interaction traits across diversity.

**Many bacteria need to shop for their vitamins**. Vitamins $B_1$, $B_7$ and $B_{12}$ are essential cofactors for microbes. Some microorganisms (including abundant phytoplankton) are auxotrophic for these vitamins and need to obtain them from co-occurring bacteria[70,71]. Vitamins are found at low concentrations in aquatic ecosystems[72,73] and their supply can limit biogeochemical cycles, e.g. through limiting primary productivity in the Southern Ocean[74]. Less than half of all the genomes in our dataset were predicted to produce all three vitamins (~39%, including all pico-Cyanobacteria, Actinobacteriota and many Gammaproteobacteria; Fig. 3a). Of the rest, ~29% synthesized at least two B vitamins (e.g. some Alphaproteobacteria, Bacteroidota and the rest of Gammaproteobacteria which could produce vitamin $B_1$ and $B_7$) and ~23% could produce only one type of B vitamin (or ~9% none at all). This suggests that there is a major market for B vitamins, and indeed almost all genomes (~83%) encoded transporters for at least one of these vitamins.

A more detailed analysis of the genomes suggested that marine bacteria could be divided into three main groups based on their predicted strategy for B vitamins acquisition: (1) "Consumers", which lacked the biosynthetic genes but harboured the vitamin transporters (we assumed transporters were for uptake; see Supplementary Note 6 and Supplementary Data 5 for details on transporters' directionality); (2) "Independents", which encoded the biosynthetic pathways but not the relevant transporters; (3) "Flexibles", which encoded both the biosynthetic pathways and transporters for a specific vitamin. Bacteria possessing the latter strategy can potentially switch from being consumers to independent or vice-versa, according to what is more efficient given the surrounding conditions (e.g. availability of extracellular vitamins). The proportion of these three groups changed with the B vitamin studied and the taxonomy of the genomes (Fig. 3b). Very few genomes were flexibles for all three vitamins (~2%), and these were mostly Actinobacteriota (Fig. 3c). There were almost equal proportions of flexibles and independents for vitamin $B_1$

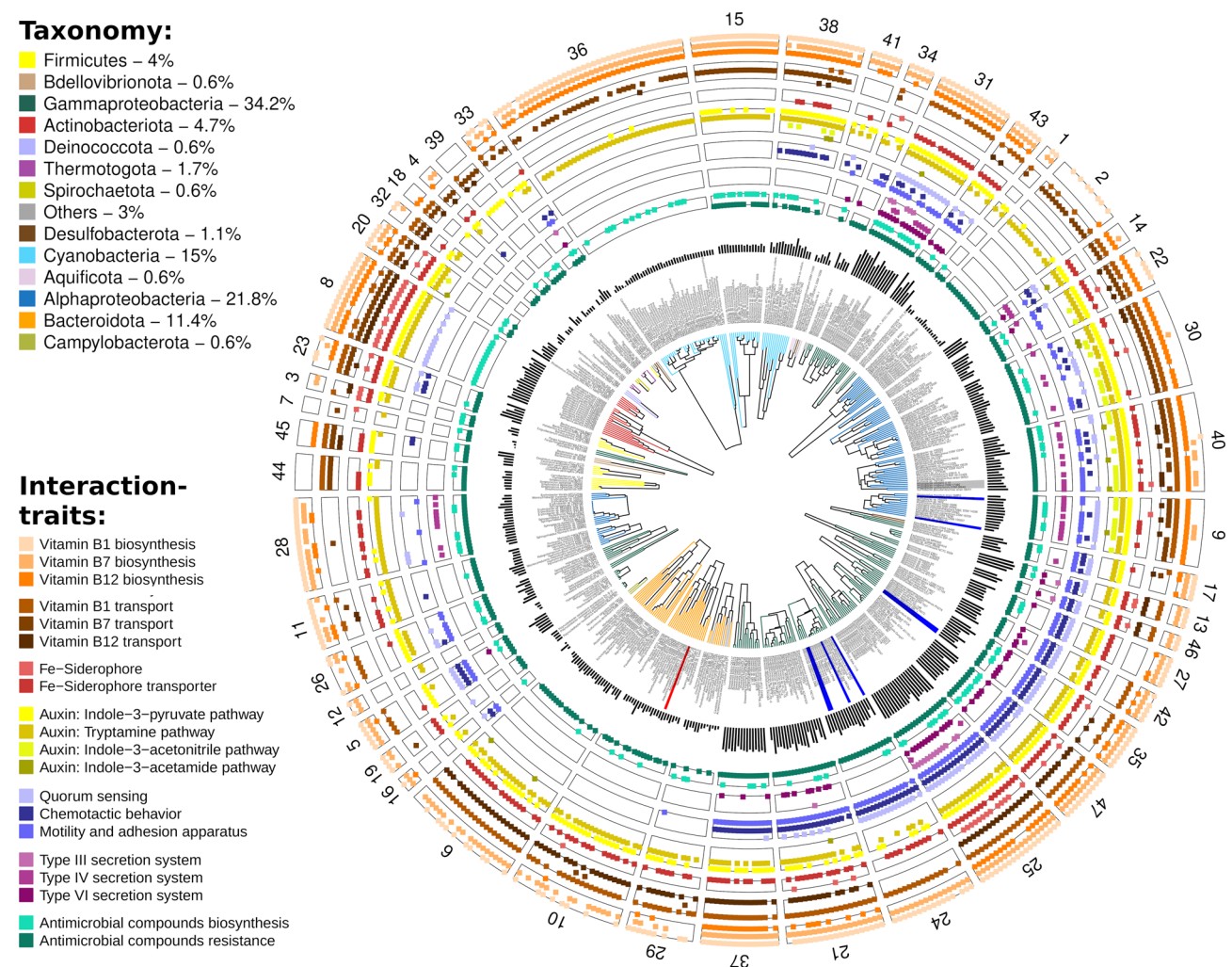

**Fig. 2 Distribution of interaction traits known to mediate cell-cell interactions in bacterial model systems.** Each slice shows the interaction traits present in a genome functional cluster (GFC) and, as a dendrogram, the functional similarity of genetic traits between the grouped genomes (hierarchical clustering of the *r*-correlation matrix with complete agglomeration algorithm). The dark bars show the number of interaction traits annotated in each genome. Genomes belonging to model bacteria, used in literature to discover some of these traits, are highlighted in blue if the interaction was positive (e.g. enhancing phytoplankton growth), in red if it was negative (e.g. kill the host) or in grey if the interaction shifted from positive to negative.

(Gammaproteobacteria and Cyanobacteria, respectively), whereas the most common bacterial strategies for vitamin $B_7$ and $B_{12}$ were independent (Fig. 3b). Many different combinations of synthesis and uptake of the three vitamins were represented in the genomes (51 out of 64 possible combinations, Fig. 3c and Supplementary Fig. 8a), several being enriched in specific GFCs (Supplementary Fig. 9 and Supplementary Note 7). Notably, most strategies required the exogenous uptake of at least one vitamin. While the perceived lack of biosynthetic capacity could be due to the utilization of precursors or to gaps in the pathway annotations (Supplementary Fig. 10 and Supplementary Note 8), we speculate that this could be a potential manifestation of the Black Queen hypothesis, which stipulates that bacteria outsource critical functions to the surrounding community, enabling a reduction of their metabolic cost[75]. In our dataset, the highest fraction of B-vitamin consumers, and hence putative auxotrophs, was observed for vitamin $B_1$, followed by $B_{12}$ and $B_7$. This order, however, does not reflect the metabolic costs of producing such vitamins, as $B_{12}$ would be the most expensive with about 20 genes involved[76], whereas only four genes are required to synthesize $B_7$[70] and five genes for $B_1$[77,78]. Therefore, we hypothesized that vitamin $B_1$ supplies might be more stable or frequent (e.g. as a result of higher export or lysis of producing bacteria) than that of

vitamin $B_{12}$. Nevertheless, very few organisms were predicted to be auxotrophic for all three vitamins, suggesting that completely relying on exogenous sources for vitamins represents a risky strategy in marine pelagic environments. Taken together, these data provide a comprehensive overview of the potential market for B vitamins in marine environments by defining specific roles (e.g. consumer, independent, flexible/source) and identifying which bacteria (taxon and GFC) fulfil each role.

**Production of siderophores and phytohormones—key mechanisms of synergistic microbial interactions.** The production and exchange of common goods such as siderophores[79], as well as of specific phytohormones like auxin[80], represent traits that may mediate synergistic microbial interactions (e.g. refs. [14,81]). As shown in Fig. 2, approximately 10% of the genomes have the capacity to produce siderophores (mainly Actinobacteriota and Gammaproteobacteria), while almost half of the genomes, from multiple taxa, encoded siderophore transporters (45% of the genomes). Occurrence of siderophore biosynthetic traits was partially consistent with GFC clustering (e.g. nearly all genomes in GFCs 8 and 25 possessed those traits) and partially scattered across single genomes in different GFCs. In contrast, the

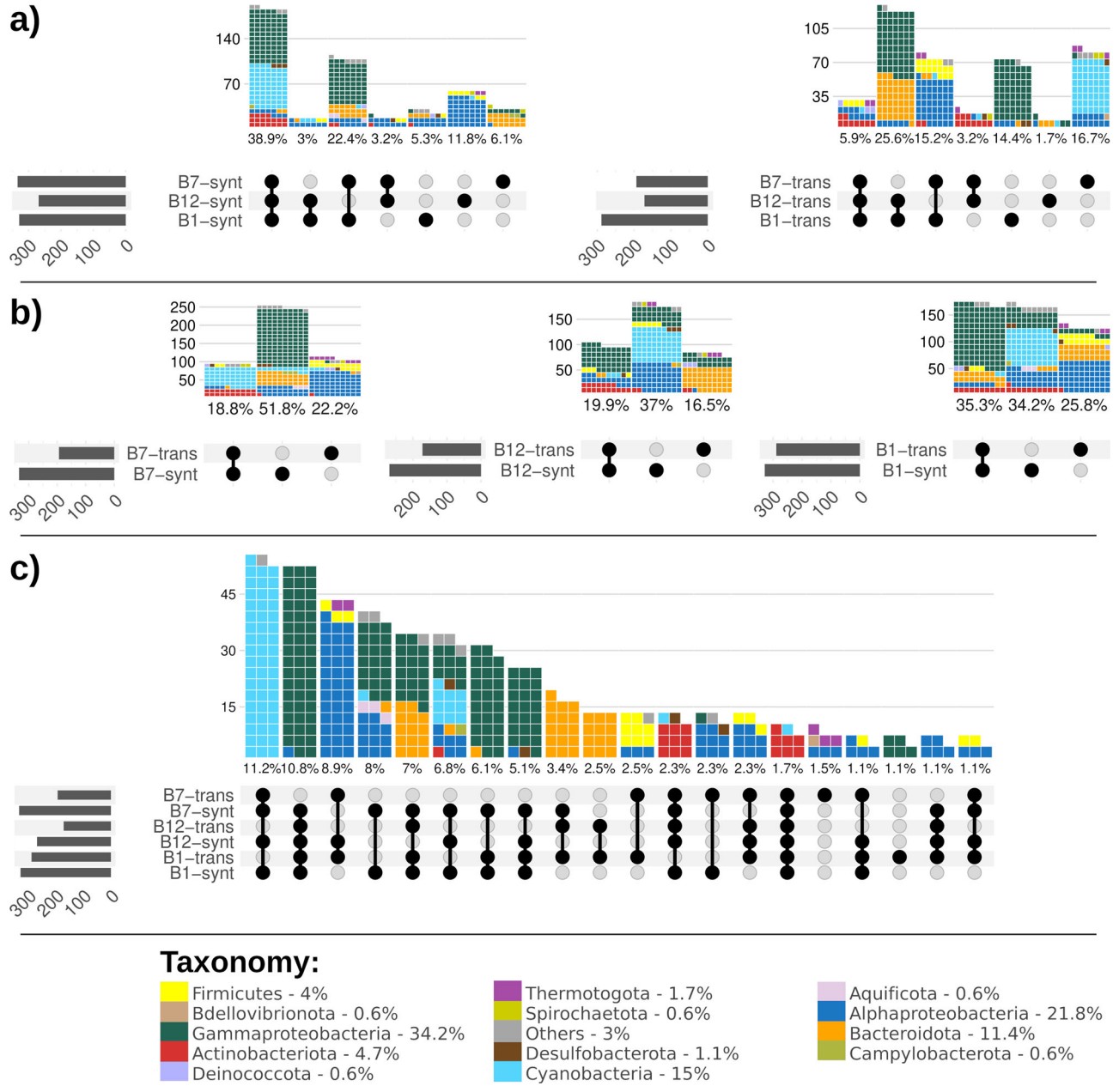

**Fig. 3 Different genomic configurations of traits responsible for the biosynthesis and transport of vitamins B₁, B₁₂, and B₇. a** Capabilities to produce or transport these vitamins, **b** different configurations to either produce and/or transport each of these vitamins and **c** the most abundant configurations to produce and/or transport these vitamins across genomes; the remaining combinations are shown in Supplementary Fig. 8a. Overall, the horizontal bar chart indicates the total number of genomes for each trait, the dark connected dots indicate the different configurations of traits and the waffle bar chart indicates the number (and percentage) of genomes provided with such a configuration; each piece of a waffle bar represents a genome and it is coloured according to the taxon.

distribution of the transporters mainly followed the GFC grouping (Fig. 2). Furthermore, microorganisms can utilize siderophore-bound iron also without the need for siderophore transporters, e.g. using ferric reductases located on the plasma membrane[82] or via direct endocytosis[83]. In this regard, 5% of the genomes encoded the capacity to produce vibrioferrin (Supplementary Fig. 11), which is available to a wide range of organisms upon photolysis[22]. Field studies revealed that siderophore biosynthesis is widespread in the ocean[84], and that bacteria producing e.g. vibrioferrin can represent a relevant percentage of the total bacterial communities[85]. Thus, siderophores can be considered as keystone molecules (*sensu*[86]), produced by a limited subset of organisms but utilizable by a wide range of bacteria[87–89].

Several recent studies have shown that bacteria can influence the growth of phytoplankton through the production of phytohormones[14,19,25], and indeed the auxin hormone indole-3-acetic acid (IAA) has been identified in natural marine samples[14]. Nearly all genomes (~92%) in our dataset are predicted to produce IAA. Four pathways for the production of IAA were identified, with some organisms encoding more than one pathway. The tryptamine pathway was the most common one and was present in nearly all GFCs comprising genomes of Alphaproteobacteria, Gammaproteobacteria, Cyanobacteria and Actinobacteriota (Fig. 2). The indole-3-pyruvate pathway was the second most common with an almost identical distribution to the tryptamine pathway (missing in GFCs 36 and 15, pico-

Cyanobacteria, and 28, Alphaproteobacteria), whereas the last two pathways were rarer (<10% of genomes) and limited to Alphaproteobacteria (indole-3-acetonitrile) and some genomes of Cyanobacteria and Actinobacteriota (indole-3-acetamide). It is tempting to speculate that the widespread distribution of the capacity to produce IAA, and the diversity of biosynthetic pathways, suggest that many heterotrophic bacteria can directly increase phytoplankton growth through specific signalling (e.g. refs. [14,19]). However, all pathways for IAA production are tightly intertwined with the metabolism of tryptophan, either involved in tryptophan catabolism (to cleave the amino group for nitrogen metabolism) or as a release valve to avoid the accumulation of toxic intermediates (e.g. α-keto acid indolepyruvate and indoleacetaldehyde). Additionally, IAA can be catabolized as a carbon source for growth (see ref. [90] and references therein). Given the wide distribution of the tryptamine and indole-3-pyruvate pathways (70–78% of genomes), we hypothesize that they might be more linked to the metabolism/catabolism of tryptophan, whereas the indole-3-acetonitrile and indole-3-acetamide pathways (3–8% of genomes) could be responsible for the production of IAA involved in phytoplankton–bacteria interactions. This hypothesis is supported by the presence of the latter traits in GFCs 9 and 40 that group model organisms known to interact through auxin with phytoplankton (Fig. 2)[14,19,25], and by the fact that genes specifically related to these two pathways were found to be upregulated in one of those studies[14].

**Traits underlying potential antagonistic interactions**. Experimental measurements of interactions among marine bacteria suggest that antagonism is common (>50% of the tested isolates)[33,35], but in most cases the mechanisms behind such antagonistic interactions are unclear. Antimicrobial compounds may underlie many antagonistic interactions in marine environments (e.g. refs. [91–93]), and indeed genes encoding for the production of such compounds were found in several bacteria in our dataset (Supplementary Tables 7 and 8). Interestingly, antimicrobial compounds were predicted to be produced in ~30% of genomes (inner ring in Fig. 2), including also GFCs poor in other interaction traits. The most abundant traits across GFCs were bacteriocin and beta-lactone production[94,95] (Supplementary Fig. 12). Traits involved in the resistance to antimicrobial compounds were also relatively common (78% of genomes; Fig. 2), however, along with specific traits (e.g. specific efflux pumps for antibiotics), we noticed that many KEGG modules annotated as resistance traits were also involved in other cellular functions (e.g. cell division, protein quality control and transport of other compounds; Supplementary Table 7)[96,97]. All GFCs which grouped genomes of Cyanobacteria, Actinobacteriota, and Bacteroidota possessed only these non-specific resistance traits (Supplementary Fig. 12), suggesting that such clades are less efficient in resisting microbial chemical warfare. In support of this hypothesis, some Cyanobacteria strains are indeed used as markers for antibiotic contamination because of their sensitivity (e.g. refs. [98,99]), and Bacteroidota are often inhibited when co-cultured with other bacteria that express antagonistic behaviour[33,35]. Overall, these genome-based predictions are in agreement with previous experimental results[33,35], which suggested that Alpha- and Gammaproteobacteria commonly inhibited other bacteria, whereas Bacteroidota had a low inhibitory capacity and were the most sensitive to inhibition by other bacteria.

Antimicrobial compounds or toxins often need to be delivered into the target organism, e.g. using type IV or type VI secretion systems (T4SS and T6SS, respectively). Approximately 24% of the strains encoded T4SS or T6SS, and these were found primarily in GFCs containing Alpha- and Gammaproteobacteria (Fig. 2). The two

secretion systems had different distributions among the GFCs, with only GFC 25 and 31 (comprising *Vibrio* and *Burkholderia*, genera of Gammaproteobacteria) bearing both systems. The T4SS system can perform multiple roles, including conjugation, DNA exchange and toxin delivery in bacteria-bacteria or bacteria-eukaryote interactions[100]. T4SSs were detected more frequently in Alphaproteobacteria (5 out of 8 GFCs). To date, T6SSs are known to be involved only in antagonistic interactions, including among marine bacteria[101], suggesting that the presence of this trait is a high-confidence predictor of the ability to directly inhibit other cells ([102] and references therein). In our dataset, T6SSs occurred almost exclusively in GFCs comprising Gammaproteobacteria, specifically in *Marinobacter* and *Vibrio*, suggesting a strong capacity for contact-mediated antagonistic interactions in these taxa. Type III secretion systems (T3SS), which deliver effector molecules that maintain the bacterial association with the host[103], were found only in a few genomes as the *Vibrio* clustered in GFC 25. This GFC grouped known zooplanktonic hosts[61], suggesting a more specific role for T3SS in metazoan host-microbe interactions.

**Linked trait clusters (LTCs) delineate functional connectivity between individual interaction traits**. While individual traits may be important in determining the outcome of microbial interactions, such interactions are often highly complex and require multiple traits such as motility, signalling and metabolic interactions to operate together (e.g. refs. [14,19,21,24,25]). If these interaction mechanisms are evolutionarily conserved, traits that are functioning together to mediate such interactions should co-evolve, meaning that selection would favour maintaining all relevant traits in the same genome[104,105]. To identify cases of co-evolving traits, we used linkage disequilibrium analysis and clustered traits which were found together more often than expected by chance (adjusted $p$-value < 0.05) into Linked Trait Clusters (LTCs; Fig. 1, Supplementary Data 8 and Supplementary Fig. 13). For example, LTC 10 includes pathways for assimilatory sulfate reduction, siroheme and heme biosynthesis, as well as vitamins $B_1$ and $B_7$ biosynthesis. These traits appeared together more often than random pairs of traits (mean r within this LTC is 0.38, compared to 0.09 among all trait pairs; Supplementary Fig. 13b) and they are also functionally linked. In fact, siroheme is a prosthetic group for assimilatory sulfite reductases[106,107] and, in sulfate-reducing bacteria, siroheme can be hijacked for the biosynthesis of heme[108]. Finally, once reduced, sulphur can be incorporated into essential molecules such as amino acids (methionine and cysteine) and membrane lipids, as well as into vitamins $B_1$ and $B_7$[109].

Similar to pangenome analyses, we divided all LTCs into core (present in >90% of genomes), common (<90% and ≥30%) and ancillary (≤30%; Supplementary Fig. 13c). Note that, while pangenome analysis is based on single gene distributions, each LTC included different genetic traits and each trait often involved >3 genes. Two core LTCs, 3 and 5 (mean $r$ of 0.30 and 0.35), occurred in nearly all genomes (>93%; Fig. 1) and, as expected, they linked traits that mediate for core metabolic functions, common to almost any cell. These include biosynthesis of nucleotide (DNA and RNA) and amino acids, as well as core metabolic pathways (glycolysis, pentose phosphate pathway and the first three reactions of the TCA cycle). In contrast, other common LTCs (i.e. 2, 4 and 7) highlighted cases in which major metabolic pathways such as the TCA cycle or pathways for the cell wall assembly were missing in specific bacterial clades. These absence patterns were consistent with previous studies (Supplementary Note 9 for detailed description). The LTC concept can therefore be used to identify traits that may function together, providing hypotheses of unknown modes of interaction that can be tested experimentally.

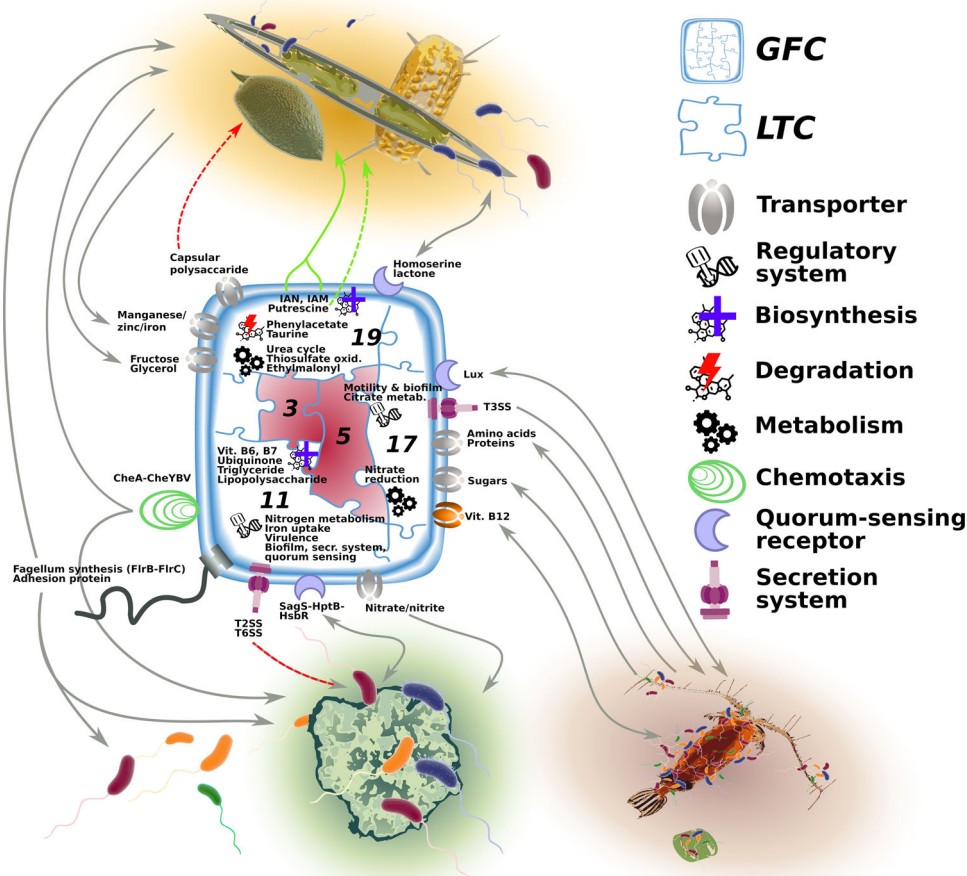

**Fig. 4 Conceptual representation of predicted interaction modes for a hypothetical bacterial genome analysed with our trait-based approach.** The bacterium is assigned to a GFC, visualized here as a jigsaw puzzle, and every puzzle piece represents one of the linked trait clusters (LTCs) possessed in that GFC. LTCs 3 and 5 (marked in red) are the core LTCs that are present in any GFC. LTC 19 holds traits mediating potential interactions with phytoplankton cells, while LTC 11 confers the capability to interact with other bacteria on organic particles and LTC 17 enables interactions with other eukaryotic hosts such as zooplankton. Green arrows indicate traits with positive effects (e.g. enhancing growth), grey arrows traits mediating metabolites/ chemical exchange, movement or attachment, and red arrows traits with negative effects (e.g. pathogenicity). A solid arrow is used when the mediated mechanism has already been described in the literature, while a dashed arrow indicates a yet uncharacterized mechanism.

Analysis of auxiliary LTCs which include interaction traits such as secretion systems suggested that these are often linked to traits encoding for chemotaxis, motility and adhesion. We posit that these traits represent a typical set a bacterium would need to locate, reach and settle on an organic matter particle or eukaryotic hosts (phytoplankton, zooplankton, fish). Moreover, other genetic traits (e.g. metabolic or regulatory) were linked within these LTCs and they may also be involved in microbial interactions (Fig. 4). For example, LTC 11 (mean $r = 0.40$) included, in addition to T6SS, traits for adhesion, a flagellar regulatory system, quorum sensing (controlling for swarming and biofilm formation), chemotaxis and a nitrogen transporter with the regulation system. The same LTC also encoded for the biosynthesis of ubiquinone, vitamins $B_6$ (pyridoxal) and $B_7$, and for two regulatory systems (*BarA-UvrY*, *RstB-RstA*), which are known to modulate virulence, cellular metabolism, biofilm formation, stress resistance, quorum sensing and secretion systems[110,111]. This LTC was common in the GFCs grouping Gammaproteobacteria such as *Pseudoalteromonas* (GFC 21) *Alteromonas* (GFC 24), *Marinobacter* (GFC 35), *Shewanella* (GFC 37) and *Vibrio* (GFCs 25 and 47; Supplementary Data 3 and 8). All these organisms are known as particle and phytoplankton associated bacteria (e.g. refs. [46–49,62]), and in such micro environments they can potentially engage in microbial

interactions using these linked traits (i.e. biosynthesis of B vitamins, quorum sensing and T6SS[101,112]). Interestingly, the other two secretion systems, T4SS and T3SS, were also linked with regulation systems for nitrogen metabolism and with vitamin $B_7$ or $B_{12}$ transporters as part of LTC 25 and LTC 17, respectively. We propose that the linkage between these traits across different LTCs suggests that these processes occur together in multiple interaction modes. In principle, there could be a direct link in which the injection of an effector molecule modifies the response, for example, to nitrogen starvation (as shown for phosphate starvation in response to the toxin cylindrospermopsin[113]). However, the linkage between these traits may also be the result of complex interactions that require the coordinated exchange of multiple metabolites and signals (e.g. ref. [14]).

Notably, LTC 17 (which encodes the T3SS; mean $r = 0.47$) included also amino acid and sugar transporters, and two regulation systems (*UhpB-UhpA*, *CitA-CitB*). The *UhpB-UhpA* genes control the motility and colonization of fish pathogens[114] and, not surprisingly, the LTC was found in GFCs 46 and 47 which included known pathogenic bacteria (grouping *Aeromonas* and *V. natriegens* genomes; Supplementary Data 3 and 8)[115,116]. LTC 17 was also found in GFCs that grouped non-pathogenic but still host-associated taxa (GFC 25, which includes *V. alginolyticus*, and GFC 13 grouping Enterobacteriaceae)[117], supporting the role

of this LTC (as well as of the T3SS it encodes) in interactions with a broad range of eukaryotic hosts including zooplankton, phytoplankton and fish (Fig. 4). Moreover, *CitA–CitB* regulation system controls the citrate metabolism in response to changes in amino acid concentration or pH[118]. We speculate that the link between microbial interaction traits, citrate metabolism and protein and amino acid transporters within LTC 17 may be relevant for a host-associated bacteria during the dispersal stage (lower amino acid concentration and change in pH).

Finally, the analysis of LTC 19 (mean $r = 0.34$) lent further support for the hypothesis that some IAA production pathways are involved in phytoplankton–bacteria interactions and not just in the tryptophan catabolism (see above). This LTC included the indole-3-acetonitrile and indole-3-acetamide pathways, along with other key microbial–phytoplankton interaction traits such as quorum sensing[25,119] and taurine degradation[14,17]. Other linked genetic traits hint to additional molecular mechanisms: the manganese/iron transporter suggests a micronutrient-dependent response, the transport of capsular polysaccharide may be involved in resistance to host defence and pathogenicity[120], and the biosynthesis of putrescine can stimulate phytoplankton growth, productivity and stress tolerance, as shown previously in plants[121]. Moreover, the LTC was found complete in GFCs 9, 30 and 40 (Supplementary Data 8) which grouped genomes of bacteria known to interact via IAA with phytoplankton (e.g. *Dinoroseobacter*, *Sulfitobacter* and *Phaeobacter*)[14,19,25,122]. We propose that possessing this LTC indicates that the relevant bacterium is capable of affecting the growth of phytoplankton through a combination of specific hormone signalling pathways and metabolic interactions.

**Conclusions.** We present a framework that extrapolates from studies of specific model organisms to predict the interaction potential of other bacteria based on the traits encoded in their genomes (Fig. 4). By focusing on biologically relevant traits (including specific interaction traits), we reduced a highly complex genomic dataset to a tractable matrix of organisms by functions. By organizing the ensuing genomic information into GFCs, we further simplified the interpretation of complex genomic datasets, while at the same time highlighting the non-trivial grouping of organisms by phenotypic traits, sometimes irrespective of taxonomic boundaries. The LTCs provided evidence for the functional and evolutionary linkage between traits, raising hypotheses as to how these traits act together in the context of complex processes such as microbial interactions. This approach can be easily scaled to different systems such as freshwater, terrestrial or other host microbiomes (e.g. zooplankton and fish; see LTC 17 in Fig. 4), and expanded to include information from additional data sources (e.g. metabolomics or high-throughput functional assays). It also facilitates the investigation of the functional and interaction potential of metagenomes (e.g. to identify communities where interactions might be more relevant than others) and high-quality metagenome-assembled genomes in field studies.

Applying this approach to a wide diversity of bacterial taxa, we showed that marine bacteria encode different configurations of interaction traits. Known particle-associated taxa of Alpha- and Gammaproteobacteria possessed the full set of traits to interact with particles and living hosts, while Bacteroidota, a known ubiquitous copiotroph taxon, did not have this capacity. Actinobacteriota and Cyanobacteria represented potential sources for $B_7$ in the B vitamin market, while most Alphaproteobacteria appeared as obligate customers. We suggest that siderophores, and vibrioferrin in particular, are keystone molecules being produced by only a few bacteria (Actinobacteriota and Gammaproteobacteria) but affecting a much larger diversity of potential users, in agreement with the Black Queen Hypothesis[75]. Finally, the production of IAA might be more common than expected, and in some cases (e.g. GFCs 9, 30 and 40 encoding LTC 19) this may be linked with other traits involved in phytoplankton–bacteria interactions (Fig. 4).

The GFC and LTC concepts are both statistical in nature, representing the probability of bacteria having similar functional capacity, and of traits being functionally and/or evolutionarily linked. In support of the GFC concept, a study of marine Vibrionaceae suggested that ecologically cohesive unit (similar to GFCs) are likely to interact using similar trait combinations[30]. However, some interaction phenotypes, such as the ability to inhibit multiple target bacteria (super-killers), are not phylogenetically conserved[30,123]. Experimental studies using both established and new model systems across multiple scales of diversity (e.g. between and within GFCs) are needed to test the GFC framework, whereas genetic manipulation of linked traits to test their effect on microbial interactions will be required to determine to what extent traits within LTCs are functionally linked. Nevertheless, GFCs and LTCs describe how putative interaction traits vary across different bacterial taxa, and thus can be used to quantify how the fraction of natural communities potentially ready to interact changes over space and time. In turn, this should help to elucidate fundamental rules that govern community dynamics and assembly in the oceans, and the roles played by microbial interactions in global ecosystem-level processes and biogeochemistry.

## Methods

**Genome selection.** A dataset of complete and high-quality draft genomes of marine bacteria was compiled performing extensive research on metadata available from NCBI (http://www.ncbi.nlm.nih.gov/genome), JGI (https://img.jgi.doe.gov/cgi-bin/m/main.cgi?section=FindGenomes&page=genomeSearch), and MegX (https://mb3is.megx.net/browse/genomes) websites. Although the focus of the analysis was on bacteria inhabiting the marine pelagic environment, some genomes from organisms isolated in extreme marine environments (i.e. thermal vents, saline and hypersaline environments, estuaries) and sediment, as well as from human and plant symbionts (*Sinorhizobium* and *Mesorhizobium*) were kept for comparison. The final list of 473 genomes includes all of the genomes that, using CheckM 1.0.11[124], were defined closed (i.e. each DNA molecule, such as chromosome and plasmids, was represented as a single sequence in the fasta file) or high-quality draft genomes (>90% of completeness, <10% of contaminations, >18 tRNA genes and all three rDNAs present)[125]. The final dataset included 473 complete genomes with 117 closed genomes and with >81% genomes that were >99% complete. Of these 473 genomes, 421 were isolated in marine pelagic and coastal zones, 34 in extreme environments (e.g. salt marsh or hydrothermal vent), 6 in marine sediment and, of the remaining, 8 were human associated and 4 plant roots associated (Supplementary Data 3).

**Genome annotation.** Genome taxonomic classification was obtained using the protein phylogeny workflow implemented in GTDB-tk 1.4.0[52] with the command *classify_wf* (standard settings), and all retrieved genomes were functionally re-annotated using a standardized pipeline. In brief, gene calling and first raw annotation steps were performed with Prokka 1.14.5 (standard settings and --rnammer for rRNA prediction)[126]. The amino acid sequences translated from the identified coding DNA sequences of each genome were annotated against hidden Markov model profiles of KEGG Orthologs (KEGG database 94.0) using KofamScan 1.2.0 (standard settings) and only matches with scores above pre-computed KO-specific thresholds were retained[127]. Additional targeted analyses were performed to annotate secondary metabolites, phytohormones, specific transporters and utilization of sulfur metabolites. The genbank files generated by Prokka were submitted to a local version of Anti-SMASH 5.1.2 (--clusterblast --subclusterblast --knownclusterblast --smcogs --inclusive --borderpredict --full-hmmer --asf --tta), which generated a list of predicted secondary metabolite biosynthesis gene clusters[128]. Pathways for the biosynthesis of the phytohormone indole-3-acetic acid (IAA) were manually identified by annotating (blastp 2.10.0+ best hit, e-value < 1e-5 and bit score > 60)[129] the translated amino acid sequences against the KEGG orthologies required to generate IAA from tryptophan in the KEGG map01070 (Supplementary Data 9). Translated amino acid sequences were also used as input for a GBlast search (BioV suite 1.0; default settings)[130] to identify transmembrane proteins mediating the transport of B vitamins and siderophores (Supplementary Data 5). For downstream analysis, only annotations with a

transmembrane alpha-helical overlap score > 1 and a blast e-value < 1e-6 were retained. Production and transport of photoactive siderophores (i.e. vibrioferrin)[22] were identified with functional annotations from the most similar (BLAST best hit) sequences. The predicted protein sequences were blasted (e-value < 1e-5 and amino acid similarity >30%)[131] against a reference dataset assembled using all available sequences of related genes (*pvsABCDE* and *pvuBCDE* operons)[132] available in UniProt (Supplementary Data 10). The same approach was carried out to annotate the dimethylsulfoniopropionate (DMSP) degradation pathways of demetilation and cleavage blasting (e-value < 1e-70) against a reference dataset that contained all Uniprot sequences of the genes listed in[133,134] (Supplementary Data 10). Catabolic pathways of other two sulfur metabolites, 2,3-dihydroxypropane-1-sulfonate (DHPS) and taurine, were identified checking for the presence of key reactions in the KEGG Orthologs annotations: all known routes to degrade DHPS share a sulfopropanediol 3-dehydrogenase (*hpsN* gene; K15509)[15], while taurine can enter the TCA cycle either via taurine-pyruvate aminotransferase (*tpa* gene; K03851) or via taurine dehydrogenase (*tauXY* genes; K07255 + K07256), followed by a sulfoacetaldehyde acetyltransferase (*xsc* gene; K03852)[135].

**KEGG module reconstruction.** KEGG Orthologs (KOs) annotations generated by KofamScan were recombined in KEGG modules (KMs) using an in-house R script. The KMs represent minimal functional units describing either pathways, structural complexes (e.g. transmembrane pump or ribosome), functional sets (essential gene sets as Aminoacyl-tRNA synthases or nucleotide sugar biosynthesis) or signalling modules (phenotypic markers as pathogenicity). Briefly, using the R implementation of KEGG REST API[136], the script fetches the diagrams of all KMs from the KEGG website. Each diagram represents a reactions' scheme of a KM listing all known KOs that can perform each of the reactions necessary to complete that scheme (Supplementary Fig. 1b). The completeness of a KM in a genome was assessed as the number of required reactions for which at least one KOs was annotated and only complete KMs were retained in downstream analyses (e.g. a KM with 7 out of 8 reactions is incomplete and would be discarded). However, to partially compensate for possible annotation issues, one missing reaction was allowed in KMs with ≥3 reactions (i.e., a KM with 7 out of 8 annotated reactions is considered complete; see Supplementary Note 10 and Supplementary Data 11 for more details).

**Genetic and interaction traits identification.** The annotated complete KMs, secondary metabolites, phytohormones and transporters represent the genetic traits identified in the genomes. From this list, the subset of interaction traits was manually extracted based on current knowledge about processes that likely play a role in microbial interactions (list of picked interaction traits in Supplementary Data 2). Within the KMs we identified traits related to vitamin biosynthetic pathways, quorum sensing, chemotaxis, antimicrobial resistance, motility and adhesion (Supplementary Data 6). Since the ecological role of most secondary metabolites is still unclear, a careful literature search was performed to identify and retain only the secondary metabolite clusters with a proposed function that can be linked to microbial interaction processes, such as siderophore production, quorum sensing and antimicrobial compound biosynthesis (Supplementary Data 7). The phytohormone annotations revealed the capability of producing indoleacetic acid (auxin) through four different pathways (Supplementary Data 9). Vitamin and siderophore transporters were identified in the transporter annotations looking for the related transporter families (e.g. TonB, Btu) and the substrate information (Supplementary Data 5).

**Mapping to environmental datasets.** The selected datasets represent amplicon time series generated with Illumina sequencing of the V4 region of the 16S rDNA of bacterial communities sampled in a coastal (between 1–5 m of depth, Canoe Cove, Nahant, MA, USA)[64] and a pelagic (between 10–500 m of depth, n-1200 station, Easter Mediterranean sea)[65] site. Mapping between the full 16S rDNAs extracted from genomes and amplicon sequences was performed with BLAST (blastn, e-value < 1e-5) using different identity thresholds to filter the best hits: 100%, 97%, 94.5% and 86.5%. The first two values represent a new proposed threshold and the most commonly used threshold to define operational taxonomic units (OTUs)[137], while the last two values were suggested as thresholds to classify OTUs at genus and family levels[138]. In addition, for each identity threshold, we inspected the top blast hits (up to 20) and calculate a mapping specificity index as the number of hits assigned to the same GFC over the total number of hits (Supplementary Figs. 4a and 5a). To avoid spurious mapping, we only retained in downstream analyses sequences with a specificity index = 1 (i.e. all best blast hits belonged to the same GFC).

The evaluation of the GFC concept was performed by using the coastal time series, because it offers a high (daily) temporal resolution of changes in the bacterial community composition and the authors applied a time deconvolution analysis to characterize the OTU temporal dynamics[64]. The OTUs included in the deconvolution analysis represented ~97% of the total sequences in the dataset and for each pair of OTUs, the authors calculated the frequency interaction score (the higher the score, the more synchronous were the temporal dynamics of both OTUs, and vice-versa). From the list of all pairs, we only considered OTU pairs for which at least one of the two OTUs mapped to a GFC. We wanted to exclude cases of OTUs pairs that could belong to the same, but yet unknown, GFC. To test for the GFC concept, we compared the frequency interaction score between OTU pairs mapped to the same OTU (i.e. both OTUs mapped to the same GFC) versus OTU pairs mapped to different GFCs (e.g. the two OTUs mapped to different GFCs, or one of the OTU was unmapped). For each identity threshold, the normal distribution of the frequency interaction score was assessed with the Shapiro test (r's function *shapiro.test*; *p*-values ≪ 0.001), and *t*-test (r's function *t*-test) was performed to test for a significant difference in the mean ranks between the two groups of OTU pairs. We repeated the test by randomly assigning OTU pairs to the same-GFC and different-GFC groups by keeping the same group sizes and by creating two groups of equal size.

**Statistics and reproducibility.** The presence/absence matrix of genetic traits across genomes served as a basis to cluster the former into linked trait clusters (LTCs) and the latter into genome functional clusters (GFCs). Both clustering approaches implemented the Pearson coefficient *r* (also known as phi coefficient when applied to dichotomous variables) to calculate the genome and trait correlation matrices. While no pair of genomes scored a negative r value (as they all shared core functional traits), negative correlations between pairs of genetic traits were thresholded to zeros to ensure that the trait clustering was only driven by positive correlation. Moreover, only pairwise correlations with a FDR-corrected *p*-value < 0.05 (chi-square test, df = 1) were retained. The parsed correlation matrices were fed as similarity matrices into the affinity propagation algorithm implemented with the apcluster function (*q* = 0.5; R package apcluster 1.4.8)[139]. This machine-learning algorithm was chosen because it does not require the number of clusters to be determined a priori, allowing instead this feature to emerge from the data[140]. Briefly, a functional similarity matrix is used to construct a network where nodes and edges are known to be genomes (or genetic traits) and their pairwise *r*-correlation, respectively. Starting from a random set of exemplar nodes, clusters are created by expansion towards the adjoining and most similar nodes. Through iterations of this procedure, the algorithm tries to maximize the total similarity between nodes within each cluster, eventually converging towards the best set of clusters. Clustering robustness and accuracy of both GFCs and LTCs were tested by performing a sensitivity analysis of the '*q*' parameter which controls the clustering sensitivity in the apcluster function, and by down-sampling overrepresented taxa at 80%, 60% and 40% of their genome coverage (Supplementary Note 10 and Supplementary Fig. 15).

For the LTC delineation, genetic traits were pre-filtered to remove noisy signals and only traits found in ≥3% of the genomes (≥14 genomes) were used in the clustering (i.e. 379 genetic traits out of 578 in total). Similar to the context of linkage disequilibrium[141], Pearson coefficient *r* indicates non-random association between genetic traits because those traits are interactively linked to fitness, or simply because they are closely located on the chromosome (i.e. lower chances of recombination). However, as the genetic traits analysed commonly involve multiple genes, the second possibility is less likely. While exploring the functional potential, an LTC was considered present in a genome when >50% of the grouped genetic traits were present and it was considered present in a GFC if it was present in >50% of the grouped genomes. The genetic traits belonging to LTCs that were never found to be complete in at least one GFC were considered as unclustered.

All analyses were performed in R 4.0.4[142]. Heatmaps were plotted using the packages ComplexHeatmap 2.8.0[143] and iheatmapr 0.5.1[144], the circular visualization with the package circlize 0.4.13[145], the intersection plots with the package ComplexUpset 1.3.0[146] and waffle 0.7.0[147]. The remaining plots were generated using ggplot2 3.3.4[148] while the packages dplyr 1.0.7[149] and reshape2 1.4.4[150] were used for data manipulation.

**Caveats of the bioinformatic analysis.** As with any bioinformatic approach, also our workflow aimed to identify functional traits on genome sequences, has inherent limitations. On average, only 63% of genes were functionally annotated across genomes (range ~40–80%), with the remaining genes either annotated as hypothetical or not annotated at all. This is a strong reminder of the limitations of current genomic and metabolic knowledge. Moreover, although the analysed genomes represented a substantial fraction of bacterial taxa in marine environments (Supplementary Figs. 4 and 5), we still lack high-quality genomes for many taxa. Future work, both culture-dependent and independent, is required to obtain an unbiased view of the numerous traits encoded in marine microorganisms.

**Reporting summary.** Further information on research design is available in the Nature Research Reporting Summary linked to this article.

# Data availability
All genomes are available in the online repositories of NCBI and JGI under the accession number listed in Supplementary Data 3. Fasta files with protein sequences for the annotation of phytohormone, vibrioferrin and DMSP pathways, as well as HTML interactive figures are provided at https://figshare.com with the https://doi.org/10.6084/m9.figshare.16942780. Source data underlying Fig. 3 is presented in Supplementary Data 4.

## Code availability

The scripts for functional annotation, reconstruction of KEGG modules and statistical analysis are available on GitHub at https://github.com/lucaz88/genome_comparison_code and are archived at https://zenodo.org with the https://doi.org/10.5281/zenodo.5662367.

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

## Acknowledgements

This work was supported by the Human Frontier Science Program (HFSP) through the grant number RGB 0020/2016. D.Sh and D.Se also acknowledge support by the National Science Foundation through grant NSFOCE-BSF 1635070. We thank Anna Godhe (who unexpectedly and sadly passed away during our studies), Mats Töpel and Oskar Johansson for providing early access to some of the genomes included in the analysis. We thank the entire IAMM team (Dikla Aharonovich, David Bernstein, Falk Eigemann, Elena Forchielli, Tal Luzzatto-Knaan, Shany Ofaim, Melissa Osborne, Solvig Pinnow, Dalit Roth-Rosenberg, Angela Vogts and Maren Voss) for the constructive discussions and ideas that led us to shape this manuscript, as well as Osnat Weissberg and Minoru Kasada for their feedbacks on debugging the code. A sincere thank you to Silvia Munari for her diligent proofreading of this manuscript.

## Author contributions

L.Z., D.Sh. and H-PG designed the research. L.Z. and D.Sh. analysed the data. T.M. contributed with analytic tools. D.Se. provided bioinformatics advice. L.Z. wrote the paper with substantial revisions from D.Sh., D.Se. and H.-P.G. All authors read and approved the final manuscript.

## Funding

## Competing interests

The authors declare no competing interests.
