## [Transparent Peer Review File · Communications Biology]

Reviewers' comments:

Reviewer #1 (Remarks to the Author):

The manuscript by Zoccarato et al examines known traits that bacteria use in microbial interactions across 473 sequenced genomes, mainly focusing on marine bacteria. They find overarching patterns of prevalence of specific traits like antimicrobial strategies and phytohormone production and limited patterns of other traits like vitamin biosynthesis/transport. This manuscript is quite interesting and provides an overview of bacterial genome metabolism in the marine environment, based on sequenced (and thus mostly cultured) genomes. In general, I believe the topic is suitable to publication in *Communications Biology*. I do have some questions, concerns and comments below.

Lines 29-33: First sentence is misleading. The genomes examined in the manuscript cannot be argued to be good representation of microbial diversity. These are mainly from lab cultures and thus are biased. Thus the examination of the genomes here only tells us about the prevalence of these interactions in mostly cultured bacteria, still. Rephrase.

I would've liked to see a tree or a pie chart in the supplementary (or as part of Fig. 1) to show the distribution of the genomes used and how much representations from different phylum level. Hard to glean this information from Fig. 1 because of the clustering. Alternatively, provide the % of each classification next to their names in Fig. 1. What is the clustering method(s) used in the figure? What do you mean relevant LTC are marked? I only see a number next to some. Be more specific.

Lines 130-132: A bit confusing the way it's phrased. Please rephrase.

Lines 155-164: The way the ecotype clustering is phrased in a way contradicts the earlier conclusion in lines 151-154. Rephrase to clarify.

Line 168: free-living

Lines 183-191: I wonder if the authors considered the fact that some bacterial groups/genera are represented by a lot more genomes than others and how that may skew conclusions about the presence of specific traits. I realize there was a good reason to focus on only closed, circular genomes but that also limits the number of representative genomes in each genus, thus complicating the confidence one should have in trait patterns across GFCs.

Lines 222-224: I'm curious if functions that by default must be in all genomes are indeed in common LTCs, such as DNA synthesis, ribosome, etc. And these should be linked I think. This would be a good positive control, much more so than pentose phosphate pathway and parts of the TCA cycle.

Lines 244-246: Why mainly associated with microbial interactions rather than response to the environment?

Fig. 2 is very small. Please expand. Also define abbreviations in the caption. When you mention vitamin transport, does that mean export or import? There's a brief mention of that (lines 283-284) but not much else. How may that explain the variations observed in biosynthesis vs transport across genomes? Have you also considered vitamin B3 (niacin)? Some bacteria have been shown to acquire it from phytoplankton (Cooper et al, 2018 *ISME J*).

Lines 296-300: Confusing as written, rephrase.

Line 301: B-vitamin consumers. Also for all vitamins, make the numbers subscript throughout.

Lines 308-311: Auxotrophy can be resolved with provision of the vitamin or a precursor (see McRose

et al, 2014 ISME J). Therefore, a genome can have only the second half of the biosynthesis of an enzyme as it may rely on the precursor's presence in the environment, which seems to be the case in some of the genomes. Can you make some predictions regarding this using your analysis? This would add tremendously to our understanding of vitamin precursors. I am curious to see if there is a way to differentiate dependency on precursors vs gene loss.

Figure 3: arrows are too large. Is there a way to represent these functions in a different way?

Line 348: tryptamine?

Figure 4: Caption is missing a lot of information. For example: What do dashed arrows represent? What is the purple plus sign mean? I believe Lux receptors are supposed to be intracellular, not membrane bound. Why are there empty LTCs inside the bacterial cell? While I realize the authors are focusing mainly on bacterial interactions with phytoplankton, many of the functions discussed (except for some specific examples) may occur between bacteria and other bacteria (antagonistic for example) or between bacteria and other eukaryotic hosts (fish, zooplankton, etc). This should be reflected in the figure. Also some bacteria probably use chemotaxis, motility and adhesion to also attach to phytoplankton not just to marine snow.

Lines 434-437: Are these phylogenetic groups represented by a lot of genomes or just a couple? It would be helpful to list the number of genomes a specific GFC represents that contain a particular LTC. This should be done wherever relevant in the manuscript.

Supplementary:

Supplementary file, line 41: What happens in the pipeline if the stringency was stricter for gap allowances. In other words, instead of 1 gap allowed in KMs with more than 3 steps, what is the effect on the outcome if you allowed 1 gap in KMs with more than 4 or 5 steps? A benchmark to specify how this cutoff was determined will be useful.

In Table S1, why only these traits? Why not add other ones that have been shown to be important as well. For example, organic sulfur compound uptake and metabolism by bacteria from phytoplankton. Many of the genes are known (e.g., uptake and metabolism of DHPS, taurine and N-acetyltaurine).

Line 66: rephrase sentence.

Suppl. Fig. 5: The GFC numbers at the top should be better labeled. Skewing them is very confusing. You can draw a line between the colored box and the skewed number to make it easier to see. Or Alternatively, increase the size of the heat map, so that each pixel is bigger and thus would accommodate these numbers.

Supplementary Fig. 9: what is l_m ?

Reviewer #2 (Remarks to the Author):

The authors analyzed publicly available genomes of bacteria from marine and other environments, focusing on genetic traits putatively involved in mediating interactions between microbes. The so-called "interaction traits" (Fig. 3) were identified based on bioinformatics tools and databases such as metabolic KEGG modules (Supplementary Table 5), and a careful literature search (Supplementary Table 6). The bacteria were clustered into the "genome functional clusters" (GFCs) based on presence/absence of the genetic traits (the column dendrogram in Fig. 1); some of the GFCs correspond to previously defined ecotypes. The genetic traits were clustered into the "linked trait clusters (LTCs)" based on their presence in bacteria (the row dendrogram in Fig. 1); the LTCs present in >90% of genomes were called "core" traits. While the authors stated that "We developed a new

genomic approach", I think the GFCs and LTCs are essentially same as clusters identified by gene content analysis (Snel et al., 1999) and phylogenetic profiling methods (Pellegrini et al., 1999). A major revision, based on the comments given below, is needed to make this manuscript suitable for publication.

#) Interactions

64 Recent studies, using specific model organisms in binary co-cultures, have started to
65 elucidate mechanisms underlying marine microbial interactions (mostly between bac-
66 teria and phytoplankton). Many of these interactions are mediated by the exchange of
67 metabolites used for growth or respiration. For example, bacteria associated with phy-

101 GFC). We propose that organisms belonging to the same GFC are likely to interact in
102 similar ways with other microorganisms. We also identify clusters of traits that are sta-
103 tistically linked, and propose that these linked trait clusters (LTCs) may have evolved
104 to function together, potentially during microbial interactions. GFCs and LTCs provide a
105 framework to extend the knowledge on microbial interactions gained from specific
106 model systems, leading to testable hypotheses as to the prevalence of microbial inter-
107 actions across bacterial diversity.

125 tary Information for a more detailed description of the clustering method). Within each
126 GFC, all genomes are inferred to encode similar traits, and thus these GFCs are ex-
127 pected to be coherent in terms of their functional and metabolic capacity, including
128 the ways in which the related organisms interact with other microbe

244 and physiological knowledge. We therefore interpreted the patterns of LTCs associated
245 with microbial interactions, keeping in mind that these represent bioinformatics predic-
246 tions requiring experimental validation.

- The authors should cite reference papers supporting the assumption/interpretation on interactions. Is there any experimental evidence showing the association between gene contents (functional categories) and interaction capabilities? Organisms whose genomes have the similar gene contents (functional categories) interact in similar ways with other microorganisms? Are microbial interactions/behaviors in microbial communities consisting of >2 microbes same as those in binary co-cultures (one-to-one microbial interactions)?

#) Novelty of approach

- The authors stated that "We developed a new genomic approach" with two keywords "genome functional clusters (GFCs)" and "linked trait clusters (LTCs)". I wonder if the genome functional clusters (GFCs) and linked trait clusters (LTCs) stated by the authors are conceptually different from clusters identified by gene content analysis (Snel et al., 1999) and phylogenetic profiling methods (Pellegrini et al., 1999)?

References:

<https://pubmed.ncbi.nlm.nih.gov/10200254/>

Comparative Study Proc Natl Acad Sci U S A

. 1999 Apr 13;96(8):4285-8. doi: 10.1073/pnas.96.8.4285.

Assigning protein functions by comparative genome analysis: protein phylogenetic profiles

M Pellegrini 1, E M Marcotte, M J Thompson, D Eisenberg, T O Yeates

<https://pubmed.ncbi.nlm.nih.gov/9916801/>

Nat Genet

. 1999 Jan;21(1):108-10. doi: 10.1038/5052.
Genome phylogeny based on gene content
B Snel 1, P Bork, M A Huynen

<https://pubmed.ncbi.nlm.nih.gov/28066816/>
mSystems

. 2016 Dec 27;1(6):e00101-16. doi: 10.1128/mSystems.00101-16. eCollection Nov-Dec 2016.
From Genomes to Phenotypes: Traitair, the Microbial Trait Analyzer
Aaron Weimann 1, Kyra Mooren 2, Jeremy Frank 3, Phillip B Pope 3, Andreas Bremges 4, Alice C
McHardy 1

<https://www.ncbi.nlm.nih.gov/pmc/articles/PMC5192078/>
FIG 5

Single-cell phenotyping with Traitair.

<https://github.com/hzi-bifo/traitair>

<https://pubmed.ncbi.nlm.nih.gov/30285626/>
BMC Evol Biol

. 2018 Oct 3;18(1):148. doi: 10.1186/s12862-018-1261-7.
Environmentally-driven gene content convergence and the Bacillus phylogeny
Ismael L Hernández-González 1, Gabriel Moreno-Hagelsieb 2, Gabriela Olmedo-Álvarez 3

<https://www.ncbi.nlm.nih.gov/pmc/articles/PMC6171248/>
<https://bmcevolbiol.biomedcentral.com/articles/10.1186/s12862-018-1261-7>

Fig. 4

Hierarchical clustering using COGs-based Jaccard's distance (Jd).

Fig. 5

Cluster analysis using Jaccard's distance (Jd) based on Figfams.

<https://pubmed.ncbi.nlm.nih.gov/30137329/>
Genome Biol Evol

. 2018 Sep 1;10(9):2255-2265. doi: 10.1093/gbe/evy178.
Phylogenetic Clustering of Genes Reveals Shared Evolutionary Trajectories and Putative Gene
Functions

Chaoyue Liu 1 2, Benjamin Wright 1, Emma Allen-Vercoe 3, Hong Gu 2, Robert Beiko 1

<https://www.ncbi.nlm.nih.gov/pmc/articles/PMC6130602/>

<https://academic.oup.com/gbe/article/10/9/2255/5076814>

FIG. 5.

—Structure, phylogenetic distribution and functional categories of a hierarchical cluster enriched in
amino-acid biosynthesis proteins.

FIG. 6.

—Structure, phylogenetic distribution and functional categories of a hierarchical cluster enriched in
flagellar motility proteins.

<https://pubmed.ncbi.nlm.nih.gov/27658251/>
PLoS One

. 2016 Sep 22;11(9):e0163098. doi: 10.1371/journal.pone.0163098. eCollection 2016.
Taxonomic Identity Resolution of Highly Phylogenetically Related Strains and Selection of Phylogenetic
Markers by Using Genome-Scale Methods: The Bacillus pumilus Group Case
Martín Espariz 1 2, Federico A Zuljan 1 2, Luis Esteban 3, Christian Magni 1 2

<https://www.ncbi.nlm.nih.gov/pmc/articles/PMC5033322/>

<https://journals.plos.org/plosone/article?id=10.1371/journal.pone.0163098>

Fig 3. Comparison of phylogenomic and functional dendrograms of Bacillus pumilus group strains.

#) "core" traits

120 As shown in Fig. 1, some traits were found across almost all genomes ("core" traits).

218 Based on their occurrence across genomes (Supplementary Fig. 4c), we divided the
219 LTCs into three subgroups: "core" (present in >90% of genomes), "common" (<90%
220 and $\geq 30\%$) and "ancillary" ($\leq 30\%$). LTCs 2 and 3 (mean r^2 of 0.90 and 0.88) were the

- Do these subgroups ("core", "common" and "ancillary") correspond to three parts (core genome;
accessory or dispensable genome; and species-specific or strain-specific genes) in pan-genome?

References:

<https://en.wikipedia.org/wiki/Pan-genome>

<https://sanger-pathogens.github.io/Roary/>

<https://pubmed.ncbi.nlm.nih.gov/31848603/>

Mol Biol Evol

. 2020 Mar 1;37(3):933-939. doi: 10.1093/molbev/msz284.

Estimating Pangenomes with Roary

Farrah Sitto 1, Fabia U Battistuzzi 1 2

<https://www.ncbi.nlm.nih.gov/pmc/articles/PMC7038658/>

<https://academic.oup.com/mbe/article/37/3/933/5652084>

<https://pubmed.ncbi.nlm.nih.gov/27006628/>

Curr Genomics

. 2015 Aug;16(4):245-52. doi: 10.2174/1389202916666150423002311.

Inside the Pan-genome - Methods and Software Overview

Luis Carlos Guimarães 1, Jolanta Florczak-Wyspianska 2, Leandro Benevides de Jesus 3, Marcus
Vinícius Canário Viana 3, Artur Silva 2, Rommel Thiago Jucá Ramos 2, Siomar de Castro Soares 4,
Siomar de Castro Soares 4

<https://www.ncbi.nlm.nih.gov/pmc/articles/PMC4765519/>

Pan-genome consists of three parts: core genome; accessory or dispensable genome; and species-
specific or strain-specific genes.

#) Reproducible research

To make their analysis reproducible, the authors should provide sufficient details about the sequence
data analyses. All the codes/scripts the authors written should be available. All the parameters used in
bioinformatics software (e.g. Prokka) should be available.

#) Robust research

The authors should check the robustness of the results; i.e. whether the conclusions remained similar,
regardless of the methods/parameters and data used.

For example, the conclusions remained similar, regardless of the databases used for functional
annotations (KEGG modules, Pfam, Gene Ontology, SEED subsystem, COG functional category) and
genome sequence data: representative genomes, reference genomes, and Complete Genome
(<https://www.ncbi.nlm.nih.gov/genome/doc/ftpfaq/>).

#) in-house R script

568 KEGG module reconstruction

569 KEGG orthologies (KOs) annotations generated by kofamscan were recombined in

570 KEGG modules (KMs) using an in-house R script ([https://github.com/lucaz88/R_script/](https://github.com/lucaz88/R_script/blob/master/_KM_reconstruction.R)

571 blob/master/_KM_reconstruction.R).

644 sion codes listed in Supplementary table 2. The code for the reconstruction of KEGG

645 modules is deposited on GitHub and freely accessible
646 (https://github.com/lucaz88/R_script).

- Error in R

Running the R script `*my_analysis.R*` printed the following Error messages as shown in the `*log.txt*` file.

```\n

Rscript --vanilla ./my\_analysis.R 2>&1 | tee log.txt

Error in `.getUrl(url, .flatFileParser)` : Forbidden (HTTP 403).

Calls: `KMdiagram_fetcher ... resolve.list -> signalConditionsASAP -> signalConditions`

Execution halted

```\n

#) Line

40 evolved together and point to features of bacteria lifestyles. Specific GFCs,

bacteria lifestyles

->

bacterial lifestyles

#) Line

129 We next sought to understand the extent to which the GFCs correlate with phylogeny.

"Taxonomy" and "Phylogeny" should not be confused.

If I understand correctly, the authors did not use "phylogenetic information" but used "taxonomic information" (i.e. genus, family, order, class or phylum). For examples, species belonging to the genus "Shigella" do not form a clade in a phylogenetic tree. Bacteria of the order SAR11 (Pelagibacterales) are monophyletic?

References:

<https://www.sciencedirect.com/science/article/pii/S0723202014001398>

Diversity and abundance of "Pelagibacterales" (SAR11) in the Baltic Sea salinity gradient - ScienceDirect

If the authors performed the phylogenetic analysis, they should describe the methods in detail.

Here is an example obtained from MEGA X (<https://www.megasoftware.net/>).

Evolutionary relationships of taxa

The evolutionary history was inferred using the Neighbor-Joining method [1]. The optimal tree with the sum of branch length = 0.34882604 is shown. The percentage of replicate trees in which the associated taxa clustered together in the bootstrap test (500 replicates) are shown next to the branches [2]. The tree is drawn to scale, with branch lengths in the same units as those of the evolutionary distances used to infer the phylogenetic tree. The evolutionary distances were computed using the p-distance method [3] and are in the units of the number of amino acid differences per site. This analysis involved 16 amino acid sequences. The coding data was translated assuming a Standard genetic code table. All positions containing gaps and missing data were eliminated (complete deletion option). There were a total of 374 positions in the final dataset. Evolutionary analyses were conducted

in MEGA X [4][5]

1. Saitou N. and Nei M. (1987). The neighbor-joining method: A new method for reconstructing phylogenetic trees. *Molecular Biology and Evolution* 4:406-425.
2. Felsenstein J. (1985). Confidence limits on phylogenies: An approach using the bootstrap. *Evolution* 39:783-791.
3. Nei M. and Kumar S. (2000). *Molecular Evolution and Phylogenetics*. Oxford University Press, New York.
4. Kumar S., Stecher G., Li M., Knyaz C., and Tamura K. (2018). MEGA X: Molecular Evolutionary Genetics Analysis across computing platforms. *Molecular Biology and Evolution* 35:1547-1549.
5. Stecher G., Tamura K., and Kumar S. (2020). Molecular Evolutionary Genetics Analysis (MEGA) for macOS. *Molecular Biology and Evolution* (<https://doi.org/10.1093/molbev/msz312>).

#) Line

151 tary Table 2 and Supplementary information). Overall, the observation that functional-
152 ity does not strictly follow phylogeny makes it difficult to infer the function of a com-
153 munity from its taxonomic structure, as previously shown across the global oceans
154 34,35.

- The genetic traits commonly present in the genome functional clusters (GFCs) may be due to the independent acquisition of functional gene sets (horizontal gene transfer)?

References:

<https://pubmed.ncbi.nlm.nih.gov/19815525/>

Comparative Study Proc Natl Acad Sci U S A

. 2009 Oct 20;106(42):17939-44. doi: 10.1073/pnas.0903585106. Epub 2009 Oct 6.

Comparative genomics reveal the mechanism of the parallel evolution of O157 and non-O157 enterohemorrhagic Escherichia coli

Yoshitoshi Ogura 1, Tadasuke Ooka, Atsushi Iguchi, Hidehiro Toh, Md Asadulghani, Kenshiro Oshima, Toshio Kodama, Hiroyuki Abe, Keisuke Nakayama, Ken Kurokawa, Toru Tobe, Masahira Hattori, Tetsuya Hayashi

<https://www.ncbi.nlm.nih.gov/pmc/articles/PMC2764950/>

<https://www.pnas.org/content/106/42/17939>

The gene contents of the 4 EHECs do not follow the phylogenetic relationships of the strains, and they share virulence genes for Shiga toxins and many other factors.

The independent acquisition of very similar virulence gene sets is predominantly attributable to mobile elements that are commonly present in the EHECs: multiple lambdoid PPs, several types of IEs, and virulence plasmids.

#) Line

194

Fig. 1: Atlas of Marine Microbial Functional Traits showing presence/absence of genetic traits across all analyzed genomes. Each column represents a genome and these are hierarchically clustered. The horizontal color bar represents the taxonomic affiliations of the genomes (mainly phyla, with the exception of Proteobacteria that are represented at the class level) and the horizontal grey bar delineates specific genome functional clusters (GFCs). Rows are the genetic traits clustered using the co-efficient of disequilibrium into linked trait clusters (LTCs) (vertical grey bar); relevant LTCs are marked. Left-side row annotations show the average abundance for the genetic traits and the annotation tool (color coded bar) with which they have been annotated. An interactive version of this figure is available as Supplementary file 1.

195

- In "Fig. 1: Atlas of Marine Microbial Functional Traits", both of the row dendrogram and the column dendrogram were computed by the same clustering algorithm and dissimilarity matrix?
- Does "coefficient of disequilibrium" indicate "square of Pearson's correlation coefficient (r^2)" in Line 619?
- "relevant LTCs are marked" with what? numbers?
- Does "Left-side row annotations show the average abundance for the genetic traits" indicate "Genome fraction" ranging from 0 to 1? relative abundance of the genetic traits in genome, averaged by each GCF?

#) Line

196 Missing common trait clusters highlight basic meta-
197 bolic differences

198 Just as the genomes (represented by the columns in Fig. 1) could be grouped into
199 GFCs, the genetic traits that drive this clustering could themselves be grouped into
200 linked trait clusters (LTCs). The traits within each LTC were found together in the
201 genomes more often than expected by chance, and thus may be linked functionally
202 (Supplementary Fig. 4). For example, LTC 10 includes pathways for assimilatory sulfate

I wonder if the authors performed any statistical test, e.g. `cor.test()` and `fisher.test()` in R, to conclude that "The traits within each LTC were found together in the genomes more often than expected by chance".

#) Line

227 and Tat protein exporters (see Supplementary Table 3 for complete list of all LTCs and
228 their respective genetic traits).

- In the [S_tab3] tab in *6477_0_data_set_158598_qhxxg7y.xlsx* file (Supplementary Table 3), does "trait identified as an interaction-triat" indicate "Known interactions (described in the referenced literature)"?

"interaction-triat"

-> should be

"interaction-trait"

#) Line

259 many Gammaproteobacteria; Fig. 2b). Of the rest, ~33% synthesize at least two B vi-
270 porters for a specific vitamin (Fig. 2a). Bacteria possessing the latter strategy can po-

- Figure should be renumbered because Fig. 2b appear before Fig. 2a.

#) Line

336 many more organisms, including phytoplankton, upon photolysis 18. Thus, in agree-
337 ment with recent considerations 75, we highlight the role of siderophores as "keystone
338 molecules" 76 and take this as an indication that the organisms producing them have
339 an important role in the functionality of microbial communities (e.g., references 77,78).

- Does the "keystone molecules" mentioned here indicate "molecules of keystone significance" reported in the following papers?

<https://pubmed.ncbi.nlm.nih.gov/27651473/>

Proc Natl Acad Sci U S A

. 2016 Sep 20;113(38):10451-2. doi: 10.1073/pnas.1612596113.

Inner Workings: Can single molecules bind together entire ecosystems?

<https://www.pnas.org/content/113/38/10451.long>

In 2013, Zimmer and his then-graduate student Ryan Ferrer borrowed a concept from ecology and dubbed such chemicals "molecules of keystone significance" (3).

<https://academic.oup.com/bioscience/article/63/6/428/225996>

#) Line

361

Fig. 3: Overview on interaction-traits across genome functional clusters (GFCs). Each slice shows the interaction traits present in a GFC and, as dendrogram, the functional similarity of genetic traits of the grouped genomes. Known interactions (described in the referenced literature) between bacteria and phytoplankton are marked with arrows; a blue arrow indicates a positive interaction (an enhancing effect on phyto- plankton growth), a red arrow a negative interaction, and a blue-red arrow a positive interaction that eventually becomes negative. A dashed arrow indicates a known in- teraction involving a bacterial genotype with a high similarity to one of the genomes included in the analysis. Asterisks in siderophores annotation indicate the presence of the specific vibrioferrin synthetic pathway along with the secondary metabolite path- way, while asterisks in the antimicrobial resistance annotation indicate that only generic resistance traits were annotated for that genome.

362

- The authors should clearly define "interaction-traits" or "interaction traits".

Are there any genomic traits which are not directly/indirectly involved in interactions between microbes?

"Known interactions (described in the referenced literature) between bacteria and phytoplankton are marked with arrows" -- What are the remaining interaction traits that are not marked with arrows?

Are these unknown interactions?

#) Line

415 Genome functional clusters differ in their overall po-
416 tential for interactions

417 When considering antagonistic and synergistic interaction traits together, it is clear
418 that some GFCs encode significantly more interaction traits than others (Fig. 3), both
419 in terms of diversity and richness (Supplementary Fig. 9a). One potential driver for the
420 difference in richness and diversity of interaction traits could be the reduction in
421 genome size associated with oligotrophic lineages such as pico-Cyanobacteria and

- I wonder how to calculate the richness and diversity of interaction traits (using the number of different interaction traits and their relative abundances to calculate Shannon entropy)?

#) Line

525 Materials and Methods

526 Genome selection

527 A dataset of complete genomes of marine bacteria was compiled performing an exten-
528 sive research on metadata available from NCBI (<http://www.ncbi.nlm.nih.gov/genome>),
529 JGI (<https://img.jgi.doe.gov/cgi-bin/m/main.cgi?section=FindGenomes&page=genome> -
530 Search) and MegX (<https://mb3is.megx.net/browse/genomes>) websites. Although the

- Line 527

A dataset of complete genomes of marine bacteria

-> should be

A dataset of complete and high-quality draft genomes of marine bacteria

I wonder if the authors could show the assembly status (complete or draft) and the number of replicons (chromosomes and plasmids) in the [S_tab2] tab in

6477_0_data_set_158598_qhxc7y.xlsx file (Supplementary Table 2).

<http://www.ncbi.nlm.nih.gov/genome>
<https://img.jgi.doe.gov/cgi-bin/m/main.cgi?section=FindGenomes&page=genome-Search>
are accessible, while
<https://mb3is.megx.net/browse/genomes>
requires Username and Password to Sign in.

#) Line

563 overlap score >1 and a blast e-value $<1-6$ were retained. Manual annotations were per-
564 formed to specifically identify production of photoactive siderophores (i.e. vibrioferrin)
565 18, blasting predicted protein sequences (blastp) 109 against reference dataset assem-
566 bled using all available sequences of related genes (pvsABCDE operon) 110 available in
567 UniProt.

- Line 563
blast e-value $<1-6$
-> should be
blast e-value $<1e-6$

"Manual annotations" performed here should be explained. Functional annotations from the most similar (BLAST best hit) sequences were assigned and edited manually?

Could the authors provide a list of UniProt accessions for the reference dataset of pvsABCDE operon?

#) Line

603 Statistical analyses

604 The presence/absence matrix of genetic traits in genomes served as basis to cluster
605 the former into linked trait clusters (LTCs) and the latter into genome functional clus-
606 ters (GFCs). The GFCs were generated feeding a genome functional similarity matrix
607 calculated using Phi coefficient (i.e. Pearson correlation for binary variables; phi func-
608 tion, package sjstats) into the affinity propagation algorithm implemented in the ap-
609 cluster function ($q=0.5$ and $\text{lam}=0.5$; r package apcluster) 112. This machine learning
610 algorithm was chosen because it does not require the number of clusters to be deter-
611 mined a priori, allowing instead this feature to emerge from the data 113. Briefly, the

612 functional similarity matrix is used to construct a network where nodes and edges are
613 known to be genomes and their pairwise Phi correlation, respectively. Starting from a
614 random set of exemplar nodes, clusters are created by expansion towards the adjoin-
615 ing, most similar, nodes. Through iterations of this procedure, the algorithm tries to
616 maximize the total similarity between nodes within each cluster eventually converging
617 towards the best set of clusters.

618 For the LTC delineation, a similarity matrix of genetic traits was built calculating the
619 square of Pearson's correlation coefficient (r^2) 114. Out of a total of 556 genetic traits,
620 434 were retained for downstream analysis as they were found in $>3\%$ of the
621 genomes (>14 genomes) and r^2 was computed using the ld function (r package
622 snpStats) 115. The LTCs were automatically extracted from the hierarchical clustering
623 (hclust function in r, method = "ward.D2") of the dissimilarity matrix ($1-r^2$) using the
624 function cutreeDynamic (method = "hybrid", deepSplit = 4 and minClusterSize = 3; r
625 package dynamicTreeCut) 116. Similar to the context of linkage disequilibrium (LD) 117,

- I wonder why the authors used different similarity/dissimilarity metric (phi and $1-r^2$) and clustering algorithm (apcluster and hclust) for GFC and LTC delineation. Have the authors checked the robustness of the results; i.e. whether the conclusions remained similar, regardless of the distance and clustering methods used?

Both of the row dendrogram and the column dendrogram in "Fig. 1: Atlas of Marine Microbial Functional Traits" were computed by the same clustering algorithm; e.g. hierarchical clustering (hclust function in R, method = "ward.D2") of the dissimilarity matrix $(1-r^2)$?

I wonder why the authors performed the hierarchical clustering on "the dissimilarity matrix $(1-r^2)$ " (one minus the square of Pearson's correlation coefficient) ranging from 0 to 1, instead of one minus the Pearson's correlation coefficient $(1-r)$ ranging from 0 to 2.

Could the authors provide all the scripts/codes used for "Statistical analyses"?

Reviewer #3 (Remarks to the Author):

Comments to Authors:

The present study entitled "Comparative whole-genome approach to identify traits underlying microbial interactions" by Zoccarato et al., shows the results of bioinformatic analyses applied to 473 genomes previously isolated from different marine ecosystems to identify their putative functional genetic and interaction traits. The Authors conclude that their study could be a basic reference for investigating clusters of functional and linked traits to better understand fundamental rules across biomes.

Overall assessment:

Investigations on microbial interactions with marine organisms and their functional diversity in marine ecosystems are timely and are relevant in ecological studies. The ms is well written, with very appealing figures. The methodologies are clearly described. The authors made a number of elaborations to justify their conclusions, but I have some major concerns with this study. First of all, the manuscript doesn't meet the expectations created by the Title and the Abstract. I found that the authors in the Results and Discussion of the ms, provided a technical report of the results obtained by their bioinformatic approach without providing evidence of new findings shading light on ecological microbial interactions, but just leaving the reader with the possibility of expanding the present knowledge in future papers. I think that a reader of Communications Biology wants to understand what has been discovered and how it advances our understanding in the field, not just reading a bioinformatic exercise.

The authors commented on the specific bacterial clusters identified and their interactions in light of the available literature (not exhaustively reported), and concluded what is obvious (e.g., "we propose that organisms belonging to the same GFC are likely to interact in similar ways with other microorganisms"; "phylogenetic affiliation of bacteria can predict the traits encoded in their genomes"; "important role of vitamins in shaping marine microbial communities"; "the production of phytohormones may be linked with other interaction traits"). At the end of the manuscript, I couldn't find a take-home message or a new finding.

Another critical issue of the study is the reliability of the approach used. The same authors highlight (for example, lack of representativeness of microbial genomes, statistical nature of presumed functional traits) technical limitations in some parts of the study, conclusions included. I appreciated their honesty, but since this scientific article is primarily based on the development of a bioinformatics approach, the lack of evidence of its performance and of trust of the authors themselves undermine the findings of the paper. I do not want to belittle the work behind this metadata analysis, but I think that the results of this work are not sufficiently robust to be published in Communications Biology.

Additional comments

Title and Abstract

The Title and Abstract are misleading, i.e., are not consistent with the results and discussions. Microbes do not include only heterotrophic bacteria, but also chemotrophs, archaea, fungi and, also viruses (despite this latter component cannot be considered as truly living organisms). In the Abstract, the Authors mentioned bacterial "ecotypes" (line 35), new ecological units and bacterial lifestyles (line 40), so I expected that they addressed their results highlighting distinct biogeographic populations adapted to specific environmental conditions and identified new populations (since in ecology an ecological unit is a population, a community or an ecosystem). Reading the Results and Discussion, I realized that they referred just in some cases to the habitats but without any in-depth discussion.

Lines 43-46. How the Authors can demonstrate and validate the efficiency of their approach?

Introduction

The introduction provides only part of the information needed on the interactions between phytoplankton and marine heterotrophic bacteria.

Line 54. This statement is too general and vague and I think that interactions are not "a feature".

Line 56. In the oceans, heterotrophic bacteria interact not only with phytoplankton. They have key roles in biodiversity and functioning of marine ecosystems, and their interactions with all trophic levels, including viruses, have been published in hundreds of studies.

I think that if the Authors' intention was to focus their study on the interactions between heterotrophic bacteria and phytoplankton they should have specified it at the beginning of the paper.

Lines 101-107. These statements are so approximate and general that they make it difficult to understand the new findings of this work beyond the bioinformatics method used.

Results and Discussion

Line 113. A question is: How and why the Authors selected 473 genomes? Are these sufficient to make broad-spectrum predictions? Frankly, I think that among NCBI, JGI and other databases, even considering only "super-clean" genomes from marine environment, more than 473 genomes can be used. So, I wonder if this subsample of genomes was created to match the processing capabilities of their data set (which is actually complex) rather than because they are truly representative of marine prokaryotic diversity.

Lines 151-154 Authors should highlight the novelty of the conclusions of their work, when compared to available literature (e.g., Louca et al. 2016, Science, Louca et al. 2015 NEE).

Lines 175-177. In my opinion, the reliability of this approach cannot be demonstrated through the correspondence of GFCs and previous literature information on bacterial species. The use of the term "ecotype" is incorrect because ecotype doesn't mean that some species have similar lifestyles.

Lines 163-188. All this part is a description of literature information.

Lines 189-191. This statement is approximate and vague.

Lines 191-192. Please, clarify.

Fig.1. Can the absence of genetic traits in most of the genomes analysed affect the reliability of the approach? And, how much? Please, clarify.

Lines 216-217. The lack of confidence in this sentence weakens the study because the message that comes to the reader is that this approach is not so reliable.

Lines 233-236. Which is the novelty compared to the previous literature?

Lines 239-241. And what does this imply?

Lines 244-246. This is a further confirmation that the approach requires validation, as reported by the authors themselves.

Lines 249-255. These parts, such as in other paragraphs, should be avoided in the discussion because they are more appropriate for an Introduction.

Lines 313-314. I'm sorry but I cannot understand the novelty of this conclusion.

Lines 320-325. This part is a description of literature information.

Lines 335-3360. I cannot understand: if in this study the authors highlight the role of siderophores as "keystone molecules" but cite a previous study, this means that it was not a new finding. The same for following sentences.

Line 366. Why is this "perhaps surprising"?

Lines 412-413. In this paragraph the authors did not draw main conclusions.

Lines 457-458. This is a finding too much approximate.

Lines 477. I agree that the information obtained from this study may provide insights for experimental studies, but once again Authors have not explained the ecological (or biogeochemical) implications of their results, which remain purely descriptive.

Figure 4. I struggle to understand what the authors are referring to when they write "a hypothetical bacterial cell" considering that a wide variety of bacterial cell types exist.

Conclusions

Lines 481-483. Not sure that GFCs and LTCs can be defined as concepts. However, this sentence is unclear.

Lines 486-489. This sentence is vague and too much general.

Lines 490-496. Here you did not highlight the new findings but just explained what you did. I agree that this approach could be a useful approach to study microbial functional traits in other studies but what about your investigation?

Lines 500-508 + Lines 511-514. I think any study may have imperfections, because scientific research proceeds even in small steps. However, a so high level of uncertainty showed by the Authors in an investigation based almost exclusively on the development of a bioinformatics approach, in my opinion, greatly weakens the study.

Lines 517-518. I agree that this approach can be useful for ecologists, but an ecologist first wants to understand if this approach works and what answers it can provide and to what extent.

Materials and Methods

Could you please clarify the number of pre-screening genomes considered? If I understood well, 473 is the number of genomes obtained after screening. I'm wondering why you obtained only 473 genomes of high-quality and not many more as one would expect.

How did you select the genomes from planktonic, benthic, extreme environments and so on?

The analysis of pathway completeness is dependent upon the use of the gene prediction software, (i.e., Prokka). This software is perfectly suitable for broad-scale prokaryotic genome annotation, but it's not accurate and (also by the authors' admission) requires manual re-annotation and review, which can add a further level of risk. In addition, the heavy reliance of the authors on KEGG (which is very refined but fails to correctly assign more distantly-related protein sequences) might lead to biases in the completeness analysis. More recent advancements, introduced by programs such as DRAM or METABOLIC, could allow a more automated, reliable and reproducible analysis for the scope of the present study.

Reviewers' comments:

Reviewer #1 (Remarks to the Author):

The manuscript by Zoccarato et al examines known traits that bacteria use in microbial interactions across 473 sequenced genomes, mainly focusing on marine bacteria. They find overarching patterns of prevalence of specific traits like antimicrobial strategies and phytohormone production and limited patterns of other traits like vitamin biosynthesis/transport. This manuscript is quite interesting and provides an overview of bacterial genome metabolism in the marine environment, based on sequenced (and thus mostly cultured) genomes. In general, I believe the topic is suitable to publication in Communications Biology. I do have some questions, concerns and comments below.

Lines 29-33: First sentence is misleading. The genomes examined in the manuscript cannot be argued to be good representation of microbial diversity. These are mainly from lab cultures and thus are biased. Thus the examination of the genomes here only tells us about the prevalence of these interactions in mostly cultured bacteria, still. Rephrase.

We thank the Reviewer for this critique, which prompted us to test to what extent the genome dataset is a useful representative of natural diversity, using two datasets, one from a coastal community and one from an oligotrophic open-water one. We have now added the results to the manuscript (L164-170): *“Firstly, a significant fraction of the natural community was represented with high fidelity in the GFCs (mean 22.9% of the 16S rDNA reads, range 12.7-44.3%). Thus, despite the inherent bias derived from using only high-quality, closed genomes available (mainly) from cultured bacteria (legend of Figure 1), the GFCs represented a significant fraction of bacterial diversity. Similar results were obtained with an open-ocean community from the Eastern Mediterranean sampled each season for 3 years⁵⁸ (mean 13.9% of the 16S rDNA reads, range 0.5-60.0%), where the GFCs represented a significantly higher fraction of the microbial community on particles >11 µm compared to free-living bacteria (5-0.22 µm; Supplementary Fig. 5)”*.

We have also removed the misleading statement (“whole diversity”) from the sentence in the abstract (L23-24): *“Yet, most interaction mechanisms are studied only in model systems and their prevalence across microbial diversity is unknown”*.

I would've liked to see a tree or a pie chart in the supplementary (or as part of Fig. 1) to show the distribution of the genomes used and how much representations from different phylum level. Hard to glean this information from Fig. 1 because of the clustering. Alternatively, provide the % of each classification next to their names in Fig. 1. What is the clustering method(s) used in the figure? What do you mean relevant LTC are marked? I only see a number next to some. Be more specific.

As suggested, we added the Phyla coverage as percentage in the caption of Figure 1 and we improved the figure legend providing all suggested details:

“Atlas of Marine Microbial Functional Traits showing presence/absence of genetic traits across all analysed genomes. Columns represent genomes grouped into genome functional clusters (GFCs) as shown by the horizontal grey bar. The horizontal colour bar represents the taxonomic affiliation of genomes (mainly phyla, with the exception of Proteobacteria that are represented at the class level). The number next to each taxon in the legend represents the percent from the total genomes analysed. Rows represent the genetic traits grouped into linked trait clusters (LTCs) as shown by the vertical grey bar. LTCs discussed in the text are labelled with the related number alongside the grey bar. Left-side row annotations show the frequency of each genetic trait across all tested genomes and the annotation tool (colour coded bar) with which they have been identified. Both dendrograms are computed using the aggExCluster function (r package aplcluster) that generates hierarchal clustering from an affinity propagation result. An interactive version of this figure is available as Supplementary file 1.”

Lines 130-132: A bit confusing the way it's phrased. Please rephrase.

We rephrased the sentence to improve its clarity: L129: “[...] we defined a GFC as taxonomically coherent (i.e. monophyletic) when all grouped genomes belonged to the same taxon and all genomes of that taxon were grouped in that GFC (see Supplementary Fig. 3 and Supplementary text for more information). Monophyletic GFCs imply that the taxonomic affiliation of these bacteria can predict the traits encoded in their genomes.”

Lines 155-164: The way the ecotype clustering is phrased in a way contradicts the earlier conclusion in lines 151-154. Rephrase to clarify.

We completely revised this section discussing the taxonomic coherence analysis:

L108-114: “Previous genome comparison approaches have identified genome clusters that match to ecologically relevant groups (e.g. ecotypes, as defined for Bacillus pumilus³⁶ and Prochlorococcus²⁸) or lifestyles (e.g. oligotrophic and copiotrophic species³⁸). Similarly, in our analysis we found GFCs that represent group of organisms with a defined ecology and life history, such as the Pelagibacterales group (GFC 2), different ecotypes of Cyanobacteria (GFCs 15 and 36), or Vibrio groups, characterized by different host-specificity and pathogenicity (GFCs 25 and 47)(see Supplementary text)”

L139-142: “Overall, half of the detected GFCs (i.e. monophyletic) support the existence of a strong correlation between taxonomy and functionality in marine bacteria, while the remaining paraphyletic GFCs highlight that, in some cases, the taxonomic partitioning (based on the Genome Database Taxonomy⁴⁸) do not completely reflect the functional differentiation.”

Line 168: free-living

Corrected

Lines 183-191: I wonder if the authors considered the fact that some bacterial groups/genera are represented by a lot more genomes than others and how that may skew conclusions about the presence of specific traits. I realize there was a good reason to focus on only closed, circular genomes but that also limits the number of representative genomes in each genus, thus complicating the confidence one should have in trait patterns across GFCs.

We thank the Reviewer for this valuable comment as this matter was also thoroughly discussed during the analysis. As asked also by Reviewer #2, we have revisited our analyses of GFCs and LTCs inspecting the robustness of the methods and of the resulting patterns. A brief summary of these analysis was added to the material and methods section, while, in a new section in the supplementary text, we addressed possible issues related to the over-representation of specific taxa:

L340-345: “[...] we tested the robustness of the detected GFCs and LTCs by down-sampling the most represented taxa, i.e. Gammaproteobacteria (34% of genomes), Alphaproteobacteria (22%), Cyanobacteria (15%) and Bacteroidota (11%). For each taxon, we randomly sampled 80%, 60% and 40% of the genomes 100 times and checked how often the genomes or the genetic traits were grouped in the same GFCs and LTCs, respectively.”

L346: “Most GFCs always clustered together throughout each of the 100 starts and at all levels of down-sampling, ...”

L351: “Similarly, also most of the LTCs clustered together throughout each of the 100 starts and at all levels of down-sampling. ...”

Lines 222-224: I’m curious if functions that by default must be in all genomes are indeed in common LTCs, such as DNA synthesis, ribosome, etc. And these should be linked I think. This would be a good positive control, much more so than pentose phosphate pathway and parts of the TCA cycle.

We agree that this is an excellent idea for an additional, potentially stronger positive control. We checked, and indeed, as hypothesized by the reviewer, pathways related to DNA synthesis and ribosomes are part of LTC 3. We rephrased the sentence to highlight this aspect. L330-333: “Two LTCs, 3 and 5 (mean r of 0.30 and 0.35), occurred in nearly all genomes (>93%; Fig. 1) and, as expected, they linked traits which mediate for core metabolic functions, common to any cell. These include biosynthesis of nucleotide (DNA and RNA) and amino acids, as well as core metabolic pathways (glycolysis, pentose phosphate pathway and the first three reactions of the TCA cycle)”

Lines 244-246: Why mainly associated with microbial interactions rather than response to the environment?

The sentence has been removed and the final part of this paragraph has been rewritten to better discuss the presented findings as commonly suggested by reviewers and editor.

Fig. 2 is very small. Please expand. Also define abbreviations in the caption. When you mention vitamin transport, does that mean export or import? There’s a brief mention of that (lines 283-284) but not much else. How may that explain the variations observed in biosynthesis vs transport across genomes? Have you also considered vitamin B3 (niacin)? Some bacteria have been shown to acquire it from phytoplankton (Cooper et al, 2018 ISME J).

We revised the panels of Fig. 2 (now Fig. 3) and increased the size of the plots to improve their readability. We have also substituted the word ‘UpSet’ with ‘Plots of intersecting sets’.

With regard to the transporter directionality, we assume they are for import, as the directionality is difficult to reliably determine, and discuss the possibility of export in the supplementary information:

L205: *“A more detailed analysis of the genomes suggested that marine bacteria could be divided into three main groups based on their predicted strategy for B vitamins acquisition: (1) “consumers”, which lacked the biosynthetic genes but harboured the vitamin transporters (we assumed transporters were for uptake, see Supplementary text for details on transporters’ directionality); ...”*,

Suppl. text L 191-197: *“Genomes with a flexible strategy could potentially act also as “source” for certain vitamins and represent key players in the vitamin market (e.g. ³³⁻³⁵). However, the transport directionality can be reliably assigned only to specific transporter families (<http://www.tcdb.org/superfamily.php>) and we could identify only efflux-transporters for vitamin B1 (see Supplementary table 4 for directionality annotation) which were encoded in Alphaproteobacteria and Gammaproteobacteria genomes (Supplementary Fig. 8b). Moreover, it has to be kept in mind that B vitamins are water soluble molecules and could passively diffuse from a producing cell ³⁶. ”*

Regarding vitamin B3, we have identified the related transporter in LTCs 29 and 30 (Supplementary table 7), as long as the biosynthetic pathway for other vitamins like B2 (LTC 5), B6 (LTC 11), E (LTC 22) and K2 (LTC 23). However, there was no clear patterns of specific bacterial strategies for such vitamins (e.g. flexible/consumer/independent). We discuss this in the supplementary information (L200-205).

Lines 296-300: Confusing as written, rephrase.

We removed this sentence as the example provided is not relevant anymore in the revised version of this paragraph.

Line 301: B-vitamin consumers. Also for all vitamins, make the numbers subscript throughout.

We have changed the text accordingly.

Lines 308-311: Auxotrophy can be resolved with provision of the vitamin or a precursor (see McRose et al, 2014 ISME J). Therefore, a genome can have only the second half of the biosynthesis of an enzyme as it may rely on the precursor’s presence in the environment, which seems to be the case in some of the genomes. Can you make some predictions regarding this using your analysis? This would add tremendously to our understanding of vitamin precursors. I am curious to see if there is a way to differentiate dependency on precursors vs gene loss.

We thank the Reviewer for this interesting comment. Reliance on B-vitamin precursors was partially discussed in the supplementary text (“Problematic vitamin cases”, now renamed “Broken vitamin pathways”). We revised the analysis and we identified cases in which it was possible to suggest potential dependencies on specific B vitamin precursors and cases in which annotation issues (e.g. unknown functional genes) were more likely. We have added this information in the supplementary text and also more clearly referred to such analysis in the main manuscript:

L220-222: *“While the perceived lack of biosynthetic capacity could be due to the utilization of precursors or to gaps in the pathway annotations (see Supplementary Fig 10 and Supplementary text), we speculate that ...”*.

Suppl text L231-237: *“We found that nearly all genomes with problematic vitamin B1 annotations possessed truncated biosynthetic pathways. However, while some of these genomes showed the capability to grow on exogenous precursors (e.g. pyrimidine moiety, HMP or the thiazole moiety, HET), others*

lacked the last enzyme of the pathway (Supplementary Fig. 10b). A few cases of genomes potentially relying on exogenous precursors were also identified for vitamin B7 (e.g. d-desthbiotin) and vitamin B12 (e.g. cobyrinic acid a,c-diamide or adenosylcobinamide). The rest of the problematic genomes possessed none or only a few annotated genes for such pathways (Supplementary Fig. 10c,d)."

Figure 3: arrows are too large. Is there a way to represent these functions in a different way?

In the new version of Figure 3 (now Figure 2), we have replaced the use of arrows to indicate specific genomes and we now simply highlight with different colours the genome names.

Line 348: tryptamine?

Corrected

Figure 4: Caption is missing a lot of information. For example: What do dashed arrows represent? What is the purple plus sign mean? I believe Lux receptors are supposed to be intracellular, not membrane bound. Why are there empty LTCs inside the bacterial cell? While I realize the authors are focusing mainly on bacterial interactions with phytoplankton, many of the functions discussed (except for some specific examples) may occur between bacteria and other bacteria (antagonistic for example) or between bacteria and other eukaryotic hosts (fish, zooplankton, etc). This should be reflected in the figure. Also some bacteria probably use chemotaxis, motility and adhesion to also attach to phytoplankton not just to marine snow.

We added a graphic caption to explain some of the symbols used in the conceptual figure and modified the figure following the useful suggestions of the reviewer. We have also improved the text caption: *"Fig. 4: Conceptual representation of the predicted interaction dynamics for an hypothetical bacterium analysed with our trait-based approach. The bacterium is assigned to a GFC, visualized here as a jigsaw puzzle. Every puzzle piece represents one of the linked trait clusters (LTCs) possessed in that GFC. LTCs 3 and 5 (marked in red) are the core LTCs and they are present in any GFC. In this example, the GFC of the hypothetical bacterium also possess the ancillary LTCs 11, 17 and 19. Each of these ancillary LTC predicts specific interaction types with bacteria, organic matter particles, phytoplankton or other eukaryotic hosts (e.g. fish, zooplankton). Green arrows indicate traits with positive effects (e.g. enhancing growth), grey arrows traits mediating metabolites/chemical exchange, movement or attachment, and red arrows traits with negative effects (e.g. pathogenicity). A solid arrow is used when the mediated mechanism has already been described in literature, while a dashed arrow indicates a potential new mechanism."*

Lines 434-437: Are these phylogenetic groups represented by a lot of genomes or just a couple? It would be helpful to list the number of genomes a specific GFC represents that contain a particular LTC. This should be done wherever relevant in the manuscript.

While we agree that this information is important, we think that it would make the text less readable when adding the size of the GFC whenever mentioned. However, we added a "GFC size" column in the supplementary table listing all GFC details.

Supplementary:

Supplementary file, line 41: What happens in the pipeline if the stringency was stricter for gap allowances. In other words, instead of 1 gap allowed in KMs with more than 3 steps, what is the effect on the outcome if you allowed 1 gap in KMs with more than 4 or 5 steps? A benchmark to specify how this cutoff was determined will be useful.

We thank the Reviewer for this suggestion. We cannot provide a “real” benchmark, e.g. a ROC curve showing the rate between False Positive and True Positive at different completeness threshold, as we lack the real observations for when a KM is “really” present in an organism. However, we added in the Supplementary text a detailed report section (L304-325) explaining how we empirically determined the best threshold for the “gap-filling” by comparing the frequency of the reconstructed KMs with different completeness thresholds (Supplementary table 10). We also included this benchmark step in our annotation workflow presented in Supplementary Fig.1.

In Table S1, why only these traits? Why not add other ones that have been shown to be important as well. For example, organic sulfur compound uptake and metabolism by bacteria from phytoplankton. Many of the genes are known (e.g., uptake and metabolism of DHPS, taurine and N-acetyltaurine).

We have manually added the annotations for genetic traits involved in the metabolisms (i.e. uptake and/or catabolism) of sulfur compounds such as DMSP, DHPS and taurine. Although these processes were showed to occur during phytoplankton-bacteria interactions, they are not specific enough to represent markers of biotic interactions in a bioinformatics analysis and we didn't include them in Table S1 (now Table S2). Nevertheless, these traits are know clearly marked in the Table S7, and their functional linkage with other processes can be easily inspected to develop future analyses.

Line 66: rephrase sentence.

We have rephrased the sentence.

Suppl. Fig. 5: The GFC numbers at the top should be better labeled. Skewing them is very confusing. You can draw a line between the colored box and the skewed number to make it easier to see. Or Alternatively, increase the size of the heat map, so that each pixel is bigger and thus would accommodate these numbers.

We thank the Reviewer for the suggestion and we updated the figure by adding lines that connect the skewed labels to the GFC colour boxes.

Supplementary Fig. 9: what is lm?

There was a mistake in the y-axis labels of panels b and c of Figure S9 (now Figure S7). We fixed it and specified in the legend that ‘lm res’ indicates the residuals of the linear regression models.

Reviewer #2 (Remarks to the Author):

The authors analyzed publicly available genomes of bacteria from marine and other environments, focusing on genetic traits putatively involved in mediating interactions between microbes. The so-called "interaction traits" (Fig. 3) were identified based on bioinformatics tools and databases such as metabolic KEGG modules (Supplementary Table 5), and a careful literature search (Supplementary Table 6). The bacteria were clustered into the "genome functional clusters" (GFCs) based on presence/absence of the genetic traits (the column dendrogram in Fig. 1); some of the GFCs correspond to previously defined ecotypes. The genetic traits were clustered into the "linked trait clusters (LTCs)" based on their presence in bacteria (the row dendrogram in Fig. 1); the LTCs present in >90% of genomes were called "core" traits. While the authors stated that "We developed a new genomic approach", I think the GFCs and LTCs are essentially same as clusters identified by gene content analysis (Snelet al., 1999) and phylogenetic profiling methods (Pellegrini et al., 1999). A major revision, based on the comments given below, is needed to make this manuscript suitable for publication.

#) Interactions

Lines 64-67: Recent studies, using specific model organisms in binary co-cultures, have started to elucidate mechanisms underlying marine microbial interactions (mostly between bacteria and phytoplankton). Many of these interactions are mediated by the exchange of metabolites used for growth or respiration.

Lines 101-107 We propose that organisms belonging to the same GFC are likely to interact in similar ways with other microorganisms. We also identify clusters of traits that are statistically linked, and propose that these linked trait clusters (LTCs) may have evolved to function together, potentially during microbial interactions. GFCs and LTCs provide a framework to extend the knowledge on microbial interactions gained from specific model systems, leading to testable hypotheses as to the prevalence of microbial interactions across bacterial diversity.

Lines 125-128: Within each GFC, all genomes are inferred to encode similar traits, and thus these GFCs are expected to be coherent in terms of their functional and metabolic capacity, including the ways in which the related organisms interact with other microbes.

Lines 244-246: We therefore interpreted the patterns of LTCs associated with microbial interactions, keeping in mind that these represent bioinformatics predictions requiring experimental validation.

- The authors should cite reference papers supporting the assumption/interpretation on interactions. Is there any experimental evidence showing the association between gene contents (functional categories) and interaction capabilities?

We revised all those sentences, providing the relevant literature, as well as restructuring the discussion of the GFC and LTC concepts to highlight their fundamental principles and implications.

L54: "Studies using specific model organisms in binary co-cultures have started to elucidate mechanisms underlying marine microbial interactions (mostly between bacteria and phytoplankton)^{1,2,5}. Although these results..."

L78: “Several previous studies have aimed to characterize and cluster genomes based on their predicted functional similarity, defined usually using individual genes (e.g. ^{34–37}) or coarse functional categories (e.g. COGs, ^{36,38,39})(Supplementary Table 1, Supplementary text). We chose to take a trait-based approach rather than a gene-based one, which is an intermediate level of resolution between individual genes and coarse functions. Trait-based approaches offer a new perspective to investigating microbial diversity with a more mechanistic understanding ⁴⁰ but have been used only in few specific cases to highlight putative bacterial interactions (e.g., ⁴¹). We focused on the following traits: 1) the metabolic capacity to synthesize or degrade specific biomolecules and to regulate such processes (KEGG modules); 2) specific functions related to microbial interactions, such as motility, chemotaxis and the ability to produce molecules such as siderophores, toxins and antibiotics. This allows to ...”

L108: “Previous genome comparison approaches have identified genome clusters that match to ecologically relevant groups (e.g. ecotypes, as defined for *Bacillus pumilus* ³⁵ and *Prochlorococcus* ²⁸) or lifestyles (e.g. oligotrophic and copiotrophic species ³⁸). Similarly, in our analysis we found GFCs that represent group of organisms with a defined ecology and life history, such as the Pelagibacterales group (GFC 2), different ecotypes of Cyanobacteria (GFCs 15 and ³⁶), or *Vibrio* groups, characterized by different host-specificity and pathogenicity (GFCs 25 and 47)(see Supplementary text). ”

L318: “If these interaction mechanisms are evolutionarily conserved, traits that are functioning together to mediate such interactions should co-evolve, meaning that selection would favour maintaining all relevant traits in the same genome ^{95,96}. To identify cases of co-evolving traits, we used linkage disequilibrium analysis and clustered traits which were found together more often than expected by chance (adjusted p-value < 0.05) into Linked Trait Clusters (LTCs; Figure 1, Supplementary Table 7 and Supplementary Fig 13). For example, ...”

Supplementary text L93: “The correlation between gene content and phenotype has been shown for some traits (e.g. motility ¹⁷), however, several genetic traits may be not constitutively active. ”

Organisms whose genomes have the similar gene contents (functional categories) interact in similar ways with other microorganisms?

This idea is the underlying hypothesis for our study, and is shared by many other studies that look at the presence of genetic traits as a mean to identify functionally coherent units of bacteria. We now clearly state this in the introduction, and provide citations for the previous studies:

L78: “Several previous studies have aimed to characterize and cluster genomes based on their predicted functional similarity, defined usually using individual genes (e.g. ^{34–37}) or coarse functional categories (e.g. COGs, ^{36,38,39})(Supplementary Table 1, Supplementary text).” .

L84: “We focused on the following traits: 1) the metabolic capacity to synthesize or degrade specific biomolecules and to regulate such processes (KEGG modules); 2) specific functions related to microbial interactions, such as motility, chemotaxis and the ability to produce molecules such as siderophores, toxins and antibiotics. This allows to classify genomes into coherent functional units, ...”

We also discuss the limitations of this in the conclusions section:

L389: “The LTCs provided evidence for the functional and evolutionary linkage between traits, raising hypotheses as to how these traits act together in the context of complex processes such as microbial interactions.”

L412: “Finally, the GFC and LTC concepts are both statistical in nature, representing the probability of

bacteria having similar functional capacity, and of traits being functionally and/or evolutionarily linked. In support of the GFC concept, a study of marine Vibrionaceae suggested that ecologically-cohesive unit (similar to GFCs) are likely to interact in a similar way²⁹. However, some interaction phenotypes, such as the ability to inhibit multiple target bacteria ("super-killers"), are not phylogenetically conserved^{29,113}. Experimental studies using both established and new model systems across multiple scales of diversity (e.g. between and within GFCs) are needed to test the GFC framework, whereas genetic manipulation of linked traits to test their effect on microbial interactions will be required to determine to what extent traits within LTCs are functionally linked. "

Are microbial interactions/behaviors in microbial communities consisting of >2 microbes same as those in binary co-cultures (one-to-one microbial interactions)?

This is an excellent point. Whether pairwise interactions are sufficient for explaining complex community dynamics is the subject of active research, especially from the theoretical side (see, for example the review by Levine et al., 2017, Nature). From an experimental perspective, microbial interactions have been studied mainly in binary co-cultures which are not representative of the natural complexity of bacteria (e.g. in organic particles, phycospheres) and the dynamics described in binary experiments might not occur at the same magnitude in nature (e.g. Amin et al., 2015, Nature). However, this knowledge represents an essential starting point to understand how microorganisms interact. We believe that our study, by expanding this knowledge to a larger diversity of marine bacteria, can help to disentangle interaction processes in more complex and "natural" microbiomes. We have amended the text to clearly state this limitation:

L54: "Studies using specific model organisms in binary co-cultures have started to elucidate mechanisms underlying marine microbial interactions (mostly between bacteria and phytoplankton)^{1,2,5}. Although these results do not reflect the complexity of natural environments and the potential for higher-order effects, they allow to identify the chemical signals and resulting changes in gene expression and physiology that underlie these interactions. For example, ..."

#) Novelty of approach

- The authors stated that "We developed a new genomic approach" with two keywords "genome functional clusters (GFCs)" and "linked trait clusters (LTCs)". I wonder if the genome functional clusters (GFCs) and linked trait clusters (LTCs) stated by the authors are conceptually different from clusters identified by gene content analysis (Snel et al., 1999) and phylogenetic profiling methods (Pellegrini et al., 1999)?

References:

<https://pubmed.ncbi.nlm.nih.gov/10200254/>

Comparative Study Proc Natl Acad Sci U S A. 1999 Apr 13;96(8):4285-8. doi: 10.1073/pnas.96.8.4285.

Assigning protein functions by comparative genome analysis: protein phylogenetic profiles

M Pellegrini 1, E M Marcotte, M J Thompson, D Eisenberg, T O Yeates

<https://pubmed.ncbi.nlm.nih.gov/9916801/>

Nat Genet. 1999 Jan;21(1):108-10. doi: 10.1038/5052.

Genome phylogeny based on gene content

B Snel 1, P Bork, M A Huynen

<https://pubmed.ncbi.nlm.nih.gov/28066816/>

mSystems. 2016 Dec 27;1(6):e00101-16. doi: 10.1128/mSystems.00101-16. eCollection Nov-Dec 2016.

From Genomes to Phenotypes: TraitAr, the Microbial Trait Analyzer

Aaron Weimann 1, Kyra Mooren 2, Jeremy Frank 3, Phillip B Pope 3, Andreas Bremges 4, Alice C McHardy 1

<https://www.ncbi.nlm.nih.gov/pmc/articles/PMC5192078/>

FIG 5 Single-cell phenotyping with TraitAr.

<https://github.com/hzi-bifo/traitar>

<https://pubmed.ncbi.nlm.nih.gov/30285626/>

BMC Evol Biol. 2018 Oct 3;18(1):148. doi: 10.1186/s12862-018-1261-7.

Environmentally-driven gene content convergence and the Bacillus phylogeny

Ismael L Hernández-González 1, Gabriel Moreno-Hagelsieb 2, Gabriela Olmedo-Álvarez 3

<https://www.ncbi.nlm.nih.gov/pmc/articles/PMC6171248/>

<https://bmcevolbiol.biomedcentral.com/articles/10.1186/s12862-018-1261-7>

Fig. 4 Hierarchical clustering using COGs-based Jaccard's distance (Jd).

Fig. 5 Cluster analysis using Jaccard's distance (Jd) based on Figfams.

<https://pubmed.ncbi.nlm.nih.gov/30137329/>

Genome Biol Evol. 2018 Sep 1;10(9):2255-2265. doi: 10.1093/gbe/evy178.

Phylogenetic Clustering of Genes Reveals Shared Evolutionary Trajectories and Putative Gene Functions

Chaoyue Liu 1 2, Benjamin Wright 1, Emma Allen-Vercoe 3, Hong Gu 2, Robert Beiko 1

<https://www.ncbi.nlm.nih.gov/pmc/articles/PMC6130602/>

<https://academic.oup.com/gbe/article/10/9/2255/5076814>

FIG. 5. —Structure, phylogenetic distribution and functional categories of a hierarchical cluster enriched in amino-acid biosynthesis proteins.

FIG. 6. —Structure, phylogenetic distribution and functional categories of a hierarchical cluster enriched in flagellar motility proteins.

<https://pubmed.ncbi.nlm.nih.gov/27658251/>

PLoS One. 2016 Sep 22;11(9):e0163098. doi: 10.1371/journal.pone.0163098. eCollection 2016.

Taxonomic Identity Resolution of Highly Phylogenetically Related Strains and Selection of Phylogenetic Markers by Using Genome-Scale Methods: The Bacillus pumilus Group Case

Martín Espariz 1 2, Federico A Zuljan 1 2, Luis Esteban 3, Christian Magni 1 2

<https://www.ncbi.nlm.nih.gov/pmc/articles/PMC5033322/>

<https://journals.plos.org/plosone/article?id=10.1371/journal.pone.0163098>

Fig 3. Comparison of phylogenomic and functional dendrograms of *Bacillus pumilus* group strains.

We thank the Reviewer for the comment and the detailed information provided in it. Our framework is indeed inspired by others works that dealt with whole-genome analysis and comparison. We expanded on the literature provided in the comment and carried out an extensive search, presented in Supplementary Table 1, and accompanied to citations to all the references mentioned by the Reviewer. We have added a section in the supplementary text discussing similarities and differences between studies. We now explain how our analysis is different from the previous ones (both in terms of completeness and number of genomes analyzed, and in terms of the resolution of the analysis – see points below) and can contribute to our understanding of microbial functional and interaction potential:

L26-34: *“Genome-based studies (including both draft and complete genomes) mainly focused on specific taxa (e.g. Bacillus, Clostridia, Roseobacter) ¹⁻⁵, although two notable exceptions focused on a wide diversity of marine bacteria ^{6,7}. Based on their genomes, marine bacteria can be classified into two main groups – oligotrophs, which are often highly abundant, and copiotrophs, which are often less common but can grow rapidly in energy-rich environments ⁶. These two groups differ in ...”*

L47-51: *“Overall, the majority of the studies presented in Supplementary Table 1 focused on gene-level annotation ^{1-4,8-13}. Analysing genomes or metagenomes at the single-gene level enabled the resolution of fine differences in the functional capacity between bacteria, e.g. defining ecotypes ² or revealing limited clonality in bacterial communities ¹¹, but often at the cost of a clear overview of the processes and/or pathways actually encoded.”*

L54-58: *“A trait-based analysis was developed to characterize the capacity of different bacteria in terms of multiple substrates utilization, oxygen requirement, morphology, antibiotic susceptibility, or proteolysis. However, the workflow was based on a commercial platform (GIDEON) and mainly focused on medical-related phenotypes and bacteria (belonging to Gammaproteobacteria, Firmicutes, Bacteroidetes, Actinobacteria) ¹⁵. ”*

L59-63: *“In our study, we chose an approach that builds upon previous knowledge but differs in two main ways. Firstly, our analysis encompassed a wide taxonomic diversity of marine bacteria (421 strains, 213 genera), using only complete genomes to minimize false negative occurrence of genetic traits. Secondly, we chose an intermediate functional resolution to annotate these genomes – that of genetic traits, defined here as the presence of complete gene pathways ...”.*

#) “core” traits

Line 120: As shown in Fig. 1, some traits were found across almost all genomes (“core” traits).

Line 218: Based on their occurrence across genomes (Supplementary Fig. 4c), we divided the...

Lines 219-220: LTCs into three subgroups: “core” (present in >90% of genomes), “common” (<90% and ≥30%) and “ancillary” (≤30%).

- Do these subgroups (“core”, “common” and “ancillary”) correspond to three parts (core genome; accessory or dispensable genome; and species-specific or strain-specific genes) in pan-genome?

References:

<https://en.wikipedia.org/wiki/Pan-genome>

<https://sanger-pathogens.github.io/Roary/>

<https://pubmed.ncbi.nlm.nih.gov/31848603/>

Mol Biol Evol. 2020 Mar 1;37(3):933-939. doi: 10.1093/molbev/msz284.

Estimating Pangenomes with Roary

Farrah Sitto 1, Fabia U Battistuzzi 1 2

<https://www.ncbi.nlm.nih.gov/pmc/articles/PMC7038658/>

<https://academic.oup.com/mbe/article/37/3/933/5652084>

<https://pubmed.ncbi.nlm.nih.gov/27006628/>

Curr Genomics. 2015 Aug;16(4):245-52. doi: 10.2174/1389202916666150423002311.

Inside the Pan-genome - Methods and Software Overview

Luis Carlos Guimarães 1, Jolanta Florczak-Wyspianska 2, Leandro Benevides de Jesus 3, Marcus Vinícius Canário Viana 3, Artur Silva 2, Rommel Thiago Jucá Ramos 2, Siomar de Castro Soares 4, Siomar de Castro Soares 4

<https://www.ncbi.nlm.nih.gov/pmc/articles/PMC4765519/>

Pan-genome consists of three parts: core genome; accessory or dispensable genome; and species-specific or strain-specific genes.

From a conceptual point of view, the LTC classes “core”, “common” and “ancillary” are similar to the pan-genome classes of core genome; accessory or dispensable genome and species-specific or strain-specific genes. However, the pan-genome analysis is usually carried out at species and genus levels while in our analysis we compared genomes of different phyla. We therefore decided not to use the pan-genome classes to avoid confusion and to allow us to focus on the interaction traits.

#) Reproducible research

To make their analysis reproducible, the authors should provide sufficient details about the sequence data analyses. All the codes/scripts the authors written should be available. All the parameters used in bioinformatics software (e.g. Prokka) should be available.

We have amended the missing information in the methods, as also suggested by the Reviewer in the following comments. Furthermore, we made available the code used to perform the different functional annotations, reconstruct complete KEGG modules from KEGG orthologues and the statistical analysis for achieving the GFCs and LTCs (see the updated repository https://github.com/lucaz88/genome_comparison_code).

#) Robust research

The authors should check the robustness of the results; i.e. whether the conclusions remained similar, regardless of the methods/parameters and data used.

For example, the conclusions remained similar, regardless of the databases used for functional annotations (KEGG modules, Pfam, Gene Ontology, SEED subsystem, COG functional category) and genome sequence data: representative genomes, reference genomes, and Complete Genome (<https://www.ncbi.nlm.nih.gov/genome/doc/ftpfaq/>).

We thank the Reviewer for this comment. We carried out several tests to provide validation/sensitivity analyses for the most critical parameters and methods adopted in our workflow.

Regarding the parameters, we have investigated: (1) the threshold to call complete a genetic trait and (2) the q value affecting the clustering sensitivity of the affinity propagation. For the assessment of KEGG module completeness (1), we tested different thresholds and provided an empirical explanation supporting our choice for the “best” threshold. The discussion is available in the Supplementary text L304-325. In brief, we used the expected absence/incompleteness of rare traits (e.g. nitrate assimilation and archaeal pentose phosphate pathway, as the analysis included mainly aerobic genomes and no Archaea), and the expected presence/completeness of core traits (e.g. biosynthesis of uridine monophosphate and isoleucine) to identify the best threshold for KEGG modules completeness. The q value (2; Supplementary text L326-337) ranges between 0 and 1, and it affects the number of clusters retrieved by the affinity propagation algorithm. By iteratively running the affinity propagation with different q values, we noticed that cluster solutions were almost identical within the q-range 0.15-0.7 (for both GFC and LTC; Supplementary Fig. 15a-b), underpinning the robustness of the approach and detected clusters. We picked q = 0.5 for both GFCs and LTCs, as it is in almost in the middle of such interval and it is also the recommended value in the r package.

Regarding the methods, we could not compare our annotation with tools such as Pfam because it is a gene based annotation, nor with GO and COG as their hierarchical structure is too coarse (e.g. Lipid metabolism, Signal Transduction or DNA repair) and cannot be compared e.g. with KEGG modules. While, in principle, a similar analysis could be performed using SEED or MetaCyc, unfortunately there is no easy way to “translate” between these databases, and thus any comparison between analyses using different annotations would be qualitative at best. Given the large amount of work needed to re-do the entire analysis in SEED or MetaCyc, the very limited information that can be gained, and the added complication to the manuscript, we do not think such a comparison is warranted. Nevertheless, we run specific analyses to inspect the robustness of GFC and LTC clustering while changing the genomic data in input. As suggested also by Reviewer #1, we performed random down-sampling of the most represented taxonomic groups (i.e. Alpha- and Gammaproteobacteria, Cyanobacteria and Bacteroidota) at 80%, 60% and 40% of their genome coverage. We showed that, in all iterations, it was possible to recover the same GFCs and LTCs, underpinning the accuracy and reproducibility of the detected functional patterns and of the methods (Supplementary text L338-360).

#) in-house R script

Lines 568-571: KEGG orthologies (KOs) annotations generated by kofamscan were recombined in KEGG modules (KMs) using an in-house R script (https://github.com/lucaz88/R_script/blob/master/_KM_reconstruction.R).

Lines 644-646: The code for the reconstruction of KEGG modules is deposited on GitHub and freely accessible (https://github.com/lucaz88/R_script).

- Error in R

Running the R script `*my_analysis.R*` printed the following Error messages as shown in the `*log.txt*` file.

...

```
Rscript --vanilla ./my_analysis.R 2>&1 | tee log.txt
```

```
Error in .getUrl(url, .flatFileParser) : Forbidden (HTTP 403).
```

```
Calls: KMdiagram_fetcher ... resolve.list -> signalConditionsASAP -> signalConditions
```

```
Execution halted
```

...

We could not replicate the error with the information provided by the reviewer. However, we did substantially improve our code repository on GitHub by adding new scripts, providing more clear instructions (on the web-page and within the scripts) on how to use and test them. We also test the mentioned code in 'KM_reconstruction.R' on different laptops to ensure it works well.

#) Line 40: evolved together and point to features of bacteria lifestyles. Specific GFCs, bacteria lifestyles -> bacterial lifestyles

We have changed the text accordingly.

#) Line 129: We next sought to understand the extent to which the GFCs correlate with phylogeny.

"Taxonomy" and "Phylogeny" should not be confused.

If I understand correctly, the authors did not use "phylogenetic information" but used "taxonomic information" (i.e. genus, family, order, class or phylum). For examples, species belonging to the genus "Shigella" do not form a clade in a phylogenetic tree. Bacteria of the order SAR11 (Pelagibacterales) are monophyletic?

References:

<https://www.sciencedirect.com/science/article/pii/S0723202014001398>

Diversity and abundance of "Pelagibacterales" (SAR11) in the Baltic Sea salinity gradient - ScienceDirect

If the authors performed the phylogenetic analysis, they should describe the methods in detail.

Here is an example obtained from MEGA X (<https://www.megasoftware.net/>).

Evolutionary relationships of taxa

The evolutionary history was inferred using the Neighbor-Joining method [1]. The optimal tree with the sum of branch length = 0.34882604 is shown. The percentage of replicate trees in which the associated taxa clustered together in the bootstrap test (500 replicates) are shown next to the branches [2]. The tree is drawn to scale, with branch lengths in the same units as those of the evolutionary distances used to infer the phylogenetic tree. The evolutionary distances were computed using the p-distance method [3] and are in the units of the number of amino acid differences per site. This analysis involved 16 amino acid sequences. The coding data was translated assuming a Standard genetic code table. All positions containing gaps and missing data were eliminated (complete deletion

option). There were a total of 374 positions in the final dataset. Evolutionary analyses were conducted in MEGA X [4][5]

1. Saitou N. and Nei M. (1987). The neighbor-joining method: A new method for reconstructing phylogenetic trees. *Molecular Biology and Evolution* 4:406-425.
2. Felsenstein J. (1985). Confidence limits on phylogenies: An approach using the bootstrap. *Evolution* 39:783-791.
3. Nei M. and Kumar S. (2000). *Molecular Evolution and Phylogenetics*. Oxford University Press, New York.
4. Kumar S., Stecher G., Li M., Knyaz C., and Tamura K. (2018). MEGA X: Molecular Evolutionary Genetics Analysis across computing platforms. *Molecular Biology and Evolution* 35:1547-1549.
5. Stecher G., Tamura K., and Kumar S. (2020). Molecular Evolutionary Genetics Analysis (MEGA) for macOS. *Molecular Biology and Evolution* (<https://doi.org/10.1093/molbev/msz312>).

We thank the Reviewer for noticing this mistake which was the result of a previous analysis and it should have been left out from the earlier version of the manuscript. We have removed this mistake and the manuscript is solely focusing on taxonomy.

#) Lines 151-154: Overall, the observation that functionality does not strictly follow phylogeny makes it difficult to infer the function of a community from its taxonomic structure, as previously shown across the global oceans 34,35.

- The genetic traits commonly present in the genome functional clusters (GFCs) may be due to the independent acquisition of functional gene sets (horizontal gene transfer)?

References:

<https://pubmed.ncbi.nlm.nih.gov/19815525/>

Comparative Study Proc Natl Acad Sci U S A. 2009 Oct 20;106(42):17939-44. doi: 10.1073/pnas.0903585106. Epub 2009 Oct 6.

Comparative genomics reveal the mechanism of the parallel evolution of O157 and non-O157 enterohemorrhagic Escherichia coli

Yoshitoshi Ogura 1, Tadasuke Ooka, Atsushi Iguchi, Hidehiro Toh, Md Asadulghani, Kenshiro Oshima, Toshio Kodama, Hiroyuki Abe, Keisuke Nakayama, Ken Kurokawa, Toru Tobe, Masahira Hattori, Tetsuya Hayashi

<https://www.ncbi.nlm.nih.gov/pmc/articles/PMC2764950/>

<https://www.pnas.org/content/106/42/17939>

The gene contents of the 4 EHECs do not follow the phylogenetic relationships of the strains, and they share virulence genes for Shiga toxins and many other factors.

The independent acquisition of very similar virulence gene sets is predominantly attributable to mobile elements that are commonly present in the EHECs: multiple lambdoid PPs, several types of IEs, and virulence plasmids.

We thank the Reviewer for the comment. Indeed horizontal gene transfer (HGT) could be a major source for the exchange of certain functions. While we did not specifically look for HGT, as it would be beyond the main topic of the manuscript, we highlight the role of such process in the updated GFC discussion.

L139-152: *“Overall, half of the detected GFCs were monophyletic support the existence of a strong correlation between taxonomy and functionality in marine bacteria, while the remaining paraphyletic GFCs highlight that, in some cases, the taxonomic partitioning (based on the Genome Database Taxonomy⁴⁹) do not completely reflect the functional differentiation. [...] One of the main biological processes that mediates such functional “homogenisation” is the horizontal gene transfer^{52,53} and, indeed, most of the paraphyletic GFCs grouped known copiotrophs or particle/host associated bacteria (e.g. Rhodobacteraceae in GFCs 9, 30 and 40, Vibrionaceae in GFCs 25 and 47, Alteromonadaceae in GFCs 21 and 24, Oleiphilaceae in GFC 35, or Halomonadaceae in GFC 43;^{12,17,44,54,55}) whose lifestyles favour events of genetic exchange (e.g.⁵⁶). Contrarily, another study suggested that ”*

#) Line 194: Fig. 1: Atlas of Marine Microbial Functional Traits showing presence/absence of genetic traits across all analyzed genomes. Each column represents a genome and these are hierarchically clustered. The horizontal color bar represents the taxonomic affiliations of the genomes (mainly phyla, with the exception of Proteobacteria that are represented at the class level) and the horizontal grey bar delineates specific genome functional clusters (GFCs). Rows are the genetic traits clustered using the co-efficient of disequilibrium into linked trait clusters (LTCs) (vertical grey bar); relevant LTCs are marked. Left-side row annotations show the average abundance for the genetic traits and the annotation tool (color coded bar) with which they have been annotated. An interactive version of this figure is available as Supplementary file 1.

- In "Fig. 1: Atlas of Marine Microbial Functional Traits", both of the row dendrogram and the column dendrogram were computed by the same clustering algorithm and dissimilarity matrix? In the previous version of the manuscript no, but in the new version we have unified the method and now *“Both dendrograms are computed using the aggExCluster function (r package apcluster) that generates hierarchical clustering from an affinity propagation result.”*.

- Does "coefficient of disequilibrium" indicate "square of Pearson's correlation coefficient (r²)" in Line 619? Yes, but we now removed this detail from the caption (it is clearly stated in material and methods).

- "relevant LTCs are marked" with what? numbers? Yes, we fixed it.

- Does "Left-side row annotations show the average abundance for the genetic traits" indicate "Genome fraction" ranging from 0 to 1? relative abundance of the genetic traits in genome, averaged by each GCF? It indicates the genome fraction that is the frequency of a genetic trait across all tested genomes.

As also highlighted by Reviewer #1, the caption of Figure 1 was not exhaustive, therefore, we rewrote it to meet all comments and suggestions.

#) Lines 196-202: Missing common trait clusters highlight basic metabolic differences Just as the genomes (represented by the columns in Fig. 1) could be grouped into GFCs, the genetic traits that drive this clustering could themselves be grouped into linked trait clusters (LTCs). The traits within each

LTC were found together in the genomes more often than expected by chance, and thus may be linked functionally (Supplementary Fig. 4). For example, LTC 10 includes pathways for assimilatory sulfate ...

- I wonder if the authors performed any statistical test, e.g. `cor.test()` and `fisher.test()` in R, to conclude that "The traits within each LTC were found together in the genomes more often than expected by chance".

We thank the Reviewer for the comment. For each trait pair, we performed a chi-square test to assess the significance of the correlation and corrected the p-values of all pairs for the false discovery rate (FDR). Only the pairwise correlations with a p-adjusted < 0.05 were then used for clustering the generic traits into LTCs. We have also updated the related paragraph in material and methods (L535-536).

#) Lines 227-228: (see Supplementary Table 3 for complete list of all LTCs and their respective genetic traits).

- In the [S_tab3] tab in *6477_0_data_set_158598_qhxx7y.xlsx* file (Supplementary Table 3), does "trait identified as an interaction-triat" indicate "Known interactions (described in the referenced literature)"?

Yes, we have rephrased the sentence accordingly.

"interaction-triat" should be "interaction-trait"

We have made the correction.

#) Line 259-270: [...] many Gammaproteobacteria; Fig. 2b). Of the rest, ~33% synthesize at least two B vitamins ... porters for a specific vitamin (Fig. 2a). [...]

- Figure should be renumbered because Fig. 2b appear before Fig. 2a.

We have changed the order of the panels in Fig. 2.

#) Lines 336-339: Thus, in agreement with recent considerations 75, we highlight the role of siderophores as "keystone molecules" 76 and take this as an indication that the organisms producing them have an important role in the functionality of microbial communities (e.g., references 77,78).

- Does the "keystone molecules" mentioned here indicate "molecules of keystone significance" reported in the following papers?

<https://pubmed.ncbi.nlm.nih.gov/27651473/>

Proc Natl Acad Sci U S A. 2016 Sep 20;113(38):10451-2. doi: 10.1073/pnas.1612596113.

Inner Workings: Can single molecules bind together entire ecosystems?

<https://www.pnas.org/content/113/38/10451.long>

In 2013, Zimmer and his then-graduate student Ryan Ferrer borrowed a concept from ecology and dubbed such chemicals "molecules of keystone significance" (3).

<https://academic.oup.com/bioscience/article/63/6/428/225996>

We thank the Reviewer for the comment and we have changed the reference as suggested.

#) Line 361: Fig. 3: Overview on interaction-traits across genome functional clusters (GFCs). Each slice shows the interaction traits present in a GFC and, as dendrogram, the functional similarity of genetic traits of the grouped genomes. Known interactions (described in the referenced literature) between bacteria and phytoplankton are marked with arrows; a blue arrow indicates a positive interaction (an enhancing effect on phyto- plankton growth), a red arrow a negative interaction, and a blue-red arrow a positive interaction that eventually becomes negative. A dashed arrow indicates a known interaction involving a bacterial genotype with a high similarity to one of the genomes included in the analysis. Asterisks in siderophores annotation indicate the presence of the specific vibrioferrin synthetic pathway along with the secondary metabolite path- way, while asterisks in the antimicrobial resistance annotation indicate that only generic resistance traits were annotated for that genome.

The authors should clearly define "interaction-traits" or "interaction traits".

We have checked the entire text for consistency and changed everything to 'interaction traits'.

Are there any genomic traits which are not directly/indirectly involved in interactions between microbes?

"Known interactions (described in the referenced literature) between bacteria and phytoplankton are marked with arrows" -- What are the remaining interaction traits that are not marked with arrows? Are these unknown interactions?

We thank the Reviewer for this comment. That sentence in the caption of Fig. 3 (now Fig. 2) was indeed unclear and we changed it as follow: *"Distribution across genome functional clusters (GFCs) of interaction traits known to mediate cell-cell interactions in bacterial model systems. Each slice shows the interaction traits present in a GFC and, as dendrogram, the functional similarity of genetic traits between the grouped genomes (hierarchical clustering of r-correlation matrix with complete agglomeration algorithm). The dark bars show the number of interaction traits annotated in each genome. Genomes belonging to model bacteria used to discover some of these traits are highlighted in blue if the interaction was positive (e.g. enhancing phytoplankton growth), in red if it was negative (e.g. kill the host) or in grey if the interaction shifted from positive to negative."*

#) Lines 415-421: Genome functional clusters differ in their overall potential for interactions. When considering antagonistic and synergistic interaction traits together, it is clear that some GFCs encode significantly more interaction traits than others (Fig. 3), both in terms of diversity and richness (Supplementary Fig. 9a). One potential driver for the difference in richness and diversity of interaction traits could be the reduction in genome size associated with oligotrophic lineages such as pico-Cyanobacteria and ...

- I wonder how to calculate the richness and diversity of interaction traits (using the number of different interaction traits and their relative abundances to calculate Shannon entropy)?

We rephrased the caption of Supplementary Fig 9a (now Supplementary Fig 7a) for clarification: *"For each Genome functional clusters (GFCs), the box-plots show the trait richness (i.e. total number; a) and the type richness (i.e. different types, visualized as different coloured squares in Fig. 2; b) of the interaction traits annotated in the grouped genomes. [...]"*

#) Line 527-530 [*maybe 535-537?*]: A dataset of complete genomes of marine bacteria was compiled performing an extensive research on metadata available from NCBI (<http://www.ncbi.nlm.nih.gov/genome>), JGI (<https://img.jgi.doe.gov/cgi-bin/m/main.cgi?section=FindGenomes&page=genome-Search>) and MegX (<https://mb3is.megx.net/browse/genomes>) websites.

- A dataset of complete genomes of marine bacteria -> A dataset of complete and high-quality draft genomes of marine bacteria

We have changed the text accordingly.

I wonder if the authors could show the assembly status (complete or draft) and the number of replicons (chromosomes and plasmids) in the [S_tab2] tab in *6477_0_data_set_158598_qhgx7y.xlsx* file (Supplementary Table 2).

<http://www.ncbi.nlm.nih.gov/genome>

<https://img.jgi.doe.gov/cgi-bin/m/main.cgi?section=FindGenomes&page=genome-Search>

are accessible, while

<https://mb3is.megx.net/browse/genomes>

requires Username and Password to Sign in.

Information on the number of replicons is available as number of scaffold and of contigs. However, a more precise distinction about number of chromosome and plasmid/chromid is hardly achievable as some genomes, although being almost (~99%) complete, are not closed and it would be difficult to reliably classify contigs as being chromosome or mobile elements.

#) Lines: 563-567: overlap score >1 and a blast e-value <1-6 were retained. Manual annotations were performed to specifically identify production of photoactive siderophores (i.e. vibrioferrin) 18, blasting predicted protein sequences (blastp) 109 against reference dataset assembled using all available sequences of related genes (pvsABCDE operon) 110 available in UniProt.

- Line 563

blast e-value <1-6 -> blast e-value <1e-6

We have changed the text accordingly.

"Manual annotations" performed here should be explained. Functional annotations from the most similar (BLAST best hit) sequences were assigned and edited manually?

We have rephrased the sentence accordingly.

Could the authors provide a list of UniProt accessions for the reference dataset of pvsABCDE operon?

We provided, as Supplementary file 3, a fasta file containing all the sequences of the assembled reference dataset.

#) Lines 604-625: The presence/absence matrix of genetic traits in genomes served as basis to cluster the former into linked trait clusters (LTCs) and the latter into genome functional clusters (GFCs). The GFCs were generated feeding a genome functional similarity matrix calculated using Phi coefficient (i.e. Pearson correlation for binary variables; phi function, package sjstats) into the affinity propagation algorithm implemented in the apcluster function ($q=0.5$ and $\text{lam}=0.5$; r package apcluster) ¹¹². This machine learning algorithm was chosen because it does not require the number of clusters to be determined a priori, allowing instead this feature to emerge from the data ¹¹³. Briefly, the functional similarity matrix is used to construct a network where nodes and edges are known to be genomes and their pairwise Phi correlation, respectively. Starting from a random set of exemplar nodes, clusters are created by expansion towards the adjoining, most similar, nodes. Through iterations of this procedure, the algorithm tries to maximize the total similarity between nodes within each cluster eventually converging towards the best set of clusters.

For the LTC delineation, a similarity matrix of genetic traits was built calculating the square of Pearson's correlation coefficient (r^2) ¹¹⁴. Out of a total of 556 genetic traits, 434 were retained for downstream analysis as they were found in >3% of the genomes (>14 genomes) and r^2 was computed using the ld function (r package snpStats) ¹¹⁵. The LTCs were automatically extracted from the hierarchical clustering (hclust function in r, method = "ward.D2") of the dissimilarity matrix ($1-r^2$) using the function cutreeDynamic (method = "hybrid", deepSplit = 4 and minClusterSize = 3; r package dynamicTreeCut) ¹¹⁶. ...

- I wonder why the authors used different similarity/dissimilarity metric (phi and $1-r^2$) and clustering algorithm (apcluster and hclust) for GFC and LTC delineation. Have the authors checked the robustness of the results; i.e. whether the conclusions remained similar, regardless of the distance and clustering methods used?

We thank the Reviewer for this remark. We have run both the GFC and LTC clustering using multiple correlation indexes (e.g. Pearson, Jaccard, Sorensen, Whittaker) and clustering agglomeration algorithms (affinity propagation, complete, ward D2). These results of all clustering showed high cophenetic correlations (>0.65 for GFCs and >0.30 for LTCs; LTC correlation is lower as the Pearson, Jaccard, Sorensen, Whittaker indexes include also negative component of traits correlation, more details in the answer below) with the main clustering (i.e. Pearson and affinity propagation) meaning that all dendrograms were very similar and they were recapitulating the main patterns of GFCs and LTCs. This method part has been updated and now the same correlation coefficient (i.e. Pearson's correlation coefficient r) and clustering methods (i.e. affinity propagation) are used for both genomes and genetic traits. A sentence describing the robustness of the clustering results has been added to the material and method section (L544-548), as well as a detailed discussion in the Supplementary text (L338-360 and Supplementary Fig. 15c-d).

Both of the row dendrogram and the column dendrogram in "Fig. 1: Atlas of Marine Microbial Functional Traits" were computed by the same clustering algorithm; e.g. hierarchical clustering (hclust function in r, method = "ward.D2") of the dissimilarity matrix ($1-r^2$)?

As requested also by Reviewer 1, we improved the caption of Fig. 1 indicating specifically how rows

and columns have been clustered.

I wonder why the authors performed the hierarchical clustering on "the dissimilarity matrix (1-r²)" (one minus the square of Pearson's correlation coefficient) ranging from 0 to 1, instead of one minus the Pearson's correlation coefficient (1-r) ranging from 0 to 2.

We thank the Reviewer for the comment as it allows us to improve our analysis. In the updated workflow we use the Pearson's correlation coefficient r and not on its square (r^2).

Initially, we used r^2 to allow a direct construction of the clusters (LTC) with the affinity propagation algorithm, as the algorithm requires a similarity matrix in input (so no negative values). By doing so, however, the trait clusters were formed based on both positive and negative correlations (the figure below shows that negative correlations were weaker than positive ones but still frequent). However, our working hypothesis focuses on detecting the functional linkage between traits and not their mutual exclusion. Therefore, in the updated analysis based on the r coefficient, we have solved the issue by making all negative r values equal to 0, as explained in the Methods section.

Could the authors provide all the scripts/codes used for "Statistical analyses"?

We created a cleaned and detailed R script including the code for statistical analysis of GFCs and LTCs. The script is available in the new repository https://github.com/lucaz88/genome_comparison_code.

Reviewer #3 (Remarks to the Author):

Comments to Authors:

The present study entitled "Comparative whole-genome approach to identify traits underlying microbial interactions" by Zoccarato et al., shows the results of bioinformatic analyses applied to 473

genomes previously isolated from different marine ecosystems to identify their putative functional genetic and interaction traits. The Authors conclude that their study could be a basic reference for investigating clusters of functional and linked traits to better understand fundamental rules across biomes.

Overall assessment:

Investigations on microbial interactions with marine organisms and their functional diversity in marine ecosystems are timely and are relevant in ecological studies. The ms is well written, with very appealing figures. The methodologies are clearly described. The authors made a number of elaborations to justify their conclusions, but I have some major concerns with this study.

We thank the Reviewer for the appreciation of our work, and have addressed the remarks as described below.

First of all, the manuscript doesn't meet the expectations created by the Title and the Abstract. I found that the authors in the Results and Discussion of the ms, provided a technical report of the results obtained by their bioinformatic approach without providing evidence of new findings shading light on ecological microbial interactions, but just leaving the reader with the possibility of expanding the present knowledge in future papers. I think that a reader of Communications Biology wants to understand what has been discovered and how it advances our understanding in the field, not just reading a bioinformatic exercise.

We thank the referee for this critical remark, which has promoted us to significantly modify the flow of the manuscript. For each paragraph, we now make sure to explain the rationale or describing the research question before describing the results. We also clarify what findings are new, and how they shed new light on the addressed research topics. Below are two of many examples:

- 1) In a new section describing to what extent the selected genomes represent the diversity in natural marine ecosystems, we first present the question: (L161) *"To what extent do the GFCs represent the diversity of natural communities, or display dynamics expected from ecological units in the oceans?"*. We then describe the analysis, and we end with the implications of the new findings: (L165) *"Thus, despite the inherent bias derived from using only high-quality, closed genomes available (mainly) from cultured bacteria (legend of Figure 1), the GFCs represented a significant fraction of bacterial diversity."*, and (L173) *"Assuming that similar temporal trends suggest similar ecological niches, these results suggest that (at least some of) the GFCs represent bona-fide ecological units."*
- 2) In the section describing enrichment for "interactions traits", we again pose a question (L178): *"We next focused on selected traits potentially involved in microbial interactions – vitamin exchange, siderophore and phytohormone production and antibiosis – asking whether we could observe patterns in their distribution across the genome dataset."*. In the extended section that describes these results, the importance of each finding is clearly stated, as are some specific hypotheses for future testing. For example: (L198) *"Less than half of all the genomes in our dataset were predicted to produce all three vitamins [...]"*; (L202) *"This suggests that there is a major "market" for B vitamins, and indeed almost all genomes (~83%) encoded transporters for at least one vitamin."*; (L227) *"Therefore, we hypothesized that vitamin B₁ supplies might be more stable or frequent (e.g. as a result of higher export or lysis of producing bacteria) than that of vitamin B₁₂."*

The authors commented on the specific bacterial clusters identified and their interactions in light of the available literature (not exhaustively reported), and concluded what is obvious (e.g., “we propose that organisms belonging to the same GFC are likely to interact in similar ways with other microorganisms”; “phylogenetic affiliation of bacteria can predict the traits encoded in their genomes”; “important role of vitamins in shaping marine microbial communities”; “the production of phytohormones may be linked with other interaction traits”). At the end of the manuscript, I couldn't find a take-home message or a new finding.

We respectfully disagree with the Reviewer that these results are obvious or trivial:

1) As remarked also by Reviewer #2, the idea that “we propose that organisms belonging to the same GFC are likely to interact in similar ways with other microorganisms” is far from being obvious. To the best of our knowledge, there are few studies that systemically analyzed microbial interactions across diversity, and which include multiple strains belonging to the same clade (e.g. Vibrionaceae clustered in GFCs 25 and 47). When such studies have been performed, antagonistic interactions were conserved within ecologically coherent units (Cordero et al, Science, 2012), but the ability to inhibit multiple target organisms (“super-killers”) was limited to a much smaller subset of strains that were broadly distributed phylogenetically (Cordero et al, Science 2012; Russel et al., PNAS, 2017). We now add this to the conclusions section (L414): *“In support of the GFC concept, a study of marine Vibrionaceae suggested that ecologically-cohesive unit (similar to GFCs) are likely to interact in a similar way²⁹. However, some interaction phenotypes, such as the ability to inhibit multiple target bacteria (“super-killers”), are not phylogenetically conserved^{29,113}.”*

2) “phylogenetic affiliation of bacteria can predict the traits encoded in their genomes” while there are reasons to expect such correlation, there are also several studies that question this statement (e.g. Newtow et al., 2010, ISME J). This is now better represented in the text: L139: *“Overall, half of the detected GFCs (i.e. monophyletic) support the existence of a strong correlation between taxonomy and functionality in marine bacteria, while the remaining paraphyletic GFCs highlight that, in some cases, the taxonomic partitioning (based on the Genome Database Taxonomy⁴⁹) do not completely reflect the functional differentiation.”*

3) “important role of vitamins in shaping marine microbial communities”. While the importance of vitamins in microbial interactions has indeed been recognized (and as we mention repeatedly in the text), to the best of our knowledge a holistic view of multiple vitamins and their potential roles in mediating interactions is still lacking. This is where our analysis provides a first comprehensive overview of the B vitamins’ market for marine environments. We also provide a conceptual framework for analyzing this “vitamin market”, by defining specific roles (e.g. consumer, independent, flexible/source) and identifying which bacteria (taxon and GFC) has the capability to perform which of such roles.

4) “the production of phytohormones may be linked with other interaction traits”. This, perhaps, represents best the novelty of our results. While several recent studies have described additional facets of the phytohormone-mediated interactions, these were limited to a small number of models (e.g. *Dinoroseobacter*, *Sulfitobacter* and *Phaeobacter*; Amin et al., 2015, Nature; Segev et al., 2016, eLife; Seyedsayamdost et al., 2011, Nature Chemistry; Wienhausen et al., 2017, Front Microbiol). In these studies, interactions mediated by phytohormones also included the exchange of taurine and tryptophan. We now extent this observation, and add additional pathways for quorum sensing, capsular polysaccharide and putrescine. We specifically described the linked traits and potential synergism

in the description of the LTC 19 (L368-379).

Another critical issue of the study is the reliability of the approach used. The same authors highlight (for example, lack of representativeness of microbial genomes, statistical nature of presumed functional traits) technical limitations in some parts of the study, conclusions included. I appreciated their honesty, but since this scientific article is primarily based on the development of a bioinformatics approach, the lack of evidence of its performance and of trust of the authors themselves undermine the findings of the paper. I do not want to belittle the work behind this metadata analysis, but I think that the results of this work are not sufficiently robust to be published in Communications Biology.

We thank the Reviewer for this candid comment. In the revised manuscript, we present several tests and analyses that establish the reliability and robustness of the results. Specifically:

- 1) We show that the set of analyzed high-quality genomes represent a significant fraction of the bacterial community (16S reads) in two contrasting marine ecosystems, one coastal and one pelagic/oligotrophic (mean 22.9% and 13.9%, respectively, range 0.5-60%). This is described in L160-175 (see also detailed response, below).
- 2) We present detailed benchmarks of the bioinformatics pipeline, including sensitivity analyses of key parameters (e.g. the threshold for trait completeness, the 'q-value' controlling clustering sensitivity of affinity propagation) and accuracy analyses of the clustering method (e.g. genome down-sampling, different clustering algorithms). This is now presented in a specific section of the supplementary information (L303-360, see also detailed responses to referee #2). We also performed a detailed re-evaluation of the code, and tested its utility and reproducibility with multiple users. These analyses show that our results are analytically robust and thus reliable.
- 3) We show that, in the coastal ecosystem (which has a detailed time-series), individual 16S phylotypes that belong to the same GFC exhibit similar temporal trends (L171-175, Suppl. Fig 6). In other words, the GFCs behave as would be expected from ecologically relevant units.

These changes (and others) establish and emphasize the reliability and robustness of our results.

With regard to stating the limitations of our study, we strongly believe in transparency (as appreciated by the referee), which is why we organized all the major caveats of our approach in the 'Summary and outlook' section (former 'Conclusions') where we thoughtfully discuss them. In this regard, while we agree that the LTCs are statistical in nature, and acknowledge this clearly, we do not think this is a weakness of the manuscript – statistical clustering of entities such as genes, transcripts, proteins or metabolites is fundamental in any analysis which aims to make sense of complex data (e.g. any pathway or function enrichment test). These statistics, as employed in the delineation of the LTCs, enable us to identify previously unknown correlations between individual traits. This suggests new mechanisms of microbial interactions, which can now be tested experimentally.

Together with the changes we have made to the manuscript to better represent the rationale, main results and "take home messages", and the clarifications mentioned above that show that these results are far from trivial, we believe the revised manuscript is much more than the description of a new bioinformatics approach. Our results provide not only a framework to classify marine bacteria and their traits, but they also highlight novel patterns allowing us to infer and predict how bacteria interact across diversity, and to suggest novel interaction mechanisms.

Additional comments

Title and Abstract

The Title and Abstract are misleading, i.e., are not consistent with the results and discussions. Microbes do not include only heterotrophic bacteria, but also chemotrophs, archaea, fungi and, also viruses (despite this latter component cannot be considered as truly living organisms).

We changed the title of the paper to more specifically reflect our analysis.

“Comparative whole-genome approach to identify bacterial traits for microbial interactions”.

In the Abstract, the Authors mentioned bacterial “ecotypes” (line 35), new ecological units and bacterial lifestyles (line 40), so I expected that they addressed their results highlighting distinct biogeographic populations adapted to specific environmental conditions and identified new populations (since in ecology an ecological unit is a population, a community or an ecosystem). Reading the Results and Discussion, I realized that they referred just in some cases to the habitats but without any in-depth discussion.

We thank the Reviewer for the comment and we rephrased the two sentences in the abstract. We did not attempt to correlate GFCs with specific environments as this is not the main focus of the manuscript (and there are not enough closed or highly complete genomes from marine environment to perform such analysis yet).

L27 “[...] we identified genome functional clusters (GFCs) which allow to classify bacterial diversity into groups with potentially common ecology and life history.”

L34 “Moreover, linked trait clusters (LTCs) identify traits that may have evolved together and point to specific modes of interactions.”

Lines 43-46. How the Authors can demonstrate and validate the efficiency of their approach?

As answered above to Reviewer #2, we carried out several tests to provide validation/sensitivity analyses for the most critical parameters and methods adopted in our workflow (supplementary text L303-360 and Supplementary Fig. 15). In the revised manuscript, we demonstrate that the presented functional patterns are relevant for natural marine bacteria by quantifying the fraction of natural bacterial communities represented in the genome dataset and by showing that the dynamics of GFCs in a natural coastal ecosystem are consistent with those expected from ecological units. These results are presented in L160-175 and detailed in the response to Reviewer #1. Overall, we show the relevance of the detected functional patterns for coastal and pelagic bacterial communities, and we support the validity of the GFC concept of clustering bacteria into groups with coherent functional profiles.

We have also rephrased the last sentence of the abstract as follow: *“Our approach translates multidimensional genomic information into an atlas of marine bacteria and their ecosystem functions, relevant for understanding the fundamental rules that govern community assembly and dynamics.”*

Introduction

The introduction provides only part of the information needed on the interactions between phytoplankton and marine heterotrophic bacteria.

Line 54. This statement is too general and vague and I think that interactions are not “a feature”.

Line 56. In the oceans, heterotrophic bacteria interact not only with phytoplankton. They have key roles in biodiversity and functioning of marine ecosystems, and their interactions with all trophic levels, including viruses, have been published in hundreds of studies.

I think that if the Authors' intention was to focus their study on the interactions between heterotrophic bacteria and phytoplankton they should have specified it at the beginning of the paper.

We carefully rephrased the beginning of the introduction adding missing references that acknowledge also for interactions with virus and protists. We also clarified that the intention of the manuscript is to focus on bacteria-bacteria and bacteria-phytoplankton interactions.

L46-49: *“Interactions among aquatic microorganisms such as symbiosis, parasitism, predation and competition, greatly shape the composition and activity of microbial communities¹⁻³. In particular, interactions between heterotrophic bacteria and primary producers (phytoplankton) influence the growth of both organisms^{4,5} with consequences for the ecosystem functioning and the biogeochemical cycles^{6,7}.”*

Lines 101-107. These statements are so approximate and general that they make it difficult to understand the new findings of this work beyond the bioinformatics method used.

We completely re-wrote the last paragraph of the introduction to provide a more specific and targeted overview of the analyses with a clear reference to some of the major outcomes.

L84-93: *“We focused on the following traits: 1) the metabolic capacity to synthesize or degrade specific biomolecules and to regulate such processes (KEGG modules); 2) specific functions related to microbial interactions, such as motility, chemotaxis and the ability to produce molecules such as siderophores, toxins and antibiotics. This allows to classify genomes into coherent functional units, some of which recapitulate known bacterial groups with well-defined ecological roles, while others refer to potential new groups. Furthermore, genetic traits can be grouped into linked trait clusters, representing functions that likely evolved together and may be functionally connected (i.e. participating in the same process). Our approach maps the mechanisms of microbial interactions elucidated in model organisms across bacterial diversity, suggests specific groups of bacteria likely to interact in a similar manner, and identifies trait combinations that likely mediate these interactions.”*

Results and Discussion

Line 113. A question is: How and why the Authors selected 473 genomes? Are these sufficient to make broad-spectrum predictions? Frankly, I think that among NCBI, JGI and other databases, even considering only “super-clean” genomes from marine environment, more than 473 genomes can be used. So, I wonder if this subsample of genomes was created to match the processing capabilities of their data set (which is actually complex) rather than because they are truly representative of marine prokaryotic diversity.

The list of 473 genomes is comprehensive, meaning that (at the time of analysis) there were no more

“super-clean” genomes available in the NCBI, JGI and MegX databases. There was no “subsampling” of genomes to match the processing capabilities. This is now clarified in L435: *“The final list of 473 genomes includes all of the genomes that, using CheckM¹¹⁴, were defined “closed” (i.e. each DNA molecule, such as chromosome and plasmids, was represented as a single sequence in the fasta file) or high-quality draft genomes (>90% of completeness, <10% of contaminations, >18 tRNA genes and all 3 rDNAs present)¹¹⁵. ”*.

With regard to whether these genomes are sufficient to make broad-spectrum predictions, we have added to the manuscript a section demonstrating that these genomes represent a significant portion of two natural bacterial communities: L161-170: *“To what extent do the GFCs represent the diversity of natural communities, or display dynamics expected from ecological units in the oceans? To test this, we mapped the 16S rDNA reads from a natural coastal community that was sampled at high temporal frequency⁵⁸ to the GFCs (Supplementary text and Supplementary Fig 4). Firstly, a significant fraction of the natural community was represented with high fidelity in the GFCs (mean 22.9% of the 16S rDNA reads, range 12.7-44.3%). Thus, despite the inherent bias derived from using only high-quality, closed genomes available (mainly) from cultured bacteria (legend of Figure 1), the GFCs represented a significant fraction of bacterial diversity. Similar results were obtained with an open-ocean community from the Eastern Mediterranean sampled each season for 3 years⁵⁹ (mean 13.9% of the 16S rDNA reads, range 0.5-60.0%), where the GFCs represented a significantly higher fraction of the microbial community on particles >11 µm compared to free-living bacteria (5-0.22 µm; Supplementary Fig. 5).”*

Lines 151-154 Authors should highlight the novelty of the conclusions of their work, when compared to available literature (e.g., Louca et al. 2016, Science, Louca et al. 2015 NEE).

As suggested by Reviewer #2, we performed an extensive literature review of papers dealing with genome comparison, functional clustering and linkage between taxonomy and functionality. We now present this in the main manuscript, as described in detail below. We also summarized this review in the new Supplementary table 1 and in a specific section of the supplementary text (L19-70).

In the introduction, we describe the available literature, pointing the referee to the relevant sections of the supplementary information, and describing how our approach differs from previous studies (L78-84): *“Several previous studies have aimed to characterize and cluster genomes based on their predicted functional similarity, defined usually using individual genes (e.g.³⁴⁻³⁷) or coarse functional categories (e.g. COGs,^{36,38,39})(Supplementary Table 1, Supplementary text). We chose to take a trait-based approach rather than a gene-based one, which is an intermediate level of resolution between individual genes and coarse functions. Trait-based approaches offer a new perspective to investigating microbial diversity with a more mechanistic understanding⁴⁰ but have been used only in few specific cases to highlight putative bacterial interactions (e.g.,⁴¹).”*

In the results section, we compare our results to previous studies, highlighting the identification of differences between closely-related clades that nevertheless belong to different GFCs (L108-125): *“Previous genome comparison approaches have identified genome clusters that match to ecologically relevant groups (e.g. ecotypes, as defined for *Bacillus pumilus*³⁵ and *Prochlorococcus*²⁸) or lifestyles (e.g. oligrotropic and copiotrophic species³⁸). Similarly, in our analysis we found GFCs that represent group of organisms with a defined ecology and life history, such as the Pelagibacterales group (GFC 2), different ecotypes of Cyanobacteria (GFCs 15 and 36), or *Vibrio* groups, characterized by different host-specificity and pathogenicity (GFCs 25 and 47)(see Supplementary text). Specific GFCs were also identi-*

fied for each of three groups of Gammaproteobacteria (*Alteromonas*, *Marinobacter* and *Pseudoalteromonas*) which are typically considered as “copiotrophs”, often associated with organic particles or phytoplankton⁴³⁻⁴⁶. A detailed analysis of the traits found in each of the respective GFCs (Supplementary Fig. 2) suggested that *Pseudoalteromonas* and *Alteromonas* bore more genetic traits involved in the resistance against antimicrobial compounds, as well as regulation for osmotic and redox stresses. They also had similar vitamin B1 and siderophore transporters, which are different from those encoded by *Marinobacter*. *Marinobacter* possessed several more transporters for phosphonate and amino acids, as well as specific regulatory systems for adhesion (e.g. alginate and type 4 fimbriae production) and chemotaxis. These patterns advocated that there might be coherent physiological and/or ecological differences between these three groups. Overall, our GFC framework recapitulates previous knowledge on bacterial groups with defined ecology and life history, and provides a way to delineate and characterize potentially new ecological groups.”.

Finally, we specifically address the correlation between function and taxonomy including the relevant literature (L139-156): “Overall, half of the detected GFCs (i.e. monophyletic) support the existence of a strong correlation between taxonomy and functionality in marine bacteria, while the remaining paraphyletic GFCs highlight that, in some cases, the taxonomic partitioning (based on the Genome Database Taxonomy⁴⁹) do not completely reflect the functional differentiation. At the whole-community level, it has been shown that taxonomically distinct communities exhibited similar functional profiles, which led to the suggestion that some bacterial clades have similar genetic capacity, and can replace each other while maintaining unchanged the community functioning^{50,51}. The polyphyletic GFCs may group such taxonomically different but functionally similar organisms, and this is supported by examples in GFCs 33 and 41 (grouping thermo- and halotolerant bacteria), or GFC 17 (grouping sulfur-oxidizing and facultative anaerobe bacteria). One of the main biological processes that mediates such functional “homogenisation” is the horizontal gene transfer^{52,53} and, indeed, most of the paraphyletic GFCs grouped known copiotrophs or particle/host associated bacteria (e.g. *Rhodobacteraceae* in GFCs 9, 30 and 40, *Vibrionaceae* in GFCs 25 and 47, *Alteromonadaceae* in GFCs 21 and 24, *Oleiphilaceae* in GFC 35, or *Halomonadaceae* in GFC 43;^{12,17,44,54,55}) whose lifestyles favour events of genetic exchange (e.g.⁵⁶). Contrarily, another study suggested that this perceived similarity in community function reflects only known metabolic pathways, and it is therefore possible that adding to our analysis also unknown genes might separate these GFCs into monophyletic ones⁵⁷. Although hypothetical genes would be of no use (not informative) in a trait-based approach, this hypothesis could complement the horizontal gene transfer hypothesis in explaining the blurred taxonomic profiles of the paraphyletic GFCs.”.

Lines 175-177. In my opinion, the reliability of this approach cannot be demonstrated through the correspondence of GFCs and previous literature information on bacterial species. The use of the term “ecotype” is incorrect because ecotype doesn’t mean that some species have similar lifestyles.

Our goal in this section was to show that the division of the GFCs corresponds to known functional divisions of microbial diversity. The ecotype definition we use here is that of “Genetically and physiologically differentiated subgroups of a species that occupy a distinct ecological niche” e.g. as used for *Bacillus pumilus* or for *Prochlorococcus*. While we are aware of the ongoing conversation regarding the best definition for ecotypes, we prefer to stick with this commonly-used definition, and now clarify this in the text (L108): “Previous genome comparison approaches have identified genome clusters that match to ecologically relevant groups (e.g. ecotypes, as defined for *Bacillus pumilus*³⁵ and *Prochlorococcus*²⁸) or lifestyles (e.g. oligotrophic and copiotrophic species³⁸).”.

Having clarified the definition of ecotype, we argue that the similarity of GFCs to known bacterial groups with defined ecology and life history can be used as partial support for the reliability of the approach. Nevertheless, following the referee's suggestion, we have shortened this section (moved part of it to the supplementary information), and we now strengthen the case for the reliability of the GFCs in two additional ways, as described also above:

- 1) We provide evidence that the division into ecotypes is robust to the methodology used. This is described in detail, above, and in response to referee #2.
- 2) We show that, in a coastal time-series, 16S phylotypes corresponding to the same GFC display similar temporal dynamics, which suggests that they are ecologically coherent units. This is now described in L171-175: "*Secondly, using a temporal deconvolution analysis of the coastal site⁵⁸ we found that individual 16S phylotypes belonging to the same GFC displayed significantly more synchronous temporal trends (p-value < 0.001) than phylotypes belonging to different GFCs (Supplementary text and Supplementary Fig. 6). Assuming that similar temporal trends suggest similar ecological niches, these results suggest that (at least some of) the GFCs represent bona-fide ecological units.*"

Lines 163-188. All this part is a description of literature information.

We are unclear as to what the referee meant here, as lines 163-188 described the link between specific GFCs and known bacterial groups. Regardless, we have streamlined the text by moving some of the detailed comparison between some GFCs and previous literature to the supplementary information, giving more space for the discussion of the novel findings.

L111-114: "*Similarly, in our analysis we found GFCs that represent group of organisms with a defined ecology and life history, such as the Pelagibacterales group (GFC 2), different ecotypes of Cyanobacteria (GFCs 15 and 36), or Vibrio groups, characterized by different host-specificity and pathogenicity (GFCs 25 and 47)(see Supplementary text). Specific GFCs were also identified for each of three groups of Gammaproteobacteria (Alteromonas, Marinobacter and Pseudoalteromonas) which are [...]*"

Lines 189-191. This statement is approximate and vague.

We agree that the sentence was not clear enough. We have removed it and replaced it with two sentences summarizing the GFC approach.

L122-125: "*These patterns advocated that there might be coherent physiological and/or ecological differences between these three groups. Overall, our GFC framework recapitulates previous knowledge on bacterial groups with defined ecology and life history, and provides a way to delineate and characterize potentially new ecological groups.*"

Lines 191-192. Please, clarify.

This sentence was aimed to describe a limitation of the GFC analysis, in that it does not reproduce some aspects of high-resolution functional differentiation between closely related bacteria, e.g. between specific high-light ecotypes in *Prochlorococcus* (which share the "high light" surface niche but vary in their temperature or nutrient optima) or between different species of *Alteromonas*, that are

also supposed to inhabit slightly different niches. We now describe this clearly in the supporting information section dealing with the GFCs and taxonomy (L118-122).

Fig.1. Can the absence of genetic traits in most of the genomes analysed affect the reliability of the approach? And, how much? Please, clarify.

We thoroughly discuss some of the major absence patterns of genetic traits in the “genome atlas” showing how they reflect known biological features of certain taxonomic groups, e.g. the broken TCA cycle in pico-cyanobacteria and some heterotrophic groups, or the lack of functional traits involved in the cell wall assembly reflecting the general distinction between gram positive and gram negative bacteria (these cases are thoroughly discussed in the Supplementary text, L256-300 and Supplementary Fig. 14). Nevertheless, we cannot fully rule out that the absence of less common genetic traits are due to possible limitations of the functional annotation process, as stated at L405-407: “[...] on average, only 63% of genes were functionally annotated across genomes (range ~40-80%), with the remaining genes either annotated as hypothetical or not annotated at all. This is a strong reminder of the limitations of current genomic and metabolic knowledge.”

Lines 216-217. The lack of confidence in this sentence weakens the study because the message that comes to the reader is that this approach is not so reliable.

As described above, we now provide additional support for the reliability of the LTC approach. Nevertheless, in order to maintain transparency, we clearly describe the statistical nature of this concept in the closing sentences of the manuscript, as well as reiterate where this concept provides important new information (L412-424): *“Finally, the GFC and LTC concepts are both statistical in nature, representing the probability of bacteria having similar functional capacity, and of traits being functionally and/or evolutionarily linked. In support of the GFC concept, a study of marine Vibrionaceae suggested that ecologically-cohesive unit (similar to GFCs) are likely to interact in a similar way²⁹. However, some interaction phenotypes, such as the ability to inhibit multiple target bacteria (“super-killers”), are not phylogenetically conserved^{29,113}. Experimental studies using both established and new model systems across multiple scales of diversity (e.g. between and within GFCs) are needed to test the GFC framework, whereas genetic manipulation of linked traits to test their effect on microbial interactions will be required to determine to what extent traits within LTCs are functionally linked. Nevertheless, GFCs and LTCs describe how putative interaction traits vary across microbial diversity, and thus can be used to quantify how the fraction of natural communities potentially “ready to interact” changes over space and time. In turn, this should help to elucidate fundamental rules that govern community dynamics and assembly in the oceans, and the roles played by microbial interactions will in global ecosystem-level processes and biogeochemistry.”*

Lines 233-236. Which is the novelty compared to the previous literature?

While the lack of a full TCA cycle had been described in cyanobacteria, as well as in several heterotrophic bacteria, our analysis extends these observation to specific GFCs. This is now clarified in the supplementary information, L277-281: *“In addition, our results showed that other heterotrophic bacteria lacked several or almost all of the reaction of this pathway. To the first case belonged the genomes of Spirochaetota and Thermotogota, (grouped in GFCs 4 and 39, respectively) while the latter case*

comprised several Firmicutes (GFCs 44 and 45) and the other Thermotogota (GFC 18) genomes. These findings broaden what was reported in previous studies which were focused on single species belonging to the mentioned taxa⁴⁶⁻⁴⁹. ”

Lines 239-241. And what does this imply?

Please see our remarks above regarding how we demonstrate the reliability of the LTC concept, and how we refer to the statistical nature of this concept.

Lines 244-246. This is a further confirmation that the approach requires validation, as reported by the authors themselves.

As suggested by Reviewer #1, the patterns highlighted by the core LTCs (e.g. synthesis of DNA, RNA, and amino acids, essential traits common to any cell), as well as, the absence pattern of some common LTC (broken TCA cycle and lack of traits involved in the cell wall assembly in gram negative bacteria), represent a first validation of our approach. Therefore we have changed the sentence as (L330-338) *“Two LTCs, 3 and 5 (mean r of 0.30 and 0.35), occurred in nearly all genomes (>93%; Fig. 1) and, as expected, they linked traits which mediate for core metabolic functions, common to any cell. These include biosynthesis of nucleotide (DNA and RNA) and amino acids, as well as core metabolic pathways (glycolysis, pentose phosphate pathway and the first three reactions of the TCA cycle). In contrast, other LTCs (i.e. 2, 4 and 7) highlighted cases in which major metabolic pathways such as the TCA cycle or pathways for the cell wall assembly were missing in specific bacterial clades. These absence-patterns were consistent with previous studies (see Supplementary Text for detailed description). The LTC concept can, therefore, be used to identify traits that may function together, providing hypotheses as to new modes of interaction that can be tested experimentally.”*

Nonetheless, being an bioinformatic approach we also remarked, in the outlook section, that (L417-422) *“Experimental studies using both established and new model systems across multiple scales of diversity (e.g. between and within GFCs) are needed to test the GFC framework, whereas genetic manipulation of linked traits to test their effect on microbial interactions will be required to determine to what extent traits within LTCs are functionally linked. Nevertheless, GFCs and LTCs describe how putative interaction traits vary across microbial diversity, and thus can be used to quantify how the fraction of natural communities potentially “ready to interact” changes over space and time.”*

Lines 249-255. These parts, such as in other paragraphs, should be avoided in the discussion because they are more appropriate for an Introduction.

We have modified those lines and just two sentences remained to set the stage for the topic of the manuscript’s section: (L195-198) *“Vitamins B₁, B₇ and B₁₂ are essential cofactors for microbes. Some microorganisms (including abundant phytoplankton) are auxotrophic for these vitamins and need to obtain them from co-occurring bacteria^{62,63}. Vitamins are found at low concentrations in aquatic ecosystems^{64,65} and their supply can limit biogeochemical cycles, e.g. through limiting primary productivity in the Southern Ocean⁶⁶.”*

Lines 313-314. I’m sorry but I cannot understand the novelty of this conclusion.

We thank the Reviewer for the criticisms. We have rephrased the sentence to clearly state how our analysis contributes to the general understanding of vitamin role for aquatic microbes. L231-233 *“Taken together, these data provide a comprehensive overview of the market for B vitamins in marine environments by defining specific roles (e.g. consumer, independent, flexible/source) and identifying which bacteria (taxon and GFC) fulfil each role.”*

Lines 320-326. This part is a description of literature information.

Same as above, we revised the text and only one sentence were kept to introducing the topic of this section: (L238-239) *“The production and exchange of “common goods” such as siderophores⁷¹, as well as of specific phytohormones like auxin⁷² represent traits that may mediate synergistic microbial interactions.”*

Lines 336-339. I cannot understand: if in this study the authors highlight the role of siderophores as “keystone molecules” but cite a previous study, this means that it was not a new finding. The same for following sentences.

The citation is for the concept of a “keystone molecule”, (in this case, Tetrodotoxin), and not for the role of siderophores. We rephrased the related sentences to better highlight the novelty of our siderophore analysis: (L247-252) *“In this regard, 5% of the genomes encoded the capacity to produce vibrioferrin (Supplementary Fig. 11), which is available to a wide range of organisms upon photolysis²⁰. Field studies revealed that siderophore biosynthesis is widespread in the ocean⁷⁵, and that bacteria producing e.g. vibrioferrin can represent a relevant percentage of the total bacterial communities⁷⁶. Thus, siderophores can be considered as “keystone molecules” (sensu⁷⁷), produced by a limited subset of organisms but utilizable by a wide range of bacteria⁷⁸⁻⁸⁰.”*

Line 366. Why is this “perhaps surprising”?

We removed it.

Lines 412-413. in this paragraph the authors did not draw main conclusions.

Based partly on this remark, we have significantly revised the sections dealing with antagonistic interactions and how they are observed in the LTCs:

L289-294: *“All GFCs which grouped genomes of Cyanobacteria, Actinobacteriota, and Bacteroidota possessed only these “generic” resistance traits (Supplementary Fig. 12), suggesting that such clades are less efficient in resisting microbial “chemical warfare”. In support of this hypothesis, some Cyanobacteria strains are indeed used as markers for antibiotic contamination because of their sensitivity (e.g., references^{90,91}), and Bacteroidota are often inhibited when co-cultured with other bacteria that express antagonistic behaviour^{32,34}.”*

L309-312: *“Type III secretion systems (T3SS), which deliver effector molecules that maintain the bacterial association with the host⁹⁵, were found only in a few genomes as the Vibrio clustered in GFC 25. This GFC grouped known zooplanktonic hosts⁵⁵, suggesting a more specific role for T3SS in metazoan host-microbe interactions.”*

We have also made sure to clarify the conclusions from each point, for example:

L353-356: *“Interestingly, the other two secretion systems, T4SS and T3SS, were also linked with the regulation of nitrogen metabolism and with vitamin B7 or B12 transport as part of LTC 25 and LTC 17, respectively. We propose that this provides further support for a putative functional link between microbial interactions (injection of effector molecules), nitrogen and vitamin metabolism.”*

Lines 457-458. This is a finding too much approximate.

We removed the sentence.

Lines 477. I agree that the information obtained from this study may provide insights for experimental studies, but once again Authors have not explained the ecological (or biogeochemical) implications of their results, which remain purely descriptive.

We removed the sentence and, as described above, we provide a new analysis showing that the analyzed genomes represent a significant fraction of marine bacteria diversity (L161-170). We also validate the concept at the basis of the GFC framework (i.e. grouping genomes with coherent functional profiles) further supporting the relevance of the functional patterns described (L171-175). Moreover, we provide a more clear description on how the presence of specific LTCs can have ecological and biogeochemical relevant implications. For examples: (L377-379) *“We propose that possessing this LTC [19] indicates that the relevant bacterium is capable of affecting the growth of phytoplankton through a combination of specific hormone signaling pathways and metabolic interactions.”*

Figure 4. I struggle to understand what the authors are referring to when they write "a hypothetical bacterial cell" considering that a wide variety of bacterial cell types exist.

We rephrased the caption of the figure: *“Conceptual representation of the predicted interaction dynamics for an hypothetical bacterium analysed with our trait-based approach. The bacterium is assigned to a GFC, visualized here as a jigsaw puzzle. Every puzzle piece represents one of the linked trait clusters (LTCs) possessed in that GFC. LTCs 3 and 5 (marked in red) are the core LTCs and they are present in any GFC. In this example, the GFC also possess the ancillary LTCs 11, 17 and 19. Each [...]”*

Conclusions

Lines 481-483. Not sure that GFCs and LTCs can be defined as concepts. However, this sentence is unclear.

Lines 486-489. This sentence is vague and too much general.

Lines 490-496. Here you did not highlight the new findings but just explained what you did. I agree that this approach could be a useful approach to study microbial functional traits in other studies but what about your investigation?

We thoroughly revised the conclusions of our manuscript (now called ‘Summary and outlook’) and we now present a concise overview of the methodological framework and of the related new findings: (L383-403) *“We present a framework that extrapolates from studies of specific model organisms to predict the interaction potential of other bacteria based on the traits encoded in their genomes. By*

focusing on biologically relevant traits (including specific interaction traits), we reduced a highly complex genomic dataset to a tractable matrix of organisms by functions. By organizing the ensuing genomic information into GFCs, we further simplified the interpretation of complex genomic datasets, while at the same time highlighting the non-trivial grouping of organisms by phenotypic traits, sometimes irrespective of taxonomic boundaries. The LTCs provided evidence for the functional and evolutionary linkage between traits, raising hypotheses as to how these traits act together in the context of complex processes such as microbial interactions. This approach can be easily scaled to different ecosystems (e.g. freshwater or terrestrial) and expanded to include information from additional data sources (e.g. metabolomics or high-throughput functional assays).

Applying this approach to a wide diversity of bacterial taxa, we showed that marine bacteria encode different configurations of interaction traits and can, therefore, engage in different interaction types. Known particle associated taxa of Alpha- and Gammaproteobacteria possessed the full set of traits to interact with particles and living hosts, while Bacteroidota, a known ubiquitous copiotroph taxon, did not have this capacity. Actinobacteriota and Cyanobacteria represented potential sources for B7 in the B vitamin market, while most Alphaproteobacteria appeared as obligate customers. We suggest that siderophores, and vibrioferrin in particular, are keystone molecules being produced by only a few bacteria (Actinobacteriota and Gammaproteobacteria) but affecting a much larger diversity of potential users, in agreement with the Black Queen Hypothesis⁶⁷. Finally, the production of IAA might be more common than expected, and in some cases (e.g. GFCs 9, 30 and 40 encoding LTC 19) this may be linked with other traits involved in phytoplankton-bacteria interactions. ”.

Lines 500-508 + Lines 511-514. I think any study may have imperfections, because scientific research proceeds even in small steps. However, a so high level of uncertainty showed by the Authors in an investigation based almost exclusively on the development of a bioinformatics approach, in my opinion, greatly weakens the study.

Lines 517-518. I agree that this approach can be useful for ecologists, but an ecologist first wants to understand if this approach works and what answers it can provide and to what extent.

As described in detail above, we now provide additional support for the reliability and robustness of our results, while maintaining transparency with regard to its limitations: (L404-424) “As with any bioinformatic approach, also our workflow aimed to identifying functional traits on genome sequences, has inherent limitations. Firstly, on average, only 63% of genes were functionally annotated across genomes (range ~40-80%), with the remaining genes either annotated as hypothetical or not annotated at all. This is a strong reminder of the limitations of current genomic and metabolic knowledge. Secondly, only a fraction of the natural microbial diversity was represented by high-quality sequenced genomes, leading to a bias in the representation of different clades in the analysed dataset (Figure 1). While the analysed genomes did represent a significant fraction of natural diversity (Supplementary Figs 4 and 5), a future work, both culture dependent and independent, is needed to close these gaps and obtain an unbiased view of the traits encoded by marine microorganisms. Finally, the GFC and LTC concepts are both statistical in nature, representing the probability of bacteria having similar functional capacity, and of traits being functionally and/or evolutionarily linked. In support of the GFC concept, a study of marine Vibrionaceae suggested that ecologically-cohesive unit (similar to GFCs) are likely to interact in a similar way²⁹. However, some interaction phenotypes, such as the ability to inhibit multiple target bacteria (“super-killers”), are not phylogenetically conserved^{29,113}. Experimental studies using both established and new model systems across multiple scales of diversity (e.g. between and within

GFCs) are needed to test the GFC framework, whereas genetic manipulation of linked traits to test their effect on microbial interactions will be required to determine to what extent traits within LTCs are functionally linked. Nevertheless, GFCs and LTCs describe how putative interaction traits vary across microbial diversity, and thus can be used to quantify how the fraction of natural communities potentially “ready to interact” changes over space and time. In turn, this should help to elucidate fundamental rules that govern community dynamics and assembly in the oceans, and the roles played by microbial interactions will in global ecosystem-level processes and biogeochemistry.”

Materials and Methods

Could you please clarify the number of pre-screening genomes considered? If I understood well, 473 is the number of genomes obtained after screening. I’m wondering why you obtained only 473 genomes of high-quality and not many more as one would might expect.

Please see the answers above (to the Editor and to general comments of Reviewer #3).

How did you select the genomes from planktonic, benthic, extreme environments and so on?

Our main focus was on marine pelagic bacteria whose genomes (~90% of the dataset) were identified filtering for relevant keywords the related metadata available in NCBI, JGI and MegX repositories. We also included some genomes isolated from related environments (e.g. marine sediment, extreme marine habitats) as “out group comparison”, or from plant roots (i.e. Rhizobacteria) as they represent well known symbiotic bacteria, that were identified in the same way.

The analysis of pathway completeness is dependent upon the use of the gene prediction software, (i.e., Prokka). This software is perfectly suitable for broad-scale prokaryotic genome annotation, but it's not accurate and (also by the authors' admission) requires manual re-annotation and review, which can add a further level of risk. In addition, the heavy reliance of the authors on KEGG (which is very refined but fails to correctly assign more distantly-related protein sequences) might lead to biases in the completeness analysis. More recent advancements, introduced by programs such as DRAM or METABOLIC, could allow a more automated, reliable and reproducible analysis for the scope of the present study.

We thank the Reviewer for the suggestion, however, as answered at the beginning to the Editor, we respectfully disagree with this recommendation. While these algorithms certainly represent exciting new approaches, they are relatively untested, certainly in comparison with Prokka, which is a highly used and very robust tool (e.g. the manuscript describing DRAM has been cited 23 times, whereas that describing Prokka has been cited more than 6,300 times). Furthermore, our pipeline for assessing the completeness of KEGG pathways is simple, clear and highly reproducible. Finally, we manually annotated only specific traits involved in microbial interactions, which are not covered, to the best of our knowledge, by other tools, and guidelines for our annotation are clarified and details are given in the materials and methods as well as in our GitHub repository. Taken together, we argue that our approach is highly robust, highly reproducible and represents the state of the art in the field.

Reviewers' comments:

Reviewer #1 (Remarks to the Author):

The authors have done a great job improving their manuscript. The new version reads significantly better than the first and overall I think this paper will advance our knowledge. I have a few minor comments that once addressed I endorse the manuscript for publication.

In Supplementary Text (lines 191-197), B vitamins are water soluble indeed. However, the rule is hydrophilic molecules require active transport, while hydrophobic molecules can diffuse through cell membranes. Please correct.

Main text:

Lines 67-71: You should also point to studies that examine these interactions directly in the field through use of Metagenomics or other meta-omics analyses.

Lines 139-142: Perhaps they occupy a unique/specific niche? Could you elaborate on the paraphyletic GFCs?

Lines 161-162: I understand this was done to respond to another reviewer, but I'd avoid asking the reader a question in the main text. Perhaps move this sentence as the section title, or rephrase/remove.

Lines 165-167: Can your workflow be useful in metagenomics? If so, it would be nice to highlight this somewhere in the discussion or summary.

Lines 352-353: I like the incorporation of quorum sensing here. Can you distinguish signal synthesizers (luxI gene) vs responders (luxR)? Due to the prevalence of solo luxR genes in bacteria, it would be very useful knowing if you can distinguish between the two in the GFCs?

Lines 407-410: These sentences are contradictory. Rephrase.

Reviewer #2 (Remarks to the Author):

The authors should revise the manuscript and code (https://github.com/lucaz88/genome_comparison_code) according to comments below.

#) "core" traits

From a conceptual point of view, the LTC classes "core", "common" and "ancillary" are similar to the pan-genome classes of core genome; accessory or dispensable genome and species-specific or strain-specific genes. However, the pan-genome analysis is usually carried out at species and genus levels while in our analysis we compared genomes of different phyla. We therefore decided not to use the pan-genome classes to avoid confusion and to allow us to focus on the interaction traits.

- To avoid confusion, it is better to mention the relationship of the pan-genome classes with the interaction traits.

Here's a quote from Vernikos et al. (2015) Review paper "Ten years of pan-genome analyses":

<https://www.sciencedirect.com/science/article/pii/S1369527414001830?via%3Dihub>

For example, from the phylogenetic resolution point of view, there are projects focused on the species

level, genus level, and even at the class, phylum or super kingdom levels (Table 1).

#) Reproducible research

We have amended the missing information in the methods, as also suggested by the Reviewer in the following comments. Furthermore, we made available the code used to perform the different functional annotations, reconstruct complete KEGG modules from KEGG orthologues and the statistical analysis for achieving the GFCs and LTCs (see the updated repository https://github.com/lucaz88/genome_comparison_code).

#) in-house R script

We could not replicate the error with the information provided by the reviewer. However, we did substantially improve our code repository on GitHub by adding new scripts, providing more clear instructions (on the web-page and within the scripts) on how to use and test them. We also test the mentioned code in 'KM_reconstruction.R' on different laptops to ensure it works well.

- I wonder if the authors could provide their environments. The R function `sessionInfo()` prints version information about R, the OS and attached or loaded packages. I tried "USAGE EXAMPLE (type commands directly in R)" at https://github.com/lucaz88/genome_comparison_code. However, the commands printed the Error messages again. Please see Review Attachments <2021-07-10.zip>.

Running the R script `*my_analysis.R*` from the command line (Running R batch mode on MacOS) printed the following Error messages as shown in the `*log.txt*` file.

```
```\nError in .getUrl(url, .flatFileParser) : Forbidden (HTTP 403).\nCalls: KMdiagram_fetcher ... resolve.list -> signalConditionsASAP -> signalConditions\nExecution halted\n```\n
```

Running the R script `*my_analysis.R*` on RStudio printed the following Error messages:

```
```\n> KM_str <- KMdiagram_fetcher(ncore = 7, create_RData = T, path = "./genome_comparison_code-main") # it takes a few minutes\nLoading required package: future\nError in .getUrl(url, .flatFileParser) : Forbidden (HTTP 403).\nIn addition: Warning message:\nIn supportsMulticoreAndRStudio(...) : \n[ONE-TIME WARNING] Forked processing ('multicore') is not supported when running R from RStudio because it is considered unstable. For more details, how to control forked processing or not, and how to silence this warning in future R sessions, see ?parallely::supportsMulticore\nCalled from: signalConditions(obj, exclude = getOption("future.relay.immediate", "immediateCondition"), resignal = resignal, ...)\nBrowse[1]>\n
```

```
```\n> KM_str <- KMdiagram_fetcher(ncore = 7, create_RData = T, path = "./genome_comparison_code-main") # it takes a few minutes\nError in .getUrl(url, .flatFileParser) : Forbidden (HTTP 403).\nCalled from: signalConditions(obj, exclude = getOption("future.relay.immediate", "immediateCondition"), resignal = resignal, ...)\nBrowse[1]>\n```\n
```

-----  
#) L139-152: "Overall, half of the detected GFCs were monophyletic support the existence of a strong correlation between taxonomy and functionality in marine bacteria, while the remaining paraphyletic GFCs highlight that, in some cases, the taxonomic partitioning (based on the Genome Database Taxonomy 49) do not completely reflect the functional differentiation.

- I wonder if "monophyletic" and "paraphyletic" were defined base on some phylogenetic tree? If this is not the case, "monophyletic" and "paraphyletic" should be rephrased.

Reviewer #3 (Remarks to the Author):

The present study entitled "Comparative whole-genome approach to identify bacterial traits for microbial interactions" by Zoccarato et al., has been extensively revised, after the first round of review, and in the present version the results and discussion have been improved. The Authors have provided a great effort to address the main concerns of the Reviewers, but I still have some concerns about the take-home message of this study, which in my opinion remains unclear and is too generic. Another critical issue deals with the level of uncertainty of the results obtained, which is highlighted by the Authors in the results and discussion section. I appreciate the "transparency" in identifying the potential bias linked to the bioinformatic analysis and to the limits in the representation of the natural biodiversity of the sequenced genomes, but the uncertainty on which the data of the ms are based limits the robustness of the results obtained (e.g., potential mechanisms, traits potentially involved in microbial interactions, potential antagonist interactions, processes that likely play, functions that likely evolved together) and their applicability to other data, since the Authors state that other studies are needed to test GFC framework and to determine to what extent traits within LTCs are functionally linked.

Abstract.

The Authors did not succeed in identifying the take-home message of their study. The abstract contains generic sentences (lines 29-34) and, except for the application of a new approach, it is not clear what they actually found new.

The Authors, indeed, reported a "unique combinations of interaction traits, some widely distributed, others less common", that two main classes of proteobacteria are predicted to preferentially interact synergistically and/or antagonistically with bacteria and phytoplankton, and that "linked trait clusters" identify traits that may have evolved together and point to specific modes of interactions. A reader cannot understand the new findings for scientific advancement over the state of the art.

Introduction.

Line 69... ". relatively little is known regarding how widely distributed such interactions are across microbial diversity" Please use...different microbial taxa. The expression "across microbial diversity" is often used throughout the text but is incorrect.

Did you include in this definition also the interactions with phytoplankton?

Line 72 .... microbial interactions. With what? With phytoplankton? Please, specify.

Line 84-87. Why did the authors focus only on these traits? Please, you should motivate and frame these goals within the introduction based primarily on bacteria-phytoplankton interactions.

Line 91-93. So, which are the new insights provided by the study?

Results and discussion.

Line 118-119. "Pseudoalteromonas and Alteromonas bore more genetic traits..." Compared to what? Is this referred to all species belonging to these genera?

Lines 123-125. This statement is not supported.

Lines 173-175. Please, revise. This statement is inferential.

Line 184. Please, please add a reference here or clarify.

Lines 188-191. Are all Bacteroidota copiotrophs? What could be the implications of this findings considering the objectives of the study?

Line 239. Please, add a reference after "interactions".

Lines 242-244. Please, rephrase.

Figure 4. This conceptual model should be a sort of final overview of the results of the ms, but I think that it should be better explained in the last paragraph of the manuscript. In this model the Authors include interactions of an hypothetical bacterial cell with zooplankton, illustrating a copepod, and in the legend they also mention other eukaryotes such as fish, and, organic matter (I suppose, since this is missing from the legend). Why did the authors based the introduction of their study almost exclusively on the interactions between phytoplankton and bacteria if they also include other organisms in the final conceptual model?

Just from an aesthetic point of view I would like to suggest to improve the symbols that represent the phytoplankton.

Lines 355-356. Please, clarify. This statement is too generic and unclear.

Lines 393-394. Again, saying that marine bacteria encode different configurations of interaction traits and can engage in different types of interactions is like saying all without saying anything.

Lines 408-409. How much does this biased representation affect the actual results of this study?

Lines 415. The Authors continue to use this expression "interact in a similar way", but this expression is unclear and sounds vague.

## Answers to the reviewers' comments:

Please note that the line numbers provided in the following answer apply only to the manuscript with the track changes in display.

### Reviewer #1 (Remarks to the Author):

The authors have done a great job improving their manuscript. The new version reads significantly better than the first and overall I think this paper will advance our knowledge. I have a few minor comments that once addressed I endorse the manuscript for publication.

We thank the reviewer for her/his positive view and address her/his minor remarks as described in detail below.

In Supplementary Text (lines 191-197), B vitamins are water soluble indeed. However, the rule is hydrophilic molecules require active transport, while hydrophobic molecules can diffuse through cell membranes. Please correct.

We have rephrase the sentence as follow (L203-206): "Moreover, one should keep in mind that B vitamins are water soluble molecules and, although they cannot passively diffuse through cell membranes, they may became available for other bacteria upon lysis of the producing cell. "

Main text:

Lines 67-71: You should also point to studies that examine these interactions directly in the field through use of Metagenomics or other meta-omics analyses.

We added a statement on field studies: (L75-78) "*The few experimental studies that measure microbial interactions across different taxa (e.g. 33-35 ) are usually constrained to a fairly narrow phylogenetic scope and are performed under conditions different from natural marine environments. Conversely, relevant field studies are still quite limited (e.g. 11,36 ).*"

Lines 139-142: Perhaps they occupy a unique/specific niche? Could you elaborate on the paraphyletic GFCs?

As discussed also in response to a remark from reviewer #2, we have decided, for simplicity, to merge the polyphyletic and paraphyletic definitions into a single “non-monophyletic” term. This results in minor changes to Supplementary figures 3d, 7, 12 and 14. With regard to the specific question of the reviewer, it is possible that convergent evolution (including e.g. horizontal gene transfer ) could lead to non-monophyletic clades, as we now state clearly at L157-160: “*Such discrepancy may be due to processes of convergent evolution (e.g., via horizontal gene transfer) which have the highest occurrence in some of the niches known to be occupied by bacteria grouped in specific non-monophyletic GFCs (e.g. inhabiting extreme environments, particles and biofilms; see Supplementary text)* 54-56 .” as well as in the Supplementary text: (L141-151) “*Non-monophyletic GFCs included organisms from multiple genera, families or even phyla. For example, GFCs 33 and 41 grouped organisms belonging to different phyla, whose genomes were isolated from extreme environments (e.g., thermal vents or hyper saline environments). Although these genomes were added in the analysis as outer groups and their taxonomic and functional diversity was not adequately covered, extreme environments are known hotspots for gene exchange, e.g., horizontal gene transfer. These processes, in turn, favour functional convergent evolution even between distantly related organisms* 29 . *Similar processes of gene exchange are known to occur at higher rates in bacteria occupying specific niches such as biofilm-forming and particle-associated bacteria* 30,31 *suggesting that convergent evolution might explain in part the non-monophyletic nature of some GFCs, e.g., 24 (Alteromonas) and 47 (Vibrio) which group bacteria with the respective lifestyles* 32,33 .”

Lines 161-162: I understand this was done to respond to another reviewer, but I'd avoid asking the reader a question in the main text. Perhaps move this sentence as the section title, or rephrase/remove.

We have rephrased the question as a statement.

Lines 165-167: Can your workflow be useful in metagenomics? If so, it would be nice to highlight this somewhere in the discussion or summary.

We thank the reviewer for this suggestion. We now mention the possibility of using our framework for the analyses of metagenomes and high-quality MAGs in the summary section: (L427-430) “*It also*

*facilitates the investigation of the functional and interaction potential of metagenomes (e.g., to identify communities where interactions might be more relevant than others) and high-quality metagenome assembled genomes in field studies."*

Lines 352-353: I like the incorporation of quorum sensing here. Can you distinguish signal synthesizers (luxI gene) vs responders (luxR)? Due to the prevalence of solo luxR genes in bacteria, it would be very useful knowing if you can distinguish between the two in the GFCs?

We thank the reviewer for this interesting question. The traits related to quorum sensing that we mentioned in these lines do not directly include the luxI/R system, as there is no related KEGG module (actually there are multiple KOs – see below). Instead, the mentioned quorum sensing traits are involved in the signal response and include the QseC-QseB quorum sensing two component regulatory system (KEGG module M00453), the LuxQN/CqsS-LuxU-LuxO regulatory system (M00513), and the SagS-HptB-HsbR regulatory system (M00820). Additional quorum-sensing traits related to the signal synthesis were captured using the pipeline for the identification of secondary metabolites (antiSMASH) and include the biosynthetic clusters of homoserine lactone and butyrolactone.

Aiming to directly address the reviewers' remark, we applied our bioinformatics approach to try and unequivocally identify LuxI/R systems using their KEGG annotations. However, multiple KO terms are associated with both the luxI and luxR genes (annotated with kofamscan) – 11 terms are associated with LuxI and 29 with LuxR. When analysing the presence of these KOs, there are indeed more genomes possessing the luxR genes (130) than luxI (71) (see figure below). There are also some interesting patterns in the distribution of these genes: luxI genes were detected only in Alpha- and Gammaproteobacteria, whereas luxR genes are found also in other clades. However, we were worried that multiple KOs do not map on a one-to-one basis with the genes, and thus may include many false positives. Additionally, because of the structural homology of the luxI/R proteins to proteins not involved in quorum sensing (e.g., transcription factors), specialized tools have generally been used to study their distribution among microbial genomes (e.g., Rajput and Kumar, 2017, Scientific Reports; Wengeret et al., 2021, BMC Research Notes). In light of these results, which are overall inconclusive and which would require more substantial follow up, we opted not to include any of these data in the current manuscript.

Figure for reviewer #1: Distribution of luxI and luxR genes across genome functional clusters (GFCs) expressed as the number of gene-bearing genomes over the number of clustered genomes. We assessed the presence of KOs related to the luxI (K13060, K13061, K13062, K20248, K20249, K20250, K22954, K22955, K22956, K22957, K22968; <https://www.genome.jp/entry/R08940>) and to the luxR (K03556, K21748, K15852, K04333, K01994, K07781, K18304, K18099, K18098, K07782, K19666, K20540, K19731, K19732, K19733, K19734, K20252, K20253, K20334, K21697, K20330, K20918, K21963, K21685, K21901, K22650, K24912, K16247, K21907; <https://www.genome.jp/brite/ko03000> Prokaryotic type → Helix-turn-helix → LuxR family) genes directly on the KO annotation generated by kofamscan.

Lines 407-410: These sentences are contradictory. Rephrase.

We have rephrased the sentences as follow: (L603-607) *“Moreover, although the analysed genomes represented a significant fraction of bacterial taxa in marine environments (Supplementary Figs 4 and 5), we still lack high-quality genomes for many taxa. Future work, both culture-dependent and independent, is required to obtain an unbiased view of the numerous traits encoded in marine microorganisms.”*

We also moved those sentences in a dedicated section on bioinformatics caveats at the end of material and methods section to provide a more concise summary and outlook, as suggested by reviewer #3.

## Reviewer #2 (Remarks to the Author):

The authors should revise the manuscript and code ([https://github.com/lucasz88/genome\\_comparison\\_code](https://github.com/lucasz88/genome_comparison_code)) according to comments below.

-----

#) "core" traits

From a conceptual point of view, the LTC classes "core", "common" and "ancillary" are similar to the pan-genome classes of core genome; accessory or dispensable genome and species-specific or strain-specific genes. However, the pan-genome analysis is usually carried out at species and genus levels while in our analysis we compared genomes of different phyla. We therefore decided not to use the pan-genome classes to avoid confusion and to allow us to focus on the interaction traits.

- To avoid confusion, it is better to mention the relationship of the pan-genome classes with the interaction traits.

Here's a quote from Vernikos et al. (2015) Review paper "Ten years of pan-genome analyses":

<https://www.sciencedirect.com/science/article/pii/S1369527414001830?via%3Dihub>

For example, from the phylogenetic resolution point of view, there are projects focused on the species level, genus level, and even at the class, phylum or super kingdom levels (Table 1).

We have rephrased the paragraph that introduces the LTC concept and we now clarify the parallelism with the pan-genome analysis. L353-356: *"Similar to pangenome analyses, we divided all LTCs into "core" (present in >90% of genomes), "common" (<90% and ≥30%) and "ancillary" (≤30%; Supplementary Fig. 13C). Note that, while pangenome analysis is based on single gene distributions, each LTC included different genetic traits and each trait often involved >3 genes."*

-----

#) in-house R script

- I wonder if the authors could provide their environments. The R function `sessionInfo()` prints version information about R, the OS and attached or loaded packages.

I tried "USAGE EXAMPLE (type commands directly in R)" at [https://github.com/lucasz88/genome\\_comparison\\_code](https://github.com/lucasz88/genome_comparison_code). However, the commands printed the Error messages again.

Please see Review Attachments <2021-07-10.zip>.

Running the R script `*my_analysis.R*` from the command line (Running R batch mode on MacOS) printed the following Error messages as shown in the `*log.txt*` file.

```
...`
```

```
Error in .getUrl(url, .flatFileParser) : Forbidden (HTTP 403).
```

```
Calls: KMdiagram_fetcher ... resolve.list -> signalConditionsASAP -> signalConditions
```

```
Execution halted
```

```
...`
```

Running the R script `*my_analysis.R*` on RStudio printed the following Error messages:

```
...`
```

```
> KM_str <- KMdiagram_fetcher(ncore = 7, create_RData = T, path = "./genome_comparison_code-main") # it takes a few minutes
```

```
Loading required package: future
```

```
Error in .getUrl(url, .flatFileParser) : Forbidden (HTTP 403).
```

```
In addition: Warning message:
```

```
In supportsMulticoreAndRStudio(...) :
```

```
[ONE-TIME WARNING] Forked processing ('multicore') is not supported when running R from RStudio because it is considered unstable. For more details, how to control forked processing or not, and how to silence this warning in future R sessions, see ?parallely::supportsMulticore
```

```
Called from: signalConditions(obj, exclude =
getOption("future.relay.immediate",
```

```
"immediateCondition"), resignal = resignal, ...)
```

```
Browse[1]>
```

```
> KM_str <- KMdiagram_fetcher(ncore = 7, create_RData = T, path
= "./genome_comparison_code-main") # it takes a few minutes
```

```
Error in .getUrl(url, .flatFileParser) : Forbidden (HTTP 403).
```

```
Called from: signalConditions(obj, exclude =
getOption("future.relay.immediate"),
```

```
"immediateCondition"), resignal = resignal, ...)
```

```
Browse[1]>
```

```
```\n
```

We thank the reviewer for the comment and detailed information about the error. After several tests, we managed to identify the issue in the code which is apparently occurring only in non-Linux systems. Since we mainly use Linux-based computer for our bioinformatics analyses, we failed to detect this issue in the previous round of revisions. We have now updated the code in the companion GitHub repository (https://github.com/lucaz88/genome_comparison_code), providing also the `sessionInfo()` on our running environment. We also give a more user-friendly code to run the text example.

#) L139-152: "Overall, half of the detected GFCs were monophyletic support the existence of a strong correlation between taxonomy and functionality in marine bacteria, while the remaining paraphyletic GFCs highlight that, in some cases, the taxonomic partitioning (based on the Genome Database Taxonomy 49) do not completely reflect the functional differentiation.

- I wonder if "monophyletic" and "paraphyletic" were defined base on some phylogenetic tree? If this is not the case, "monophyletic" and "paraphyletic" should be rephrased.

Following the reviewers remarks, we consulted with two specialists in the field of phylogenetic reconstruction. Indeed, when considering a taxonomic group, we are actually referring to the phylogeny of the tree of species used to define such taxonomy. In our case, we used the GTDB-tk tool which assigned the genome taxonomy by placing the query genome into a domain-specific tree built by using 120 concatenated protein sequences. Therefore, when using the terms monophyletic or paraphyletic for a given cluster of genomes, we are

referring to the topology of the subset of a “real” phylogenetic tree, and the use of these terms is warranted. Nevertheless, after carefully revising our analysis assessing the taxonomic coherence of GFCs, we chose for simplicity to merge the paraphyletic and polyphyletic classes into the “non-monophyletic” class defined as (L149-152) “*The remaining non-monophyletic GFCs, contained genomes of multiple taxa (differing at genus, family or even phylum rank) or included taxa which were partitioned among multiple GFCs.*”

Reviewer #3 (Remarks to the Author):

The present study entitled “Comparative whole-genome approach to identify bacterial traits for microbial interactions” by Zoccarato et al., has been extensively revised, after the first round of review, and in the present version the results and discussion have been improved.

The Authors have provided a great effort to address the main concerns of the Reviewers, but I still have some concerns about the take-home message of this study, which in my opinion remains unclear and is too generic. Another critical issue deals with the level of uncertainty of the results obtained, which is highlighted by the Authors in the results and discussion section. I appreciate the “transparency” in identifying the potential bias linked to the bioinformatic analysis and to the limits in the representation of the natural biodiversity of the sequenced genomes, but the uncertainty on which the data of the ms are based limits the robustness of the results obtained (e.g., potential mechanisms, traits potentially involved in microbial interactions, potential antagonist interactions, processes that likely play, functions that likely evolved together) and their applicability to other data, since the Authors state that other studies are needed to test GFC framework and to determine to what extent traits within LTCs are functionally linked.

We are glad the reviewer found the revised manuscript improved and thank her/him for her/his help in focusing on the take-home messages of the manuscript by removing parts that are “too generic”. Based on the reviewer’s remarks we have re-phrased and clarified multiple parts of the manuscript to better highlight the specific take-home messages (see below, and Lines 136-139, 192-194, 210-212, 380-389, 431-434). With regard to the uncertainties and caveats of the study – as discussed previously, we think that the results presented in our study are robust, informative, and important steps in understanding how microbial interactions vary across different

organismal lineages. Nevertheless, we acknowledge again the limited certainty that any computational study like ours can achieve without further dedicated experimental testing. We did our best to express as transparently as possible the limitations and caveats of our approach. At the same time, we contend that computational studies, as long as they are carried in a rigorous and reproducible manner do not have to be necessarily tested or validated within the paper presenting them. In fact, an important role of computational biology is to put out there new predictions that can inspire future experimental measurements, and provide the ground for independent testing by other groups.

Abstract.

The Authors did not succeed in identifying the take-home message of their study. The abstract contains generic sentences (lines 29-34) and, except for the application of a new approach, it is not clear what they actually found new.

The Authors, indeed, reported a "unique combinations of interaction traits, some widely distributed, others less common", that two main classes of proteobacteria are predicted to preferentially interact synergistically and/or antagonistically with bacteria and phytoplankton, and that "linked trait clusters" identify traits that may have evolved together and point to specific modes of interactions. A reader cannot understand the new findings for scientific advancement over the state of the art.

We thank the reviewer for these comments which allowed us to re-elaborate several sentences of the abstract and to better highlight the relevant new findings in our manuscript. In particular: L28-38 "Most GFCs revealed unique combinations of interaction traits, including the production of siderophores (10% of genomes), phytohormones (3-8%) and different B vitamins (57-70%). Specific GFCs, comprising Alpha- and Gammaproteobacteria, displayed more interaction traits than expected by chance, and are thus predicted to preferentially interact synergistically and/or antagonistically with bacteria and phytoplankton. Moreover, linked trait clusters (LTCs) identify traits that may have evolved and act together (e.g., secretion systems, nitrogen metabolism regulation and B vitamin transporters), providing testable hypotheses for mechanisms of microbial interactions."

Introduction.

Line 69... “. relatively little is known regarding how widely distributed such interactions are across microbial diversity” Please use...different microbial taxa. The expression “across microbial diversity” is often used throughout the text but is incorrect.

Throughout the entire manuscript, we have changed the expression to “different bacterial taxa” or “multiple bacterial taxa”, as suggested by the reviewer.

Did you include in this definition also the interactions with phytoplankton?

Line 72 microbial interactions. With what? With phytoplankton? Please, specify.

We have rephrased three sentences in the introduction to clarify this point:

L58-60: *“Studies using specific model bacteria in binary co-cultures have started to elucidate mechanisms underlying specific interactions with other marine microbes (mostly phytoplankton, but also with zooplankton or other bacteria e.g. on particles)^{1,4,5,7,11} .”*

L72-75: *“While much is known about how model organisms interact with other bacteria and with phytoplankton (e.g., specific strains of *Roseobacter*^{14,19-21,25} , *Alteromonas*²⁷⁻²⁹ , *Vibrio*^{30,31} or *Cyanobacteria*^{20,32}), relatively little is known regarding how widely distributed the relevant interaction mechanisms are across natural bacterial taxa.”*

L95-96: *“2) specific gene pathways related to the main discovered mechanisms of bacteria-bacteria and bacteria-phytoplankton interactions, such as ...”.*

Line 84-87. Why did the authors focus only on these traits? Please, you should motivate and frame these goals within the introduction based primarily on bacteria-phytoplankton interactions.

We have rephrased the sentences to clearly state that a) the list of analysed traits depicts quite comprehensively the functional and metabolic potential of marine bacteria and b) the identified specific “interaction traits” which include most of the known metabolic, regulatory, signalling and interaction mechanisms previously described.

L91-98: *“We focused on the following traits: 1) KEGG modules representing the overall functional and metabolic capacity (i.e. pathways for the synthesis and degradation of specific biomolecules, or gene sets for processing of genetic and environmental information,*

cell signalling and drug resistance); 2) specific gene pathways related to the main discovered mechanisms of bacteria-bacteria and bacteria-phytoplankton interactions, such as motility, chemotaxis and the capability to produce molecules such as siderophores, phytohormones and antibiotics."

Line 91-93. So, which are the new insights provided by the study?

We thank the reviewer for this remark. We have slightly modified these sentences to emphasize that the LTC concept points to cooperating traits: (L102-105) "*Our approach maps the mechanisms of microbial interactions identified in model organisms across multiple bacterial taxa, suggests specific groups of bacteria likely to interact using similar trait combinations, and helps hypothesise how these traits act together to mediate microbial interactions.*". We did not make any more substantial/specific changes (e.g., describing specific insights), as these sentences come at the end of the introduction and are aimed to provide a wide-angle view of our approach. We believe that the specific new insights are presented in the abstract, results, discussion and conclusions in a much clearer manner in our revised version.

Results and discussion.

Line 118-119. "Pseudoalteromonas and Alteromonas bore more genetic traits..." Compared to what? Is this referred to all species belonging to these genera?

In comparison to *Marinobacter*. We have fixed the sentence: (L128-131) "*A detailed analysis of the traits found in each of the respective GFCs (Supplementary Fig. 2) suggested that Pseudoalteromonas and Alteromonas bore more genetic traits involved in the resistance against antimicrobial compounds, as well as regulation for osmotic and redox stresses in comparison to Marinobacter.*"

Lines 123-125. This statement is not supported.

We have changed the sentence by adding reminders of the specific examples discussed above that support our statement: (L136-139) "*Overall, our GFC framework recapitulates previous knowledge on bacterial groups with defined ecology and life history (e.g., the Pelagibacterales, different Cyanobacteria and Vibrio), and provides a way to delineate and characterize potentially new ecological groups (e.g., Alteromonas, Pseudoalteromonas and Marinobacter).*"

Lines 173-175. Please, revise. This statement is inferential.

We have rephrased the sentence to clarify that the statement is inferential: (L192-194) *"Assuming that similar temporal trends suggest similar ecological niches, these results advocate that (at least some of) the GFCs display dynamics which are expected from ecological units in the oceans."*

Line 184. Please, please add a reference here or clarify.

We modified the text by clearly stating our working hypothesis with the relevant reference and rephrasing the sentence that presents our new results (L201-206) *"As the number of genes is strongly correlated with genome size ⁶⁴ , we expected that large genomes may encode for more interaction traits than small genomes, as previously demonstrated, e.g., for the biosynthetic pathways of secondary metabolites ^{64,65} . However, while the number of interaction traits depended to some extent on genome size, we found that Gammaproteobacteria and several Alphaproteobacteria encoded more interaction traits than expected just by their genome size, while Bacteroidota encoded fewer (Supplementary Fig. 7c-d)."*

Lines 188-191. Are all Bacteroidota copiotrophs? What could be the implications of this findings considering the objectives of the study?

We have added a reference (Ho et al., 2017, FEMS) to further support our statement that Bacteroidota are copiotroph and we have modified the sentence to clearly show the implication of this finding. L210-212: *"Conversely, some ubiquitous copiotrophs (e.g. Bacteroidota) ^{59,66} and known free-living taxa (e.g., pico-Cyanobacteria and Pelagibacterales) ^{30,67} possess only a scarce and scattered combination of such traits and are expected to exhibit a rather independent lifestyle."*

Line 239. Please, add a reference after "interactions".

We have added the missing references.

Lines 242-244. Please, rephrase.

We have modified the sentence as follow: (L265-268) "*Occurrence of siderophore biosynthetic traits was partially consistent with GFC clustering (e.g., nearly all genomes in GFCs 8 and 25 possessed those traits) and partially scattered across single genomes in different GFCs.*"

Figure 4. This conceptual model should be a sort of final overview of the results of the ms, but I think that it should be better explained in the last paragraph of the manuscript. In this model the Authors include interactions of an hypothetical bacterial cell with zooplankton, illustrating a copepod, and in the legend they also mention other eukaryotes such as fish, and, organic matter (I suppose, since this is missing from the legend). Why did the authors based the introduction of their study almost exclusively on the interactions between phytoplankton and bacteria if they also include other organisms in the final conceptual model?

Just from an aesthetic point of view I would like to suggest to improve the symbols that represent the phytoplankton.

We agree with the reviewer's remark, and have added a sentence to the introduction to clarify that, while we focus on bacteria-phytoplankton interactions (as these are the most well understood), we also include other types of interactions: (L58-60) "*Studies using model bacteria in binary co-cultures have started to elucidate the mechanisms underlying specific interactions with other marine microbes (mostly phytoplankton, but also zooplankton or other bacteria, e.g., on particles)* ^{1,4,5,7,11} .".

We have modified the caption of Figure 4 and have added a sentence in the related discussion to clarify this point: "*Fig. 4: Conceptual representation of the predicted interaction dynamics (analysed with our trait-based approach) between a hypothetical bacterium and phytoplankton cells (LTC 19), other bacteria on an organic particle (LTC 11) and other eukaryotic host such as zooplankton (LTC 17). The bacterium is assigned to a GFC, ...*".

L394-397: "*LTC 17 was also found in GFCs that grouped non-pathogenic but still host associated taxa (GFC 25, which includes *V. alginolyticus*, and GFC 13 grouping *Enterobacteriaceae*)* ¹¹⁴ , supporting the role of this LTC (as well as of the T3SS it encodes) in interactions with a broad range of eukaryotic hosts including zooplankton, phytoplankton and fish (see Fig. 4). "

We have improved the symbols representing the phytoplankton in Fig. 4 according to the reviewer's suggestion. Also, we have added clear references to the conceptual model in Fig. 4 within the first

paragraph of the summary section: (L417-418) *"We present a framework that extrapolates from studies of specific model organisms to predict the interaction potential of other bacteria based on the traits encoded in their genomes (see Fig. 4)."* and (L425-426) *"This approach can be easily scaled to different systems such as freshwater, terrestrial or other host microbiomes (e.g., zooplankton and fish; see LTC 17 in Fig. 4), and expanded ..."*.

Lines 355-356. Please, clarify. This statement is too generic and unclear.

We have rephrased the sentence: (L380-389) *"Interestingly, the other two secretion systems, T4SS and T3SS, were also linked with a regulation system for nitrogen metabolism and with vitamin B7 or B12 transporters as part of LTC 25 and LTC 17, respectively. We propose that the linkage between these traits across different LTCs suggests that, in multiple interactions, these processes occur together. In principle, this could be a direct link in which the injection of an effector molecule modifies the response, for example, to nitrogen starvation (as shown for phosphate starvation in response to the toxin cylindrospermopsin, ¹¹³). However, the linkage between these traits may also be the result of complex interactions that require the coordinated exchange of multiple metabolites and signals (e.g., ¹⁴).*

Lines 393-394. Again, saying that marine bacteria encode different configurations of interaction traits and can engage in different types of interactions is like saying all without saying anything.

We have removed the last part of the sentence as it was indeed too generic. The main differences in the interaction capabilities and mechanisms are highlighted in the following sentences. L431-434: *"Applying this approach to a wide diversity of bacterial taxa, we show that marine bacteria encode different configurations of interaction traits. Known particle associated taxa of Alpha- and Gammaproteobacteria possess the full set of traits to interact with particles and living hosts, ..."*

Lines 408-409. How much does this biased representation affect the actual results of this study?

As asked also by Rev#1, we have rephrased this sentence to clarify our statement. L603-607: *"Moreover, although the analysed*

genomes represented a significant fraction of bacterial taxa in marine environments (Supplementary Fig.s 4 and 5), we still lack high-quality genomes for many taxa and future work, both culture dependent and independent, is required to obtain an unbiased view of the numerous traits encoded in marine microorganisms.”. Thus the analysed genomes did represent a significant fraction of bacterial taxa in marine environments and we have demonstrated that the detected patterns are robust (see discussion ‘GFC & LTC clustering robustness’ in the supplementary text). The future addition of more high-quality genomes from yet uncultured marine bacteria will likely enrich the picture of our ‘Atlas of Marine Microbial Functional Traits’ rather than subverting it. We moved this paragraph dealing with caveats of the bioinformatics approach in a dedicated section at the end of material and methods section to sharpen the summary and outlook section and render it more concise.

Lines 415. The Authors continue to use this expression “interact in a similar way”, but this expression is unclear and sounds vague.

We have changed this expression with “interact using similar trait combinations”.

REVIEWERS' COMMENTS:

Reviewer #1 (Remarks to the Author):

I'm happy with the current version of the manuscript and thus endorse it for publication.

Reviewer #2 (Remarks to the Author):

Please see Review Attachments. Running the R script `*my_analysis.R*` on Linux and MacOS printed Error messages as shown in the `*log.CentOS_Linux_7.txt*` and `*log.macOS_Big_Sur_10.16.txt*` files. I wonder if the authors can test on computers other than Ubuntu (e.g. CentOS and macOS) to update the code at https://github.com/lucaz88/genome_comparison_code.

Reviewer #3 (Remarks to the Author):

The Authors of the manuscript entitled "Whole genome comparative approach to identify bacterial traits for microbial interactions" by Luca Zaccarato et alii, made a great effort to respond all the reviewers' requests and to improve the work. Therefore, I'm happy to endorse its publication in Communications Biology.

REVIEWERS' COMMENTS:

Reviewer #1 (Remarks to the Author):

I'm happy with the current version of the manuscript and thus endorse it for publication.

Reviewer #2 (Remarks to the Author):

Please see Review Attachments. Running the R script `*my_analysis.R*` on Linux and MacOS printed Error messages as shown in the `*log.CentOS_Linux_7.txt*` and `*log.macOS_Big_Sur_10.16.txt*` files. I wonder if the authors can test on computers other than Ubuntu (e.g. CentOS and macOS) to update the code at https://github.com/lucaz88/genome_comparison_code.

We have run a new debugging of the code with multiple users, different OS and different versions of R (see tables below). We confirm that the code runs successfully on every system (Linux, mac and Windows) when the users prompted the commands from a RStudio environment. In contrast, when running the code from a "stand-alone" instance of R, we could replicate the error reported by Reviewer #2 when using the macOS (the error associated to CentOS is likely due to an issue with the Reviewer's conda environment, see a similar error report at <https://github.com/satijalab/seurat/issues/2513>).

We have identified the function leading to this issue (`read_html` from `xml2` package) and contacted the developer of the package to find a solution. Nevertheless, we argue that our code is properly working when executed from Rstudio and, while waiting to issue a patch, we have added a remark in our GitHub repository to inform any potential user of this bug and how to avoid it. We will of course update the code once we will manage to find a solution for the bug.

User	R version	OS	RStudio	Execution	Issue
Luca	4.1.1	Ubuntu 20.04.3 LTS	V	V	None
Osnat	3.6.1	Win 10	V	V	None
Sven	4.1.1	Win 10 Pro	V	V	None
Julia	4.0.5	Ubuntu 18.04.5 LTS	V	V	She had to install libxml2-dev (sudo apt-get install libxml2-dev)
Jason	3.6.0	Win 10	V	V	None
Pau	4.1.1	Ubuntu 20.04	V	V	None
Doris	4.0.3	Win 10	V	V	None
Minoru	3.6.2	Mac OS Monterey 12.0.1	V	V	None
Rev#2	3.5.1	CentOS 7	?	X	"Error in if (nzchar(SHLIB_LIBADD)) SHLIB_LIBADD else character) :" This seems to be an issue related to conda and to how the reviewer has installed R on is system (see https://github.com/satijalab/seurat/issues/2513), therefore is not related to our code.
Rev#2	4.0.5	macOS Big Sur 10.16	?	X	"Error in curl::curl_fetch_memory(url, handle = handle) : "
Luca	4.1.1	Ubuntu 20.04.3 LTS	X	X	"Error in curl::curl_fetch_memory(url, handle = handle) : "
Minoru	3.6.2	Mac OS Monterey 12.0.1	X	X	"Error in curl::curl_fetch_memory(url, handle = handle) : "

Reviewer #3 (Remarks to the Author):

The Authors of the manuscript entitled "Whole genome comparative approach to identify bacterial traits for microbial interactions" by Luca Zaccarato et alii, made a great effort to respond all the reviewers' requests and to improve the work.

Therefore, I'm happy to endorse its publication in Communications Biology.